# Excised DNA circles from V(D)J recombination promote relapsed leukaemia

Zeqian Gao[1], James N. F. Scott[1,11], Matthew P. Edwards[1,8,11], Dylan Casey[1], Xiaoling Wang[1,9], Andrew D. Gillen[1,10], Sarra Ryan[2], Lisa J. Russell[2], Anthony V. Moorman[3], Ruth de Tute[4], Catherine Cargo[4], Anthony M. Ford[5], David R. Westhead[1,6] & Joan Boyes[7✉]

Extrachromosomal DNA amplification is associated with poor cancer prognoses[1]. Large numbers of excised signal circles (ESCs) are produced as by-products of antigen receptor rearrangement during V(D)J recombination[2,3]. However, current dogma states that ESCs are progressively lost through cell division[4]. Here we show that ESCs replicate and persist through many cell generations and share many properties in common with circular extrachromosomal DNAs. Increased ESC copy numbers at diagnosis of B cell precursor acute lymphoblastic leukaemia were highly correlated with subsequent relapse. By taking advantage of the matching recombination footprint that is formed upon the generation of each ESC, we measured ESC persistence and replication and found increased ESC replication in patients who later relapsed. This increased replication is controlled by cell-intrinsic factors and corresponds to increased expression of DNA replication- and repair-associated genes. Consistent with high ESC levels having a role in disease progression, the number of mutations typical of those caused by the V(D)J recombinase–ESC complex was significantly increased at diagnosis in patients who later relapsed. The number of such mutations in genes associated with relapse increased between diagnosis and relapse, and corresponded to clonal expansion of cells with high ESC copy numbers. These data demonstrate that the by-product of V(D)J recombination, when increased in abundance, potently associates with the V(D)J recombinase to cause adverse disease outcomes.

Circular extrachromosomal DNAs (ecDNAs) are present in most cancer types and are associated with poor patient outcomes[1]. ecDNAs typically span 50 kb to 1 Mb and have low nucleosome densities and high levels of transcription, resulting in increased expression of oncogenes when they are present[5,6]. This confers a growth advantage to recipient cells and preferential retention of oncogene-expressing ecDNAs. ecDNAs replicate autonomously, approximately once per cell cycle[7]; however, the absence of a centromere results in their unequal segregation at mitosis[8,9], driving cancer heterogeneity and oncogene amplification in daughter cells[10]. This contributes to tumour evolution, treatment resistance and increased ecDNA copy numbers as the cancer progresses[11]. Consistent with this, ecDNAs predict 43% of high-grade oesophageal cancers and persist from early-stage to late-stage cancers[12].

V(D)J recombination is vital to generate diversity of immunoglobulin and T cell receptor (TCR) genes. It is catalysed by the recombination-activating gene (RAG) proteins RAG1 and RAG2, which bring complementary V, D or J gene segments into a synaptic complex by binding to their adjacent recombination signal sequences (RSSs)[2]. The RSSs consist of conserved heptamer and nonamer sequences separated by a non-conserved spacer of 12 ± 1 or 23 ± 1 bp, with recombination almost exclusively occurring between RSSs of different spacer lengths[13]. Following cleavage at gene segment–RSS boundaries, gene segments are joined to generate the variable exons of immunoglobulins or TCRs, whereas the intervening DNA is normally excised[2,3]. The signal sequences on the excised DNA are joined together, generating a signal joint (SJ) on an ESC[2,3] (Fig. 1a). ESCs from immunoglobulin light chain loci are typically 50 kb to 1 Mb in size.

Initially, ESCs were thought to be inert, non-replicative entities that become diluted through cell division[4]. However, ESCs are now known to trigger genome instability through two related mechanisms. First, RAGs reassociate with the ESC SJ to catalyse ESC reintegration at RSS-like sequences known as cryptic RSSs (cRSSs), in a reaction that requires cleavage of both the ESC SJ and a genomic cRSS[14–16]. Although the frequency of reintegration is not known, it can cause insertional

[1]School of Molecular and Cellular Biology, Faculty of Biological Sciences, University of Leeds, Leeds, UK. [2]Wolfson Childhood Cancer Centre, Newcastle University, Newcastle upon Tyne, UK. [3]Leukaemia Research Cytogenomics Group, Translation and Clinical Research Institute, Newcastle University, Newcastle upon Tyne, UK. [4]Haematological Malignancy Diagnostic Service (HMDS), St James's University Hospital, Leeds, UK. [5]Centre for Evolution and Cancer, The Institute of Cancer Research, London, UK. [6]Leeds Institute of Data Analytics, University of Leeds, Leeds, UK. [7]School of Molecular and Cellular Biology, Astbury Centre for Structural Molecular Biology, Faculty of Biological Sciences, University of Leeds, Leeds, UK. [8]Present address: Pathology Sciences, Southmead Hospital, Bristol, UK. [9]Present address: State Key Laboratory of Experimental Hematology, National Clinical Research Center for Blood Diseases, Haihe Laboratory of Cell Ecosystem, Institute of Hematology and Blood Diseases Hospital, Chinese Academy of Medical Sciences and Peking Union Medical College, Tianjin, China. [10]Present address: School of Molecular Biosciences, College of Medical, Veterinary and Life Sciences, University of Glasgow, Glasgow, UK. [11]These authors contributed equally: James N. F. Scott, Matthew P. Edwards. ✉e-mail: j.boyes2@leeds.ac.uk

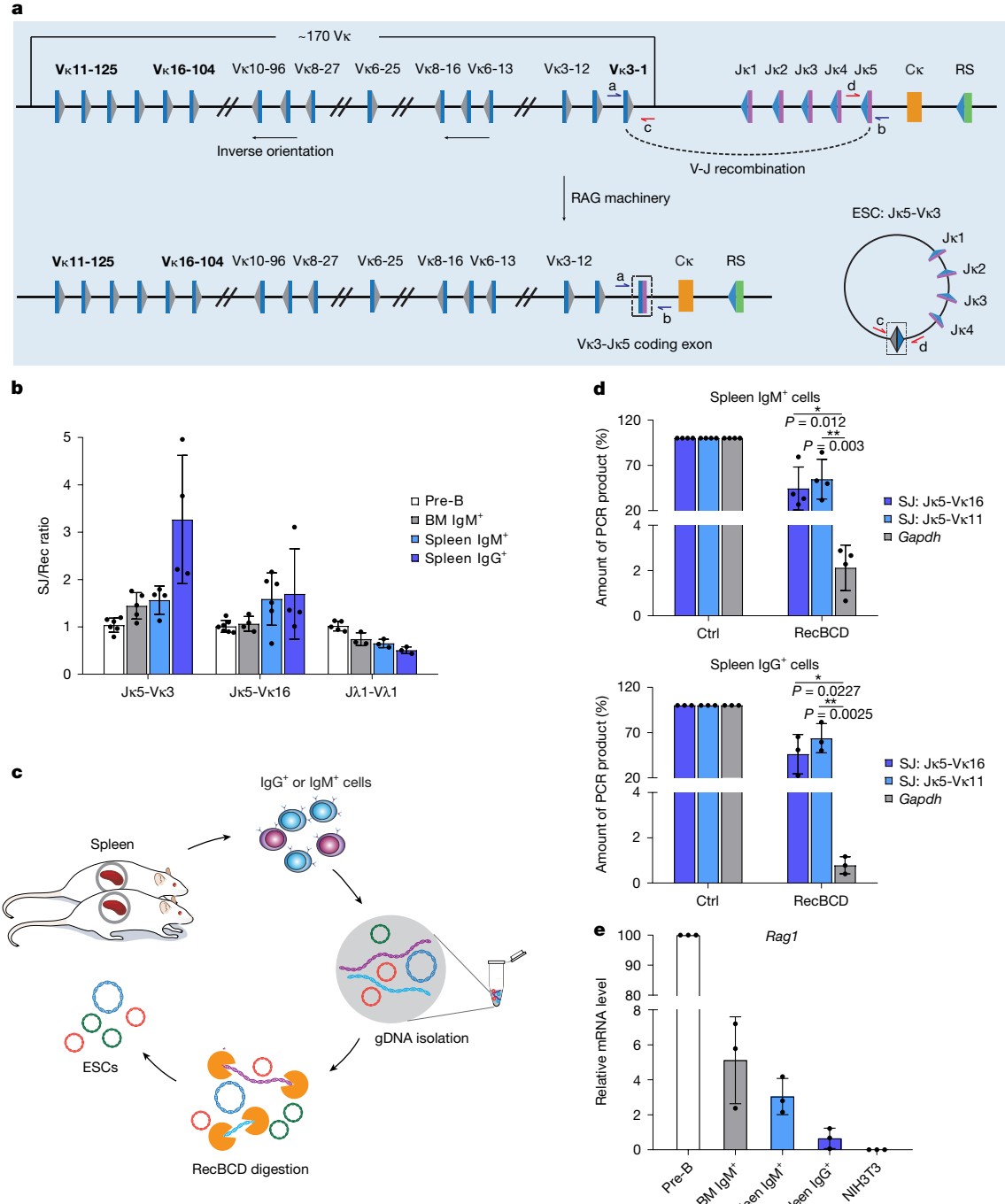

**Fig. 1 | RAG-generated extrachromosomal circular DNAs persist throughout mouse B cell development. a**, Schematic of the mouse immunoglobulin kappa (*Igk*) locus, highlighting the gene segments studied: Vκ16-104, Vκ3-1 and Vκ11-125, which undergo deletional recombination to generate an ESC. All *Igl* recombination reactions are deletional. Grey and blue triangles represent 12- and 23-RSSs, respectively; blue and purple rectangles represent V and J gene segments, respectively; the orange rectangle represents the constant region exon; the RS element is depicted by a green rectangle. **b**, Ratio of SJs to recombination junctions (Rec) at sequential stages of B cell development as determined by absolute qPCR for Jκ5-Vκ3-1, Jκ5-Vκ16-104 and Jλ1-Vλ1 and compared with the ratio in pre-B cells, which was set at 1:1 for ease of comparison. *n* = 3 samples. BM, bone marrow. **c**, Schematic of the isolation of ESCs from mouse mature B cells. Splenocytes were isolated from six-week-old mice and purified by flow cytometry to obtain IgM+ and IgG+ B cells (Supplementary

Fig. 1); genomic DNA (gDNA) was extracted and digested with RecBCD to remove linear DNA. **d**, Jκ5-Vκ11-125 and Jκ5-Vκ16-104 ESCs persist as circles in mouse mature B cells (IgM+, upper and IgG+, lower). The amount of undigested DNA was determined by qPCR. *Gapdh* (encoded by linear, genomic DNA) was used as a negative control; untreated DNA was a further control (Ctrl). *P* values were determined by an unpaired, two-tailed Student's t-test. Comparison versus *Gapdh*, 95% confidence interval: IgM+ Jκ5-Vκ11-125, −71.36 to −13.20; IgM+ Jκ5-Vκ16-104, −79.49 to −25.81; IgG+ Jκ5-Vκ11-125, −80.44 to −10.43; IgG+ Jκ5-Vκ16-104, −88.83 to −37.15. *n* = 3 samples. **e**, *Rag1* expression levels at different stages of mouse B cell development as determined by quantitative PCR with reverse transcription (RT–qPCR). Data are normalized to *Hprt* expression levels. NIH3T3 cDNA is a negative control. *n* = 3 samples. Mean values are shown; error bars represent s.d.

mutagenesis at tumour suppressor gene hotspots in T cell acute lymphoblastic leukaemia (T-ALL)[17]. Second, the RAG–ESC complex triggers double strand DNA breaks (DSBs) at cRSSs via a 'cut-and-run' reaction. In this reaction, the RAG–ESC complex synapses with a genomic cRSS, but unlike in reintegration, only the cRSS is cut[18]. The cleaved cRSS is released as a DSB, whereas the RAG–ESC complex remains intact to potentially trigger further DSBs[18]. Crucially, the DSBs caused by the cut-and-run reaction colocalize with many structural variants (SVs) found in ETV6–RUNX1 acute lymphoblastic leukaemia (ALL)[19] and map to frequently mutated genes in B cell precursor ALLs (BCP-ALLs)[18], implying that cut-and-run contributes to the development of BCP-ALL.

Current dogma states that ESCs are diluted during cell division and gradually decrease in number to negligible levels[20,21]. This, together with the downregulation of RAG proteins following productive immunoglobulin light chain (IgL)[22] or TCRα/γ recombination was believed to limit the potential harmful effects of the RAG–ESC complex. However, we show here that ESCs replicate and persist through multiple cell divisions. Moreover, similar to ecDNAs, high ESC copy numbers correlate with poor cancer prognosis, indicating that ESCs also have a pivotal role in cancer progression.

## ESCs persist to mature B cells in mice

ESCs, in the form of TCR excision circles, are present in naive thymic emigrants and persist in circulating lymphocytes for approximately two weeks in chickens and potentially much longer in primates[23–25]. Similarly, kappa-deleting recombination excision circles, generated by recombination between the Jκ–Cκ intron RSS and kappa-deleting element (KDE in humans, RS in mice), are found in about 30% of Igκ$^+$ B cells and nearly all newly generated Igλ$^+$ B cells[4]. RAG gene expression is downregulated following productive TCRα/γ and IgL rearrangement and editing[22], and therefore ESC production outside the thymus and bone marrow is negligible. Furthermore, ESCs are believed to be non-replicative[4] and lost via cell division. However, to test whether, like ecDNAs, ESCs are replicated and inherited, we purified genomic DNA from pre-B cells and IgM$^+$ B cells from mouse bone marrow, as well as IgM$^+$ and IgG$^+$ B cells from mouse spleen.

Several IgL recombination products were examined from both immunoglobulin kappa (Igk) and lambda (Igl) loci, together with their corresponding ESCs, via quantitative PCR (qPCR) using standards with known numbers of copies of the PCR product under investigation. The Igk locus undergoes both deletional recombination, where the SJ is found on the extrachromosomal ESC, and inversional recombination, where SJs are retained in the genome; our analyses focussed solely on deletional events (Fig. 1a). Remarkably, the SJ:recombination junction ratio shows only modest differences across all stages of B cell maturation and even appears to increase in IgG$^+$ cells for Jκ5-Vκ3-1 and Jκ5-Vκ16-104 ESCs (Fig. 1b and Extended Data Fig. 1a). This increase cannot be explained by removal of the recombination junction via secondary recombination, as the ratio of the recombination junction to other genomic regions is maintained throughout B cell development (Extended Data Fig. 1b). Given that at least six cell divisions are required for maturation from IgM$^+$ to IgG$^+$ cells[26], these data imply that SJs on ESCs are replicated and inherited. Indeed, in the absence of replication, ESCs would be diluted to 1.6% or less of their corresponding recombination junction in IgG$^+$ cells.

It remains possible, however, that ESCs have reintegrated into the genome and been replicated as part of genomic DNA. Reintegration cannot have occurred via opening of the SJ, as the assay to detect ESCs involves amplification of intact SJs. However, in principle, reintegration could occur via recombination between an RSS on the ESC (Fig. 1a) and one in the genome, resulting in the insertion of an intact SJ. The frequency of such events is expected to be low[15–17], but to address this possibility, we tested whether ESCs remain as extrachromosomal circles in IgM$^+$ and IgG$^+$ B cells.

High-molecular-mass genomic DNA was prepared from IgM$^+$ and IgG$^+$ B cells using conditions that minimize DNA shearing. Following exonuclease V (RecBCD) treatment to digest linear DNA but leave closed circular or nicked DNA intact[27] (Fig. 1c), we measured the amounts of residual Jκ5-Vκ11-125 and Jκ5-Vκ16-104 SJs by qPCR, using standards with known copy numbers. Remarkably, both SJs were present at more than 50% of the undigested level, whereas the Gapdh control from linear genomic DNA was nearly completely lost (Fig. 1d). The presence of significantly higher fractions of ESCs compared with Gapdh following treatment strongly indicates that ESCs are circular in mouse splenic IgM$^+$ and IgG$^+$ B cells. Moreover, very low Rag1 expression in IgG$^+$ cells implies that the circular ESCs have replicated and persisted from earlier stages, rather than being newly generated (Fig. 1e).

## ESCs are present in BCP-ALL samples

In principle, the presence of ESCs in mature lymphocytes is unlikely to have functional consequences, owing to their very low RAG gene expression. However, RAGs are aberrantly expressed in most BCP-ALL subtypes[28–30] (Extended Data Fig. 1c), and analysis of cDNA from more than 85 patient samples shows that expression of RAG1, but not RAG2, is significantly increased at diagnosis in patients who subsequently relapse (Extended Data Fig. 1d; RAG1: P = 0.0022). Therefore, the presence of RAGs and ESCs in these cancers could increase RAG–ESC complex formation and the risk of genome instability via cut-and-run or reintegration reactions. To investigate whether ESCs are detectable in primary BCP-ALL samples, we first determined the major recombination event(s) in nine ETV6–RUNX1 BCP-ALL samples using degenerate primer sets[31] to amplify rearrangements at the human IGK and IGL light chain loci. The resulting PCR products were cloned and sequenced, followed by sequence alignment to the IGK and IGL loci (Extended Data Fig. 2a).

The presence of the corresponding ESC SJs was then investigated using primers specific to each sample. Using this strategy, SJs with the predicted sequences were detected in six out of nine patients analysed (Extended Data Fig. 2b), with between one and three distinct SJs per patient. Since each recombination event generates just a single ESC and there are millions of cells in these malignancies, the ability to detect SJs in these samples suggests that these ESCs have replicated and persisted through many cell divisions.

Similar analyses (and high-throughput analyses, discussed below) using samples from patients with BCR–ABL1, CRLF2-r and low-hypodiploid BCP-ALL showed that SJs from deletional recombination events are also present in other BCP-ALL subtypes (Extended Data Fig. 2c).

To verify that SJs in BCP-ALL are present as extrachromosomal circles, high-molecular-mass DNA was prepared from patient samples using conditions that minimize DNA shearing. Following RecBCD treatment to remove linear DNA, SJs were amplified by droplet digital PCR (ddPCR). Consistent with their presence on extrachromosomal circles, SJs were amplified to a similar level with or without RecBCD treatment; by contrast, RecBCD reduced control linear DNA (GAPDH) to around 7% of untreated DNA levels (Extended Data Fig. 2d).

To better determine the fraction of patients with BCP-ALL carrying ESCs, we next capitalized on available whole-genome sequencing (WGS) data from patients with ETV6–RUNX1 BCP-ALL and re-analysed them for the presence of ESCs. DNA is prepared for WGS by shearing into fragments that average 500 bp, and approximately 35–50 bp at each end is sequenced. If the sequenced fragment includes an ESC SJ, the corresponding paired-end reads are flagged as 'discordant' when mapped to the genome, because the reads map much further away from each other than expected and appear to point towards each other, rather than away from each other. To detect ESCs, we therefore aligned WGS data from 61 patients with ETV6–RUNX1 BCP-ALL (European Genome-phenome Archive: EGAD00001000116) to the human

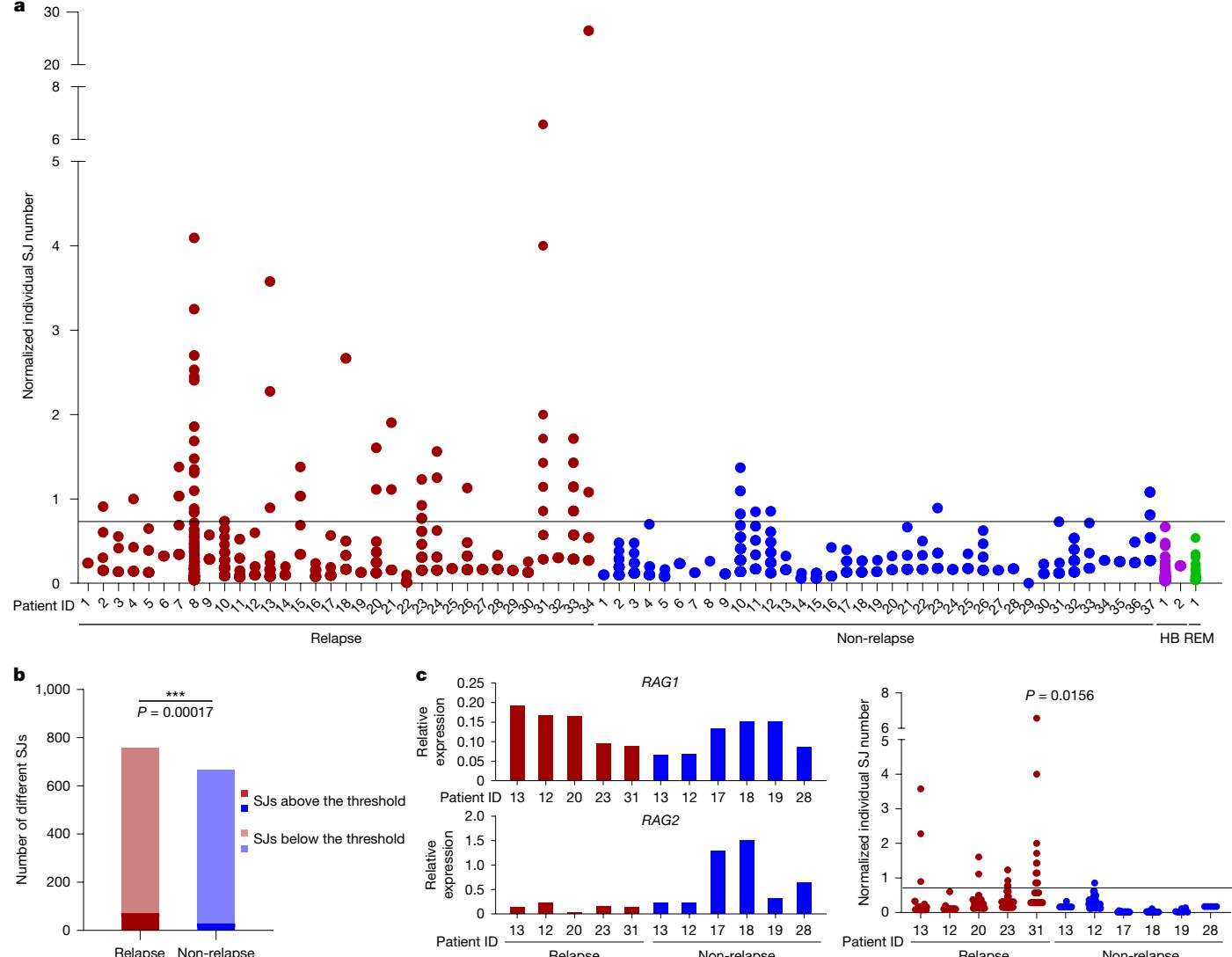

**Fig. 2 | High SJ copy numbers at diagnosis correlate with subsequent relapse. a**, Total *IGK/IGL* SJ levels were determined in samples from patients with BCP-ALL (from VIVO Biobank) at diagnosis using LAM-ESC. Normalized copy number of each SJ at diagnosis for patients who later relapsed (*n* = 34) and those who remained in remission (non-relapse; *n* = 37) are plotted. Normalized reads were calculated by dividing the reads obtained per SJ by the total LAM-ESC reads in that experiment. A threshold (horizontal line) was set at the highest normalized ESC level detected in healthy blood (HB, *n* = 2). SJ levels in a bone marrow sample taken at remission (REM) are shown for comparison. Only SJs resulting from deletional recombination events are shown. **b**, A significantly higher number of distinct SJs is present at levels above the threshold in patients who subsequently relapse compared with those who remain in remission. *P* value was determined by a two-tailed Fisher's exact test. *n* = 34 (relapse) and 37 (non-relapse). **c**, Left, expression of *RAG1* and *RAG2* mRNA at diagnosis in patients who later relapsed (*n* = 5) and those who did not (*n* = 6); comparable *RAG1* expression levels were observed in both groups. Right, normalized SJ copy numbers. The *P* value was determined by a two-tailed Fisher's exact test.

genome and filtered for reads that fit the above criteria. We detected ESCs in 51 out of 61 patients, with between 1 and 27 different ESCs per patient (Extended Data Fig. 2e and Supplementary Table 1). Owing to low sequencing depth and the fact that an ESC will be identified only if a split read is positioned exactly across the SJ, these data cannot quantify ESCs. Nevertheless, they fully support the presence of ESCs in the majority of patients with *ETV6–RUNX1* BCP-ALL and demonstrate that multiple ESCs can be present in each individual.

The coexistence of ESCs and RAGs in BCP-ALL could lead to increased RAG–ESC-mediated mutations and disease progression. However, the potential effect of the increased RAG–ESC activity depends on the quantity of ESCs and the timeframe over which ESCs are present. To investigate this, we established assays to detect all *IGK* and *IGL* recombination events and all *IGK* and *IGL* SJs. These assays, linear amplification-mediated (LAM)-recombination and LAM-ESC, are derived from LAM-high-throughput genome-wide translocation

sequencing (LAM-HTGTS)[32] and involve linear amplification with biotinylated primers against J regions across VJ coding junctions or ESC SJs (Extended Data Fig. 3a,b). The biotinylated products are then selected, adapters are ligated and the coding junctions or SJs are amplified by PCR. Following amplicon sequencing of the products, the resulting sequences are mapped to bespoke reference databases that include all possible *IGK/IGL* recombination events or *IGK/IGL* SJs[33] (https://github.com/Boyes-Lab/LAM-ESC-Recombination) but exclude intra-KV region recombination events[34] and their SJs.

Using the LAM-recombination and LAM-ESC assays, we analysed 71 samples, taken at diagnosis, from patients with BCP-ALL, 34 of whom subsequently relapsed. In addition to the primary recombination event, many secondary recombination events were present, indicating continued RAG activity[29] (Supplementary Table 2). Complementary LAM-ESC data show multiple ESCs per patient and that ESC copy numbers varied widely (Fig. 2a and Supplementary Table 3). Remarkably, comparison

of normalized LAM-ESC sequencing reads between patients who did and who did not subsequently relapse shows that significantly more ESCs were present at increased levels in patients who later relapsed (Fig. 2a,b; $P = 0.00017$). By contrast, in most patients who did not relapse, ESC copy numbers were close to those in healthy blood or bone marrow (Fig. 2a). Given that absolute ESC copy numbers, determined by ddPCR, correlated well with the number of normalized sequencing reads (Extended Data Fig. 4a) and that ddPCR experiments independently confirmed a similar fold increase of SJ levels in patients who later relapsed (Extended Data Fig. 4b), these data imply that there was increased ESC replication and/or persistence in patients who subsequently relapsed. No correlation was observed between *RAG1* or *RAG2* expression and ESC copy numbers (Extended Data Fig. 4c) and even in BCP-ALL samples in which *RAG1* expression was very similar, increased ESC copy numbers were observed in patients who later relapsed but not in those who did not (Fig. 2c; $P = 0.0156$). Differences in tumour infiltration also do not explain altered ESC levels, as leukaemic blast levels were similar (greater than 90%) in both patient groups (Supplementary Table 4).

## High ESC replication coupled to relapse

The increased ESC copy numbers in patients who later relapse implies that ESCs have replicated. Consistent with this, we identified seven SJs that were more abundant than their corresponding recombination junction using ddPCR (Extended Data Fig. 4d). Similarly, comparison of numbers of SJs and recombination junctions from KDE-KV2-30 and KDE-KV3-20 rearrangements across 48 samples shows that SJ numbers were significantly higher in patients who later relapsed (KDE-KV2-30: $P = 0.0461$, KDE-KV3-20: $P = 9.25 \times 10^{-5}$). By contrast, a much smaller increase (KDE-KV2-30: $P = 0.0756$, KDE-KV3-20: $P = 0.0173$) was observed in the corresponding recombination junctions (Extended Data Fig. 4e,f).

To investigate the extent of ESC replication, we capitalized on the fact that when each ESC is generated, a corresponding recombination 'footprint' is formed in the genome. If cells undergo multiple divisions following generation of an ESC, there will be multiple copies of the corresponding recombination junction. By contrast, if an ESC was generated recently, fewer copies of its recombination junction will be present (Fig. 3a). To measure ESC replication when the influence of ESC persistence is minimal, we focussed on recently generated ESCs. To this end, we examined ESCs corresponding to the lower limits of detectable LAM-recombination reads (≤0.2 normalized reads); as shown in Fig. 3b, the ratio of these SJs to the corresponding recombination junction was significantly higher in patients who later relapsed compared with those who did not ($P = 0.004$), implying increased ESC replication in patients prone to relapse. Notably, of the recently generated ESCs, 11 were identical in both patient groups (Extended Data Fig. 5a). After verifying that these ESCs were indeed recently generated by measuring the corresponding recombination junction by ddPCR, we quantified the SJ levels. Recently generated SJs were present at significantly higher levels in patients who later relapsed (Fig. 3c; $P = 0.001$). Given that the exact same SJ sequence was examined in the two patient groups, this increased replication cannot be due to the ESC sequence. Instead, it is likely that cell-intrinsic factors that are present at diagnosis trigger increased ESC replication in patients who later relapse.

To investigate what these cell-intrinsic factor(s) might be, we capitalized on available RNA sequencing (RNA-seq) data (https://www.cancer.gov/ccg/research/genome-sequencing/target, dbGaP sub-study ID: phs000464) obtained at diagnosis from 123 patients with BCP-ALL, 74 of whom later relapsed. Using DESeq2, followed by gene set enrichment analysis (GSEA), we found significantly increased expression of DNA repair genes ($P = 0.0095$; Fig. 3d) in patients who were prone to relapse. Published single-cell RNA-seq data from neuroblastoma cell lines similarly showed increased expression of the replication-associated genes *PCNA*, *POLE3* and *RPA2* corresponding to high ecDNA levels[35]. It is therefore plausible that replication and repair-associated gene products enhance ESC replication; consistent with this, significantly increased expression of *PCNA*, *RBX1*, *POLE3* and *POLE4* was observed in patients with high levels of SJs who later relapsed (Fig. 3e), but not in those with low levels of SJs, regardless of whether they relapsed (Fig. 3e). Furthermore, analysis of RNA-seq data (EGAS00001006863) from seven patients with known SJ levels (Fig. 2a) using DESeq2 and GSEA showed that expression of *PCNA*, *RBX1*, *POLE3* and *POLE4* was significantly increased in patients with high SJ levels compared with those with low SJ levels (Extended Data Fig. 5b) and that there was a significant increase in expression of DNA repair genes in patients with high SJ levels (Fig. 3f). These data therefore link higher expression of replication and repair-associated genes with increased ESC copy numbers, although a causal relationship with ESC replication has yet to be established. Nonetheless, previously identified replication initiation sites[36] are present within *IGK* and *IGL* ESC sequences (Extended Data Fig. 6), suggesting that ESCs may replicate via eukaryotic replication origins.

## ESCs persist through many cell divisions

The effects of an ESC are dependent not only on how many copies there are, but also on the timeframe during which it co-exists in cells with RAG proteins. Therefore, we investigated the extent of ESC persistence by measuring how many ESCs correspond to recombination events that took place many cell divisions ago—that is, where the corresponding recombination junction was present at a high copy number (Fig. 3a). These were categorized as 'major' recombination events and were distinguished from other events by the maximal change in the gradient on plots of the distribution of sequencing reads[37] (Extended Data Fig. 7a). Although we detected many ESCs corresponding to such major recombination events, implying that ESCs persist, there was no significant difference in the percentage of major recombination events with a corresponding SJ (or SJs) between patients who later relapsed and those who did not (relapse: 25.85%, non-relapse: 28.3%). However, SJs that corresponded to major recombination events were present with higher copy numbers in patients who later relapsed (Fig. 4a; $P = 0.011$). Moreover, when only SJs with increased copy numbers (above the threshold in Fig. 2a) were considered, almost half corresponded to major recombination events (Fig. 4b, left). This is consistent with persistence of high-copy ESCs through multiple cell divisions, increasing the risk that they will trigger mutations.

In control experiments, we considered the possibility that rather than persisting, identical ESCs could have been generated by more recent, secondary recombination events[38,39]. However, by capitalizing on the presence of ESCs that correspond to the primary recombination event (that is, those with the most sequencing reads) in three patients and analysing the clonotypes of the respective recombination junctions, our data indicate that it is very likely that the ESCs persisted from the primary recombination event in at least two cases (Extended Data Fig. 7b–g).

## ESC distribution in BCP-ALL

BCP-ALL progression is also likely to be influenced by the number of ESCs per cell and their intratumoral heterogeneity. To explore this possibility, we performed DNA fluorescence in situ hybridization (FISH) using probes that detect all ESCs from *IGL* and *IGK* loci. Actively cycling cells were blocked in mitosis and the presence of non-chromosomal DNAs was confirmed by DAPI staining (Fig. 4c). Since ecDNAs were not detectable in primary BCP-ALL samples by AmpliconArchitect[40] in this and a separate study[41] (20 and 44 patients[41], respectively), these non-chromosomal DNAs are likely to be ESCs. To further test

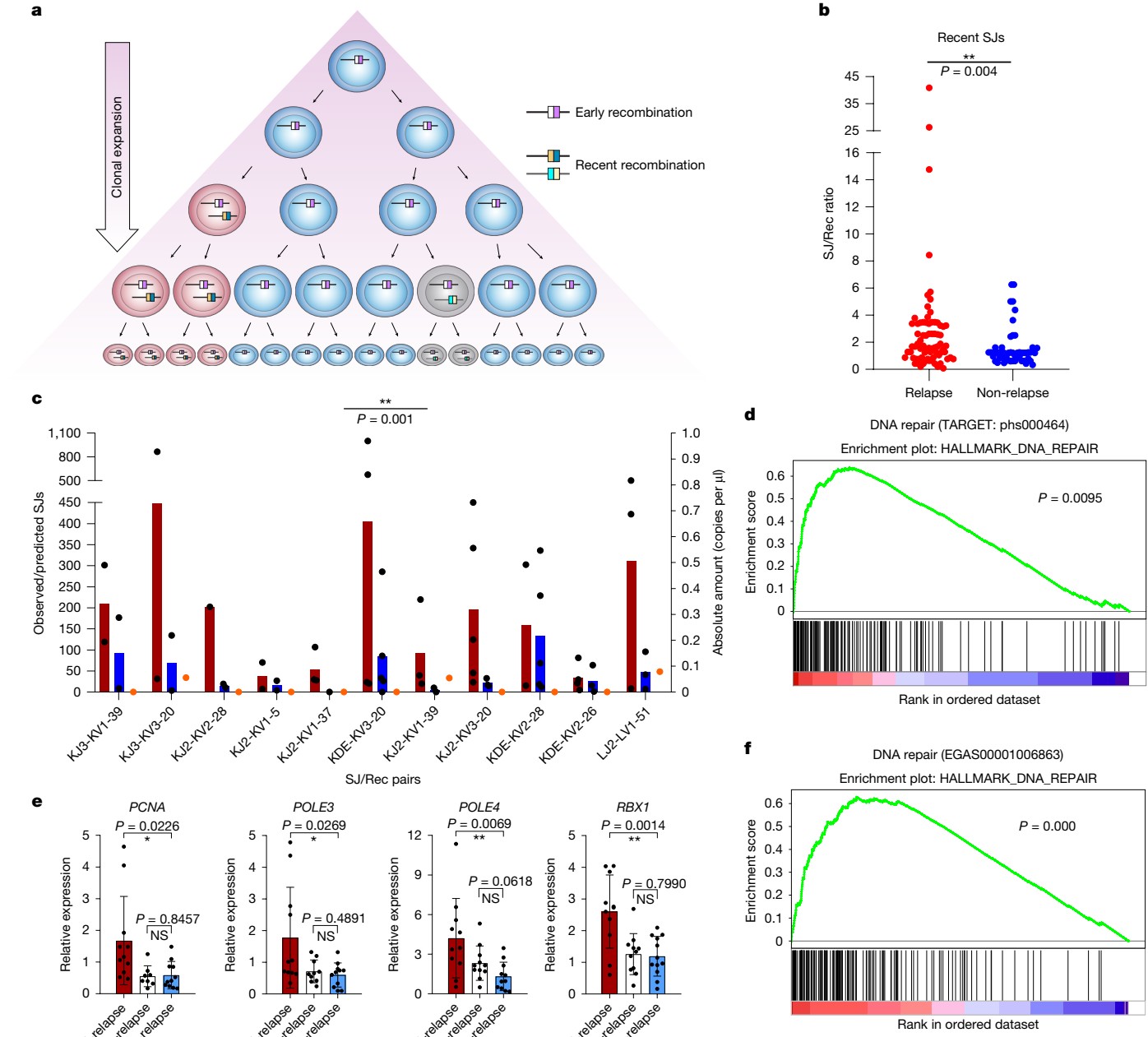

**Fig. 3 | Increased ESC replication in patients with BCP-ALL who later relapse.** **a**, Schematic highlighting that multiple copies of a recombination junction (white and pink) are present if recombination took place many cell divisions ago. **b**, SJs corresponding to very recent recombination events (≤0.2 normalized LAM-recombination reads) were identified. Normalized SJ reads were divided by corresponding normalized recombination reads. *P* value for the increase in patients who later relapsed determined by two-tailed Mann–Whitney *U* test; $n = 74$ SJs from 12 patients (relapse) and 51 SJs from 8 patients (non-relapse). **c**, Eleven SJs corresponding to ≤0.2 normalized recombination reads are in both patient groups. SJ levels were measured by ddPCR for patients who later relapsed (red) or remained in remission (blue). Data are mean observed/predicted SJ values, assuming a twofold SJ dilution at each cell division (Methods). Black dots indicate values for individual patients. *P* values determined using a

two-tailed Wilcoxon signed-rank test. $n = 11$ pairs. SJ levels only shown for HeLa controls (orange dots; $n = 2$). **d**, Enrichment plot for DNA repair genes from GSEA of RNA-seq data from 123 patients at diagnosis, 74 of whom relapsed. Significance determined as described[53,54] with corrections for multiple comparisons. **e**, RT–qPCR analysis of expression of genes associated with DNA replication and repair in patients who later relapse (for patients with high (H-) or low (L-) SJ levels) or remain in remission (non-relapse). $n = 11$ for each group except L-relapse: $n = 8$ (*PCNA*) and 10 (*POLE3*); and non-relapse: $n = 12$ (*RBX1*). Data are normalized to *HPRT* expression. *P* values determined using an unpaired, two-tailed Student's *t*-test. The 95% confidence interval is presented in Methods. Data are mean ± s.d. **f**, GSEA of RNA-seq data (EGAS00001006863) from patients with known SJ levels. Enrichment plot for DNA repair genes for patients with high versus low SJ levels ($n = 4$ and $n = 3$, respectively). *P* values determined as in **d**.

this and more accurately determine ESC distribution, we screened interphase nuclei for ESC FISH signals that were distinct from control (chromosomal) signals. For a patient with ESC levels similar to those in patients who later relapsed (patient P; Extended Data Fig. 8a), ESCs were observed in around 50% of cells, with a noticeable clustering of 3–7

ESCs in some cells (Fig. 4d–f and Extended Data Fig. 8b). By contrast, for patients with ESC levels similar to those in patients who remained in remission (patients M and N; Extended Data Fig. 8a), ESCs were detected in fewer cells, with only one or occasionally two ESCs per cell (Fig. 4d). Notably, these interphase FISH signals showed high correspondence

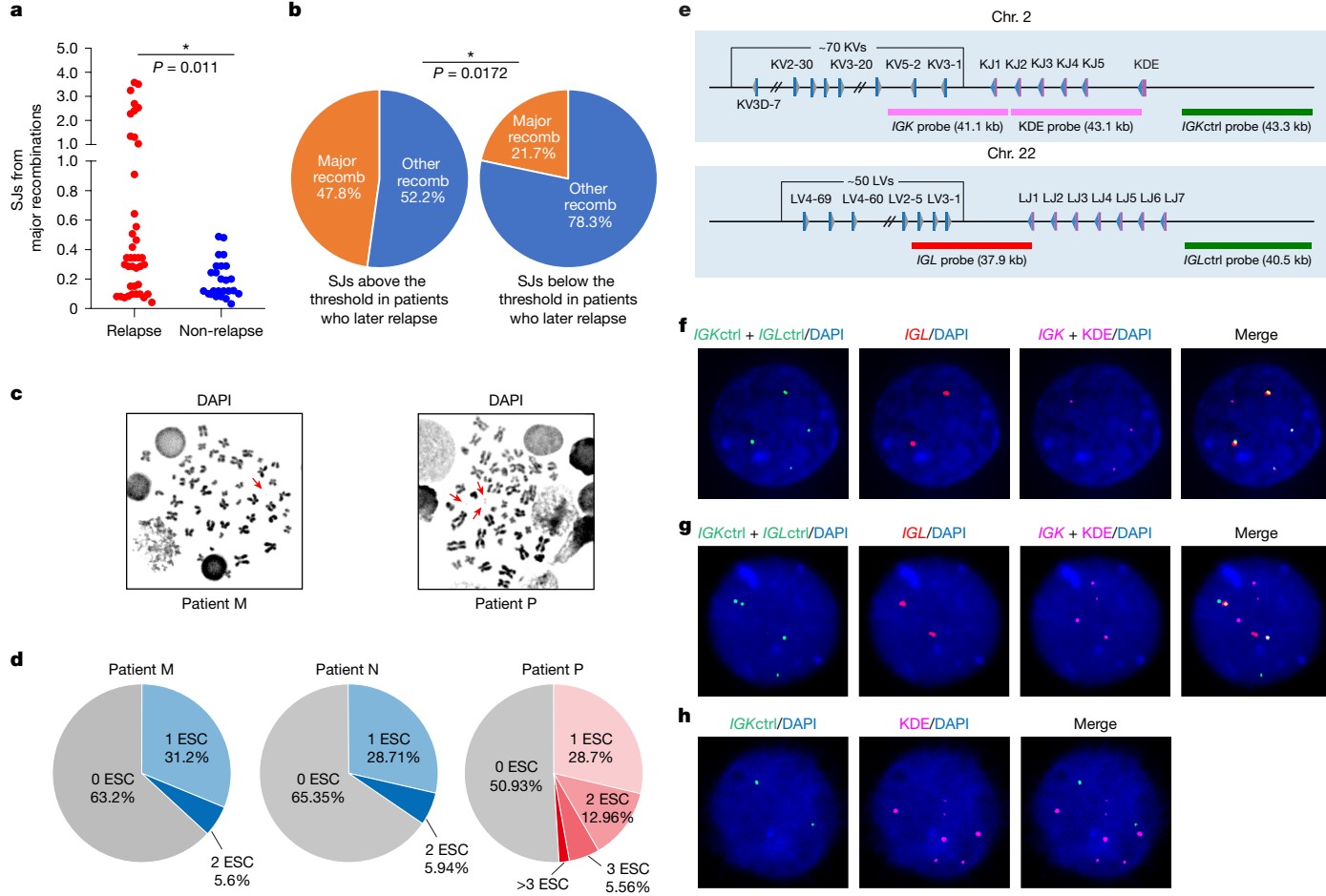

**Fig. 4 | ESCs persist at higher copies in patients who subsequently relapse. a**, Major recombination signifies multiple cell divisions since the recombination event (Fig. 3a and Extended Data Fig. 7a). Normalized sequencing reads for SJs corresponding to major recombination events are plotted for each patient group. *P* value for the increase in patients who later relapse versus those who do not determined by a two-tailed Mann–Whitney *U* test; *n* = 40 from 23 patients (relapse) and *n* = 25 from 20 patients (non-relapse). **b**, Pie chart showing percentage of SJs with normalized sequencing reads above or below the threshold in healthy blood (Fig. 2a) that correspond to major recombination events versus other recombination events. *P* value for the difference determined by a two-tailed Fisher's exact test. *n* = 23 high-copy SJs from 7 patients and 124 low-copy SJs from 12 patients. Recomb, recombination events. **c**, Representative DAPI-stained metaphase chromosome spreads from BCP-ALL samples with

ESC levels similar to patients who remain in remission (left; *n* = 29 images) or who later relapse (right; *n* = 26 images). Red arrows indicate non-chromosomal DNAs. Images deconvoluted using CellSens Dimension software. **d**, ESCs detected by interphase FISH in patients with ESC levels similar to those who relapse (patient P, *n* = 108 images) or remain in remission (patient M, *n* = 101 images; patient N, *n* = 127 images). **e**, Schematic showing ESC and control FISH probes to *IGK* and *IGL* loci. Blue and grey triangles represent 12- or 23-RSSs. **f**, Representative interphase FISH image. Control (ctrl) probes against *IGK* and *IGL* are in green; the regions excised to generate ESCs are in red (*IGL*) or magenta (*IGK*). *n* = 99. **g**, *IGK* (magenta) and *IGL* (red) ESCs in the same cell. *n* = 10. **h**, Multiple ESCs detected per cell using *IGK* control probe (green) and KDE probe (magenta). *n* = 8. All images obtained at 60× magnification.

with the relative levels of DAPI-stained non-chromosomal DNAs (Extended Data Fig. 8c), and, when present at increased levels, ESCs exhibited remarkable intratumoral heterogeneity (Fig. 4d). We also detected the presence of *IGK* and *IGL* ESCs in the same cell (Fig. 4g). Given that simultaneous recombination of both loci is highly unlikely, the coexistence of three *IGK* ESCs and one *IGL* ESC supports the idea that ESCs replicate and persist.

Similarly, seven ESCs resulting from recombination to KDE were found within a single cell (Fig. 4h). There are only two KDE RSSs; therefore, the maximum number of ESCs that could be generated by recombination to KDE is two. Thus, the presence of seven ESCs in one cell was a strong indication of ESC replication. To verify this, we prepared metaphase chromosome spreads from BCP-ALL cells that were cultured in the presence of bromodeoxyuridine (BrdU) and then blocked in mitosis. Following hybridization with BrdU antibodies, we observed signals coinciding with DAPI-stained non-chromosomal DNAs, confirming ESC replication (Extended Data Fig. 8d).

## ESC copy number and BCP-ALL progression

The presence of high ESC copy numbers over multiple generations increases the cumulative risk of DNA damage and may lead to cells with high ESC copy numbers undergoing clonal expansion. Consistent with this, SJs with high copy numbers were associated with a larger fraction of cells that had undergone many divisions (major recombination events; Fig. 4b, left) than those with low copy numbers (below threshold) (Fig. 4b, right). Next, we aimed to determine how ESC copy number at diagnosis influenced BCP-ALL progression. In other cancers, ecDNAs lead to a worse prognosis by increasing oncogene copy numbers[1,42], acting as mobile enhancers[43], or by ecDNA integration into a tumour suppressor gene[44]. There are no known oncogenes on ESCs from the *IGK–IGL* or *TCRA–TCRG* loci and no known enhancers in the excised V-J regions of human loci. Similarly, although recombination to KDE generates ESCs that incorporate two strong enhancers—iEκ and 3′Eκ[45,46]—and such ESCs are significantly enriched and found in

patients who later relapse (Extended Data Fig. 8e–g), patients who lack KV-KDE ESCs nonetheless relapse (Supplementary Table 3). Therefore, although ESCs may promote malignancy by acting as mobile enhancers, ESCs are likely to influence disease progression by other mechanisms.

ESCs cause increased mutations when complexed with RAG proteins, triggering either ESC reintegration at cRSSs or the cut-and-run reaction that generates DSBs at cRSSs. Both reactions produce SVs that have a cRSS on only one side of the breakpoint[14,18]. RAGs also trigger genome alterations that are independent of ESCs via off-target recombination between two cRSSs, leading to insertion–deletion mutations or chromosome translocations with cRSSs on both sides of the breakpoint[19,47]. To test whether the RAG–ESC complex contributes significantly to the mutations found in BCP-ALL, we examined the SVs in WGS data of patients with BCP-ALL using Therapeutically Applicable Research to Generate Effective Treatments (TARGET) data (https://www.cancer.gov/ccg/research/genome-sequencing/target; dbGaP sub-study ID: phs000464). Using 150 patient sequences obtained at diagnosis across all BCP-ALL subtypes, we first calculated the fraction of SVs with one or two cRSSs at the breakpoint. Consistent with previous data[19], cRSSs were present at nearly 40% of breakpoints (38.4%); of these, nearly 62% had a single cRSS and 36.4% had two cRSSs (Extended Data Fig. 9a). To determine the probable cause of the breaks at single cRSSs, we used an in-house script[48] (https://github.com/Boyes-Lab/NGS-Analysis), which showed that cut-and-run[18] occurs more than 60-fold more frequently than either reintegration[14–16] or RAG-mediated insertions[49] (Extended Data Fig. 9a).

Next, we compared the numbers of SVs in patients, and found a significant increase in SVs with cRSSs on only one side of the breakpoint in patients who subsequently relapsed compared with those who did not (Fig. 5a, left). Moreover, consistent with the above analysis, the frequency of SVs with a single cRSS at the breakpoint (Fig. 5a, left) was much higher than those with cRSSs on both sides of the breakpoint (Fig. 5a, right; mean relapse: 27.19 versus 8.27, mean non-relapse: 23.48 versus 4.635), suggesting a greater role for RAG–ESC-mediated mutations.

Increased ESCs in patients at diagnosis imply that the RAG–ESC complex continues to cause damage between diagnosis and relapse. We therefore compared WGS data from 83 matched samples taken at diagnosis and relapse. By focusing on SVs near genes that are frequently mutated at relapse[50,51], we observed significantly more SVs with cRSSs on one side of the breakpoint that are specifically present at relapse compared with other SVs (Fig. 5b). Likewise, re-analysis of available LAM-HTGTS data[18] (NCBI SRA: PRJNA483469) showed a significant increase in targeting of relapse-associated genes by the RAG–ESC complex compared with the RAG–RSS complex in cells derived from a patient with relapsed *ETV6–RUNX1* BCP-ALL (Fig. 5c). Together, these data imply that ongoing activity of the RAG–ESC complex triggers mutations at genes that are associated with relapse.

To test whether there is a direct link between increased ESC levels and SVs at single cRSSs, we analysed WGS (EGAS00001006863) and whole-exome sequencing (WES) data of patients with BCP-ALL for whom we had measured SJ and RAG gene expression levels (Fig. 2a and Extended Data Fig. 1d). Consistent with the idea that the RAG–ESC complex triggers relapse-associated mutations, we observed a significant increase in SVs with a single cRSS at the breakpoint at relapse-associated genes in patients with high SJs plus high *RAG1* expression compared with those with high *RAG1* expression plus low SJs or low *RAG1* expression plus low SJs (Extended Data Fig. 9b–d). Moreover, in a patient with high SJs, we observed clonal expansion of an SV at a single cRSS in the spleen tyrosine kinase (*SYK*) gene between diagnosis and relapse (Extended Data Fig. 9e).

If reactions involving the ESC truly underpin the mutations that lead to relapse, the presence of sufficient copies of ESCs at or before diagnosis would be expected to lead to expansion of the ESC-harbouring cells by relapse. To test this, we capitalized on the ability to trace cells in which ESCs have been generated by virtue of their corresponding recombination junction. We therefore performed LAM-recombination on samples taken at relapse, in cases where patients have high ESC copies at diagnosis. The frequencies of some recombination junctions stayed the same or even decreased between diagnosis and relapse, possibly owing to treatment-mediated loss of the corresponding cells (Supplementary Table 2). However, others showed marked increases in normalized sequencing reads (Extended Data Fig. 10a), an observation that was confirmed for ten cases by ddPCR (Fig. 5d).

We next tested whether clonal expansion, measured by the increase in recombination junctions, was linked to the presence of a matching ESC at diagnosis. Remarkably, a corresponding ESC was detected in every case of clonal expansion, but in only 58% of cases where no expansion of recombination junctions was observed, and in these latter cases, ESC copy numbers were often lower (Extended Data Fig. 10b). This correlation is notable and consistent with the idea that ESC-mediated mutagenesis promotes disease progression to relapse.

A further prediction is that if ESCs indeed replicate and persist, the same ESCs should be detectable at both diagnosis and relapse. Remarkably, we found that all ESCs persisted (Fig. 5e), in one case for more than seven years and in two cases for more than four years. Quantification of SJs at diagnosis and at relapse showed that some increased, whereas others decreased slightly, supporting the idea that the SJ is on an extrachromosomal circle rather than integrated into the genome.

Finally, if ESC activity causes disease progression, high SJ copy numbers at diagnosis should predict disease outcome. We therefore re-analysed the data showing SJ levels at diagnosis (Fig. 2) according to BCP-ALL subtype (Supplementary Table 4). Although genomic integration of SJs in some *DUX4*-r samples (Extended Data Fig. 10c,d) precluded such analyses for this subtype, there was a good correlation between the SJ copy number above the threshold and subsequent relapse for subtypes that usually have a good prognosis (*ETV6–RUNX1* and high hyperploid (HeH) BCP-ALL) or an intermediate prognosis (*TCF3::PBX* BCP-ALL) (Fig. 5f and Extended Data Fig. 10e). Moreover, in BCP-ALLs where *PAX5* was altered or mutated, SJ levels were higher overall, but were noticeably different between patients who relapsed and those who did not, suggesting that a higher threshold may better identify patients who are at risk of relapse (Extended Data Fig. 10e). Our data therefore imply that the presence of ESCs at diagnosis above subtype-specific threshold levels is frequently associated with subsequent relapse. This strongly suggests a central role for ESCs in disease progression.

## Discussion

ESCs were long believed to be inert and diluted during cell division. However, it is now clear that ESCs have biological activity and replicate and persist in healthy lymphocytes and in BCP-ALL. In BCP-ALL, persistence of increased copy numbers of ESCs is strongly linked with worse disease outcomes, thereby resembling ecDNAs. ESCs and ecDNAs are both extrachromosomal circles; *IGK/IGL* ESCs are a similar size to ecDNAs; and ESCs and ecDNAs replicate and persist through multiple cell divisions and both confer a growth advantage to cancer cells when present at elevated levels[5,6]. However, there are key differences. Whereas ecDNAs often confer a growth advantage by increasing oncogene copy numbers[10], ESCs trigger mutations, including at cancer driver genes and relapse-associated genes, via the cut-and-run reaction[18]. These mutations accumulate through time and are inherited by all daughter cells, regardless of whether those cells inherit the ESC. Similarly, once the RAG–ESC complex has triggered sufficient mutations in key genes, the continued presence of the ESC may not be required. By contrast, continued oncogene amplification via ecDNAs is required for a growth advantage. This may explain why ecDNA levels gradually increase as cancers progress[12], whereas ESC levels are lower overall and may increase or decrease between diagnosis and relapse. Moreover, the stochastic nature by which the recombinase, and thus

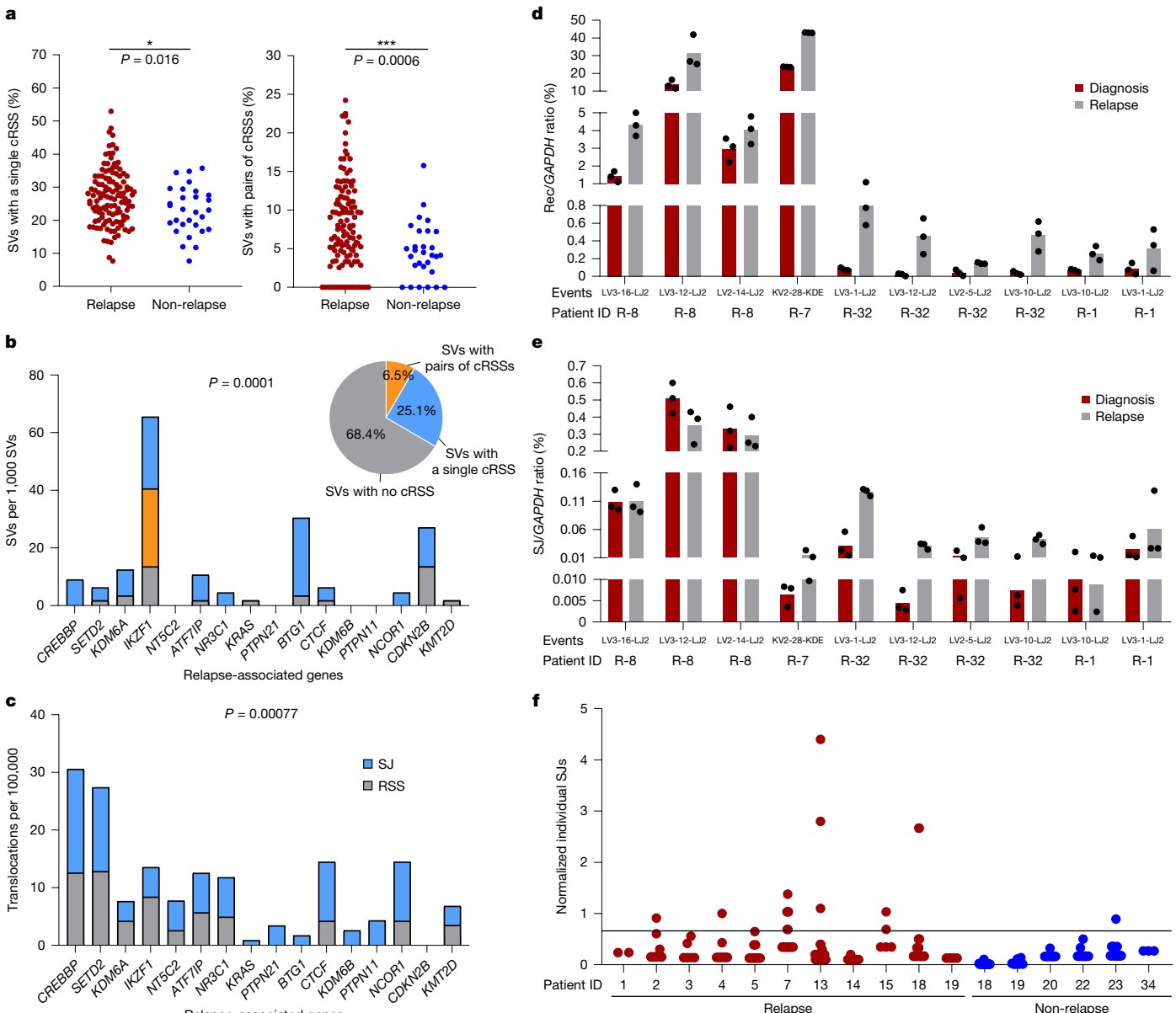

**Fig. 5 | Association between increased ESC-mediated mutations and relapse. a**, SVs per patient at diagnosis with a single cRSS (left) or two cRSSs (right) at the breakpoints plotted for patients who later relapsed (121 patients) versus those who remained in remission (29 patients). *P* value for the difference determined by a two-tailed Mann–Whitney *U* test. Single cRSS: *P* = 0.016; two cRSSs: *P* = 0.0006. SVs are from the TARGET study. Breakpoint junctions within antigen receptor loci are omitted. **b**, Right, pie chart showing all SVs at relapse with one cRSS at the breakpoint, two cRSSs or no cRSS for 83 matched patient samples; SVs at diagnosis were subtracted from those at relapse. Left, bar chart showing SVs at genes commonly mutated at relapse. *P* value for increased SVs involving single cRSSs at relapse-associated genes compared with all SVs involving a single cRSS calculated using a one-sided hypergeometric distribution; *P* = 0.0001. **c**, Frequency of breaks in 10 genes that most often acquire somatic

mutations in relapsed BCP-ALL[51] in the presence of the ESC (SJ) versus 12-RSS and 23-RSS controls (RSS) in a LAM-HTGTS experiment[18]. *P* value for co-localization of breaks detected by LAM-HTGTS and frequently mutated genes in relapsed ALL versus all genes mutated in ALL calculated using a one-sided hypergeometric distribution; *P* = 0.00077. *n* = 4. **d**, ddPCR of selected recombination junctions at diagnosis and relapse. Cell numbers normalized using *GAPDH*. Bars show mean values. *n* = 10 recombination junctions from four patients. **e**, ddPCR analysis of selected SJs at diagnosis and relapse for the indicated patients. Black dots show technical repeats. Bars show mean values. *n* = 10 SJs from 4 patients. **f**, Copy numbers of individual SJs at diagnosis for patients with *ETV6–RUNX1* BCP-ALL (*n* = 16) who later relapsed or remained in remission. The horizontal line shows the highest normalized SJ copy number detected in healthy blood.

the RAG–ESC complex, finds its targets means that different mutations will be triggered in each cell. This may explain why some cells with high ESC copy numbers expand between diagnosis and relapse whereas others, possibly those in which fewer cancer driver genes have been mutated, either respond better to treatment or become diluted. Although high ESC copy numbers correlate with subsequent relapse in around 50% of cases, some patients with low ESC levels at diagnosis nonetheless go on to relapse. In these cases, other mutations,

such as deletion of *IKZF1*[52] or initiating mutations that alter signalling to dysregulate multiple pathways[50], may predispose the patient to relapse. Nonetheless, the good correlation between high ESC levels and poor prognosis, particularly for *ETV6–RUNX1* and HeH BCP-ALL, may be of clinical utility. Collectively, our data demonstrate that ESCs are replicated and inherited and form a complex with RAG proteins that can lead to relapse-associated mutations and clonal expansion (Extended Data Fig. 11).

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

## Methods

### Purification of mouse B cells

Non-transgenic CBA/C57BL/6 J mice were obtained from the University of Leeds animal facility, which is a full barrier facility, with a light/dark cycle of 12 h on/12 h off, an ambient temperature of 21 °C (range 20–22 °C) and 45–65% humidity. No more than six animals are housed per cage and all mice are free of common pathogens, including murine norovirus, *Pasteurella* and *Helicobacter*. Animal procedures were performed under Home Office licence P3ED6C7F8, following review by the University of Leeds ethics committee. Mouse femurs and spleens were collected from 5- to 7-week-old mice, using 12 mice in total to minimize animal numbers used, according to 3Rs principles. Roughly equal numbers of male and female mice were used; randomization and blinding were not necessary, as all mice were wild type. Bone marrow cells were flushed from femurs with PBS whereas splenocytes were prepared by flushing cells from finely diced pieces of spleen with PBS through a 50-µm cell strainer. Following preparation of single-cell suspensions in PBS, bone marrow cells and splenocytes were centrifuged at 600$g$ for 3 min and resuspended in 10 ml of 0.168 M $NH_4Cl$ to lyse erythrocytes. After 10 min, cells were washed with 40 ml PBS and resuspended in 1 ml staining buffer (2% FCS, 1 mM EDTA, 25 mM HEPES-KOH pH 7.9 in PBS).

Cells were stained with the appropriate antibodies prior to purification by flow cytometry. For bone marrow pre-B cells, cell suspensions were stained with 6 µl each (6:1,000 dilution) of FITC anti-CD19 (BD Pharmingen, 553785) and PE anti-CD43 (BD Pharmingen, 553271). Bone marrow or spleen IgM$^+$ cells were stained with 10 µl (1:100 dilution) FITC anti-IgM (BD Pharmingen, 553408) whereas spleen IgG$^+$ cells were stained with 15 µl PE (3:200 dilution) anti-IgG (eBioscience, 12-4010-82). Following incubation at room temperature for 10 min, cells were washed with PBS and resuspended in 0.5 ml staining buffer prior to purification using a FACSMelody (BD) cell sorter running BDFACSChorus 3.0 software. CD19$^+$/CD43$^-$ cells were isolated as the pre-B population whereas cells stained with anti-IgM$^+$ or anti-IgG$^+$ were isolated as their respective populations.

### Patient samples

Patient samples, taken as part of routine diagnostics, were supplied by VIVO Biobank, HMDS or a hospital in the Czech Republic. Collection and use of patient samples were approved by the appropriate institutional review board (IRB). Each organization obtained informed patient consent for anonymized samples to be used by third parties for research. The use of surplus diagnostic material for research by HMDS and collaborators was approved by the Health Research Authority (HRA): 04_Q1205_125. Local ethics approval was obtained from the Biological Sciences Research Ethics Committee, University of Leeds: BIOSCI 18-031 & 2308 and CCR 2285, Royal Marsden Hospital NHS Foundation Trust.

### Cell culture

hTERT-RPE-1 cells were from ATCC where they were authenticated by morphology, STR profiling and karyotyping and verified mycoplasma-free. NIH3T3 cells were from the laboratory of C. Bonifer, whereas HeLa cells were from the laboratory of T. Enver. Both cell types were authenticated using species-specific PCR primers. They were verified free from mycoplasma using MycoAlert Mycoplasma Detection Kit (LT07-318). HeLa cell contamination has caused misidentification of other cell lines. However, HeLa cells were used here only to prepare human DNA and we verified that the DNA was human using human-specific PCR primers. hTERT-RPE-1, NIH3T3 and HeLa cells were maintained in Dulbecco's modified Eagle's medium (DMEM) supplemented with 10% fetal calf serum, 4 mM L-glutamine, 50 U ml$^{-1}$ penicillin and 50 µg ml$^{-1}$ streptomycin in a humidified incubator at 37 °C with 5% $CO_2$.

### Preparation of genomic DNA

Genomic DNA from mouse B cells, NIH3T3 cells, HeLa cells and patient samples was prepared by gently resuspending $1 \times 10^6$ to $5 \times 10^6$ cells in a 200 µl digestion buffer (200 mM NaCl, 10 mM Tris-HCl pH 7.5, 2 mM EDTA, 0.2% SDS). Proteinase K was added to a final concentration of 0.4 mg ml$^{-1}$, followed by incubation at 56 °C overnight, with rotation. The next day, an equal volume of isopropanol was added and the sample mixed thoroughly but gently by inversion to precipitate the DNA. The DNA pellet was recovered by centrifugation at 20,000$g$ for 5 min at room temperature and washed twice with 70% ethanol. DNA was then resuspended in 100 µl TE and incubated at 56 °C for at least 3 h to ensure complete resuspension; the concentration was measured using a DeNovix DS-11 spectrophotometer.

High-molecular-mass genomic DNA was prepared from fresh BCP-ALL bone marrow aspirates (that were surplus to diagnostic needs and that had been maintained at 4 °C), using a Promega Wizard HMW DNA extraction kit according to the manufacturer's instructions for whole blood. Samples were resuspended in TE at room temperature overnight prior to measuring the concentration as above.

### Isolation of total RNA and reverse transcription

Two million mouse B cells, NIH3T3 cells, HeLa cells or BCP-ALL cells were resuspended in 500 µl of TRIzol (Invitrogen, 3289) and total RNA was isolated according to the manufacturer's instructions. DNA contaminants were removed by treatment with 2 U DNase I (Thermo Scientific, EN0521) for 45 min at 37 °C in 100 µl of 1× DNase I buffer (10 mM Tris-HCl pH 7.5, 2.5 mM $MgCl_2$, 0.1 mM $CaCl_2$). Following phenol-chloroform extraction and ethanol precipitation, total RNA was resuspended in 20 µl of dd$H_2O$ and the concentration was determined using a DeNovix DS-11 spectrophotometer.

One microgram of total RNA was reverse transcribed with M-MuLV reverse transcriptase (Invitrogen, 28025-013). In brief, 1 µg of RNA was added to 2.5 µM oligo dT primer, 500 µM dNTPs and dd$H_2O$ to give a total volume of 12 µl. This was incubated at 65 °C for 5 min and immediately placed on ice before addition of 4 µl first strand buffer (Invitrogen), 10 mM DTT and 1 µl RNase inhibitor (PCRBIO, PB30.23-02). The reaction was incubated at 37 °C for 2 min, followed by addition of 1 µl M-MuLV and incubation at 37 °C for 50 min prior to heat inactivation at 70 °C for 15 min.

### Exonuclease V treatment of genomic DNA

Linear DNA was removed from genomic DNA using Exonuclease V (RecBCD, NEB M0345S). Reactions comprised 1 µg genomic DNA, 1× NEBuffer 4, 1 mM ATP and 10 U RecBCD in 100 µl total volume. Negative control reactions were identical except RecBCD was omitted. Following incubation at 37 °C for 1 h (mouse DNA) or 3 h (BCP-ALL DNA), EDTA was added to a final concentration of 11 mM and reactions were heat inactivated at 70 °C for 30 min. DNA was then ethanol precipitated and resuspended in 50 µl dd$H_2O$; 5 µl (100 ng) was used directly for PCR.

### Quantitative PCR

qPCR was performed using a Rotor-Gene 6000 cycler (Corbett) and analysed using the Corbett Rotor-Gene 6000 Series Software (v.1.7, build 87). All reactions were carried out in a final volume of 10 µl, containing 1× qPCRBIO SyGreen Mix (PCRBIO, PB20.14), 400 nM each primer and 100 ng genomic DNA, 1–5 ng cDNA or 1 µl first round PCR product (for nested PCR). All reactions were performed in duplicate. In each case, a standard curve of the amplicon was analysed concurrently to evaluate the amplification efficiency and to calculate the relative amount of amplicon in unknown samples. $R^2$ values were 1 ± 0.1. A typical cycle consisted of: 95 °C for 3 min, followed by 40 cycles of 95 °C for 5 s, $T_m$ for 10 s and extension at 72 °C for 10 s, where $T_m$ = melting temperature of the primers (Supplementary Table 5). A melt curve, to determine amplicon purity, was produced by analysis of fluorescence

as the temperature was increased from 72 °C to 95 °C. Amplicons were 100–200 bp.

Standard curves for absolute quantification were generated by 35 cycles of conventional PCR and purification of the desired product via a 1.2% agarose gel and a QIAquick gel extraction kit, (QIAGEN, 28704). Following measurement of the concentration via absorbance at 260 nm (using a DeNovix DS-11 spectrophotometer), DNA was diluted to 1–10 ng/µl before more accurate concentration determination using a QuantiFluor dsDNA kit (Promega, E2670) and a FLUOstar OPTIMA plate reader (BMG Labtech). An appropriate range of each standard was used in qPCR, ensuring that all unknown samples were within the standard curve.

Primers and melting temperatures are shown in Supplementary Table 5 for quantitative analysis of recombination, SJs and *Rag1* expression in mouse bone marrow and spleen, and for quantitative (qPCR) analysis of recombination, SJ levels, *RAG1* and *RAG2* expression as well as *PCNA*, *POLE3*, *POLE4* and *RBX1* expression in BCP-ALL patient samples. BLAST was used to check primer specificity. *HPRT* was used as a reference gene for expression studies, using primers that span an intron. Genomic *GAPDH* sequences were used to normalize for cell numbers. These housekeeping genes were chosen for their widespread expression (*HPRT*) and low likelihood of mutation.

### Detection of recombination in BCP-ALL patient samples

PCR was performed using Taq DNA polymerase (NEB, M0267) in reactions comprising 1× ThermoPol buffer, 200 µM dNTPs, 0.5 µM each primer, 1.25 U Taq DNA polymerase and 100 ng genomic DNA template in a final volume of 50 µl. Primers for recombination at the immunoglobulin kappa and lambda loci were as described by the BIOMED-2 consortium[31]. Cycling conditions involved initial denaturation at 95 °C for 5 min, followed by 95 °C for 30 s, 60 °C for 30 s and 68 °C for 30 s for 35 cycles, followed by a final extension of 5 min at 68 °C. PCR products were separated by gel electrophoresis; products of the expected sizes[31] were excised and cloned prior to Sanger sequencing.

### Nested PCR of recombination junctions and SJs in mouse and patient samples

To achieve sufficient specificity and sensitivity, nested PCR was performed using Taq DNA polymerase (NEB, M0267). First round reactions consisted of 1× ThermoPol buffer, 200 µM dNTPs, 0.5 µM each primer, 1.25 U Taq DNA polymerase and 20–100 ng genomic DNA template in a final volume of 50 µl. To detect mouse recombination and SJs, thermocycling conditions involved denaturation at 94 °C for 3 min, followed by 18 cycles of 94 °C for 20 s, $T_m$ for 20 s and 72 °C for 20 s. A 7 min of final extension was performed. One microlitre of 1:10 diluted first round PCR product was used as the template for a second round of qPCR using the primers shown in Supplementary Table 5.

As the BIOMED-2 primers to detect human SJs do not robustly identify specific J gene segments, nested PCRs were carried out with a mixture of first round primers that bind upstream of all five KJ RSSs or LJ1, LJ2 and LJ3 RSSs, which account for ~99% of *IGL* recombination events[31], followed by second round PCRs specific for each individual J RSS. First round reactions were set up as described above with 50–100 ng template DNA, followed by denaturation at 95 °C for 5 min, and 25 cycles of 95 °C for 30 s, $T_m$ for 30 s and 68 °C for 30 s, and a final extension of 5 min at 68 °C. Second round reactions were identical except 1 µl first round PCR product was used as template and the number of cycles was optimized for each amplicon, which was typically 36 cycles. Primers and melting temperatures are shown in Supplementary Table 5.

### Sequencing of PCR products

PCR products were separated on a 1.2% agarose gel; DNA was purified from the gel using a QIAquick gel extraction kit (QIAGEN, 28704) and cloned into a T-tailed pBluescript II SK (+) vector (Stratagene, 212205) that had been digested with EcoRV-HF (NEB, R3195). Positive clones

were sent for Sanger sequencing (Eurofins Genomics, LightRun Tube) using a M13 forward sequencing primer.

Sequencing traces (in .ab1 format) were aligned using SnapGene (v4; GSL Biotech). For recombination, sequences were aligned against the human immunoglobulin kappa (NCBI Gene ID: 50802) or lambda loci (NCBI Gene ID: 3535), as appropriate. For ESCs, the possible head-to-head SJ sequence was assembled from the appropriate genomic sequence and sequences were aligned to this. All alignments were verified by BLASTN (NCBI; accessed at https://blast.ncbi.nlm.nih.gov), where the search set was limited to the *Homo sapiens* (taxid: 9606) RefSeq Genome Database (refseq_genomes).

### ddPCR

ddPCR reactions were conducted in a total volume of 20 µl with 1× ddPCR Supermix for probes (Bio-Rad 1863026), 900 nM of each primer, 250 nM probe and 63 ng template DNA. Droplets were generated in an 8-well droplet generation plate using a Bio-Rad QX100 droplet generator. Nanodroplets were carefully transferred to a 96-well plate, which was sealed with foil prior to thermocycling. The latter involved an initial denaturation at 95 °C for 10 min, followed by 40 cycles of 94 °C for 30 s and 60 °C for 1 min, followed by 98 °C for 10 min and 4 °C for 30 min to allow droplets to equilibrate. The presence of amplified products was determined using the Bio-Rad QX100 Droplet Reader and QuantaSoft v1.7.4 software. For positive droplet identification, a manual threshold (2000) was applied to 1D amplitude data to minimize the occurrence of false positives. Primers are shown in Supplementary Table 5, for absolute quantitative analysis of *GAPDH*, recombination junctions and SJs in BCP-ALL samples.

### Targeted sequencing of recombination and SJs (LAM-ESC and LAM-recombination)

Targeted sequencing of light chain recombination and SJs was performed using modified versions of the LAM-HTGTS technique[32] with bespoke analysis pipelines. Recombination junctions were detected using LAM-recombination, where bait primers (Supplementary Table 5) were designed against regions adjacent to J gene segments, allowing recombination to V gene segments to be determined. SJs were detected via LAM-ESC where bait primers (Supplementary Table 5) were designed against regions adjacent to J segment RSSs, allowing any sequence (for example, V RSSs) joined to J RSSs to be determined.

Libraries were generated as described[32] with minor modifications. Specifically, 500 ng genomic DNA was used as template in 90 cycles of the initial Bio-PCR. For the final PCR step (Tagged PCR), primers were used to add sequencing adaptors (Amplicon-EZ-I7-blue and Amplicon-EZ-I5-nested; Supplementary Table 5). Following library generation, samples were sent for 2× 250 bp paired-end sequencing using the Amplicon-EZ service (Azenta).

To analyse the data, FASTQ files were initially demultiplexed using each J gene segment or J RSS nested primer, for recombination and SJ libraries, respectively. The paired-end reads were then combined into a single read, using the overlap at the 3′ end of read 1 and 5′ end of read 2. If reads could not be combined, read 1 only was analysed. A custom Python script was used to automate BLAST searches against a custom BLAST database, consisting of all V-J recombination events or all head-to-head RSS combinations from the immunoglobulin kappa and lambda loci, for recombination and SJ libraries, respectively https://github.com/Boyes-Lab/LAM-ESC-Recombination[33].

### Clonotype determination

Recombination junctions were amplified using LV3-1_REC_F, LV5-45_REC_F or LV2-11_REC_F with LJ2/3_REC_R (Supplementary Table 5). Following gel purification of the amplified products using the QIAquick gel extraction kit (QIAGEN, 28704), samples were sent for 2 ×250 bp paired-end sequencing using the Amplicon-EZ service (Azenta) that included addition of Amplicon-EZ-I7 and Amplicon-EZ-I5 sequencing

adaptors. Paired-end reads in the .fastq format were combined by overlapping the 3′ end of read 1 with the 5′ end of read 2 (EGAD50000001518). The reads were compared to the reference motifs near the breakpoint junction of interest using the script Clonotype_analysis.py (https://github.com/Boyes-Lab/NGS-Analysis)[48], which identified each unique clonotype and the frequency at which it occurred. The reference motifs consisted of 5 bp from each of the respective V and J motifs, derived from sequences that lie 20 bp from the V-J junction that would be formed in the absence of processing. Specifically, the reference motifs were: 5 bp of V gene reference sequence; omit 20 bp of V gene sequence to the boundary; omit 20 bp of J gene sequence; use 5 bp of J gene sequence as reference. If the amplified sequence contained both of the 5 bp motifs, then the code identifies each unique sequence that intervenes. The identified sequences were then inputted into IgBlast[55] which determined if the insert at the V-J junction was derived from elsewhere in the genome. The number of unique clonotypes was determined from these inserts: two sequences with the same V-J insert were classed as being the same clonotype.

The number of recombination copies attributable to each distinguishable minor clonotype was determined by calculating the absolute copy number of the recombination event by ddPCR and multiplying that by the percentage of each minor clonotype. Assuming each recombination is present on a single allele, the number of cells harbouring the recombination event was then estimated (that is, one recombination copy equates to one cell harbouring the recombination). The number of cell divisions ($n$) required to generate the recombination levels measured for each minor clonotype was subsequently calculated using the formula for cell population doublings: $N_2 = N_1(2^n)$ where $N_1$ = number of cells at beginning (that is, 1 cell) and $N_2$ = number of cells at end (that is, the estimated number of cells). From this, the minor clonotypes were calculated to be present at copies equivalent to 0.81–3.69 relative cell divisions (patients R-8 and R-13).

To investigate if the SJ levels observed via ddPCR (Extended Data Fig. 7c) could have resulted from replication of SJs from the minor clonotypes, the number of SJs predicted to remain if ESCs are diluted at each cell division were calculated using the formula for exponential decay: $x_t = x_0/2^t$ where $x_t$ = predicted SJ level, $x_0$ = initial SJ level (that is, 1) and $t$ = number of cell divisions (calculated above). Theoretically, the SJ measured by ddPCR could correspond to one or more of the minor clonotypes; we therefore took a conservative approach and summed the predicted SJ values for all the distinguishable clonotypes. We then divided the observed SJ value (Extended Data Fig. 7c) by the predicted SJ levels for the minor clonotypes. This observed/predicted value was compared to the observed/predicted values for SJs generated a similar number of relative cell divisions ago, using the values shown in Fig. 3c, where the SJs were estimated to result from 0.33–6.32 relative cell divisions. The SJ copies found in patient R-8 are substantially higher than those generated by replication of SJs from similarly recent recombination events (Extended Data Fig. 7d), implying that at least some of the SJs have persisted from the primary recombination event. Similar analyses cannot discount the possibility that replication of recently generated SJs generated the SJs observed in patients R-13 and R-9.

## Calculation of observed/predicted ESC levels

The observed SJ/predicted SJ ratio provides a more accurate measure of ESC replication as it takes the extent of cell division into account. It ensures that ESCs from cells that have undergone marked differences in cell division do not artefactually show the same level of replication. Predicted SJ levels were calculated using the formula for exponential decay given above. For Fig. 3c, only recombination junctions that had undergone ≤6 cell relative divisions per ddPCR sample were examined.

## Phi29 amplification

Freeze-thawing of cells causes DSBs[56,57] and depletion of circular DNAs in patient samples. Phi29 was therefore used to amplify remaining circular DNAs in BCP-ALL samples via rolling circle replication, using the Illustra TempliPhi 500 Amplification kit (Cytiva 25640010) according to the manufacturer's protocol. Specifically, 10 ng of DNA was diluted with 50 μl of sample buffer, incubated at 95 °C for 3 min and cooled to 4 °C. The reaction was then incubated in 1× Phi29 reaction buffer (50 mM Tris-HCl pH 7.5, 10 mM MgCl$_2$, 10 mM (NH$_4$)$_2$SO$_4$, 4 mM DTT) with 2 μl of enzyme mix at 30 °C for 18 h (human DNA) or 6 h (mouse DNA), followed by 65 °C for 10 min to inactivate the enzyme. DNA was precipitated with isopropanol, washed with 70% ethanol and resuspended in 25 μl ddH$_2$O. Control experiments omitted the enzyme. Sample concentrations were determined via absorbance at 260 nm (using a DeNovix DS-11 spectrophotometer) and diluted to 7 ng/μl. SJs were quantified by ddPCR and the SJ/*GAPDH* ratio of treated versus untreated sample determined.

## Fluorescence in situ hybridization

DNA-FISH was performed as described[58]. Fosmid clones targeting *IGK* (ABC10-44246300H4), *IGK*-JK-KDE (ABC8-2123240B1) and *IGL* (ABC10-44455600K21) ESCs as well as *IGK* and *IGL* control regions (ABC10-43608900D2 and ABC10-44444000A2, respectively) were gifts from E. Eichler. Each fosmid probe was directly labelled by nick translation as described[59], except the amount of DNA labelled was reduced to 1 μg and both aminoallyl-dUTP and aminoallyl-dCTP were incorporated, followed by coupling to fluorescent dyes (Alexa Fluor 488/555/647, Invitrogen). Fosmid probes were purified using a QIAquick PCR purification kit and elution in 10 mM Tris-HCl pH 8.5. Patient bone marrow samples were cultured in StemSpan SFEMII medium (Stem Cell Technologies) supplemented with 20% fetal calf serum, 1% L-glutamine, 100 μg ml$^{-1}$ Primocin (InvivoGen), 20 ng ml$^{-1}$ IL-3 and 20 ng ml$^{-1}$ IL-7 (Cell Guidance Systems) in a 37 °C humidified incubator, 5% CO$_2$ prior to Colcemid treatment (0.2 μg ml$^{-1}$, KaryoMAX, ThermoFisher) for 2 h. hTERT-RPE-1 cells were maintained in supplemented DMEM and treated with Colcemid as described above. Cells were then centrifuged and resuspended in prewarmed 75 mM KCl, followed by incubation at 37 °C for 20 min. After further centrifugation, cells were resuspended in Carnoy's fixative (methanol: glacial acetic acid 3:1) and dropped onto humidified microscope slides. Slides were incubated in 2× SSC/RNase A 100 μg ml$^{-1}$ at 37 °C for 1 h, followed by successive dehydration in 70%, 90%, and 100% ethanol. Slides were heated to 70 °C for 5 min on a hot plate, followed by DNA denaturation in preheated denaturant (2× SSC/70% formamide) at 70 °C for 30 min. Subsequently, slides were placed in ice-cold 70% ethanol, then 90% and 100% ethanol at room temperature before air-drying. For each slide, 100 ng of fosmid probe was combined with 6 μg human Cot-1 DNA (Invitrogen) and 5 μg single stranded DNA from salmon testes (Invitrogen), followed by ethanol precipitation. The DNA pellet was washed with 70% ethanol and resuspended in hybridization buffer (2× SSC, 50% deionized formamide, 10% dextran sulfate, 1% Tween-20). FISH probes were denatured at 92 °C for 5 min, pre-annealed at 37 °C for 15 min and then were immediately hybridized with DNA on slides overnight at 37 °C in a light-tight humidified chamber. Slides were washed in 2× SSC at 45 °C, 0.1× SSC at 60 °C, 1× PBS with 10 μg ml$^{-1}$ DAPI at room temperature and finally mounted with SlowFade Gold antifade reagent (Invitrogen). Slides were imaged using an Olympus IX83 widefield fluorescence microscope with a 60× (60×/1.4 Oil, Plan Apo (oil)) objective and a Photometrics Prime BSI CMOS camera with a motorized *xyz* stage. Filter sets are DAPI (excitation 365/10 nm) emission 440/40 nm, GFP (excitation 482/24 nm) emission 530/40 nm, RFP (excitation 545/10 nm) emission 600/50 nm, Cy5/A647 (excitation 628/40 nm) emission 692/40 nm. Images were acquired using Olympus CellSens Dimension 3.2 (Build 23706) software and analysed using FIJI 2.16.0 software.

## BrdU immunofluorescence

Bone marrow cells were resuspended in StemSpan SFEMII medium, supplemented as described above, and labelled with 10 μM BrdU

(Merck B5002) for 28 or 48 h in a 37 °C humidified incubator, 5% $CO_2$. Primary BCP-ALL cells[60] have a doubling time of 26 to 240 h. Therefore, for cells incubated with BrdU for 28 h, incorporation should be limited to a single S phase for any cells in metaphase. Chromosome spreads were prepared as described for DNA-FISH. DNA was denatured by incubating the slides in 1 M HCl for 40 min at room temperature, followed by neutralization in 0.1 M Borate buffer pH 8.5 for 15 min at room temperature. Slides were then washed in 0.1% Triton X-100/PBS and blocked in 0.1% Triton X-100/PBS/5% goat serum (Merck G9023) for 1 h at room temperature. Following immunostaining with a 1:500 dilution of anti-BrdU antibody (BD Pharmingen, 555627) for 1 h at room temperature, cells were incubated with goat anti-mouse secondary antibody (Jackson ImmunoResearch, 115-001-003) at a 1:1,000 dilution for 1 h at room temperature, followed by incubation with Alexa Fluor Plus 488 labelled donkey anti-goat antibody (ThermoFisher, A32814), also at 1:1,000 dilution for 1 h at room temperature. Following counterstaining with DAPI (10 μg ml$^{-1}$, Invitrogen), images were captured using an Olympus IX83 widefield fluorescent microscope and CellSens software and analysed using FIJI 2.16.0 software as above.

## PCR amplification of the *SYK* gene
PCR was performed using Taq DNA polymerase (NEB, M0267) using the conditions described above for BIOMED-2 primers except the final volume was 25 μl. Primers are given in Supplementary Table 5. Cycling conditions involved initial denaturation at 95 °C for 3 min, followed by 95 °C for 20 s, 64 °C for 30 s and 68 °C for 30 s for 39 cycles, followed by a final extension of 5 min at 68 °C. PCR products were separated by gel electrophoresis.

## Analysis of *ETV6*–*RUNX1* BCP-ALL WGS data for ESCs
WGS datasets from patients with BCP-ALL (downloaded from the European Genome-phenome Archive (EGA), dataset ID: EGAD00001000116) were analysed for SJs using an in-house Python script (https://github.com/Boyes-Lab/NGS-Analysis)[48]. In brief, paired-end sequencing data was aligned to the hg19 build of the human genome using Bowtie2 in local alignment mode. Following alignment to the genome, the data were filtered for discordant reads at the immunoglobulin and TCR loci using Samtools. Reads were further filtered (using the above in-house Python script) to extract divergent reads, indicative of SJs. Similar tools that capitalize on discordant paired-end reads have been developed to map ecDNAs. However, AmpliconArchitect[40] also requires increased circular DNA copy number whereas CircleSeq involves removal of linear DNA prior to sequencing and analysis[44,61]. AmpliconArchitect[40] did not detect circular DNAs in WGS from 20 patient samples where ESCs were detected by LAM-ESC (Fig. 2; EGA accession code: EGAS00001006863). This is likely because unlike ecDNAs, individual ESC sequences do not undergo copy number amplification that is required for detection by AmpliconArchitect[40].

## Identification of differentially expressed genes and GSEA analysis
RNA-seq data of patients at diagnosis were downloaded from the TARGET database (dbGaP Sub-study ID: phs000464; Fig. 3d) or EGA (Accession Code EGAS00001006863; Fig. 3f). Differentially expressed genes were identified using DESeq2 with |logFC| > 0.585 and FDR < 0.05. Adjustment for multiple comparisons was performed via the DESeq2 programme. Gene set enrichment analysis (GSEA) was carried out according to the user guide provided by the BROAD Institute (https://docs.gsea-msigdb.org/#GSEA/GSEA_User_Guide/).

## Whole-exome sequencing
DNA was prepared from germline and BCP-ALL patient samples taken at diagnosis and relapse, using the Promega Wizard HMW DNA extraction kit. The concentration was measured via Qubit and whole-exome paired-end read sequencing was performed at 100× depth by Azenta.

## Analysis of SVs
SV data of patients at diagnosis and relapse were downloaded from Complete Genomics (CGI, from within the TARGET database: dbGaP Sub-study ID: phs000464; Fig. 5a,b). For Extended Data Fig. 9d, WGS of patients with BCP-ALL in the VIVO Biobank cohort was downloaded from EGA under the Accession Code EGAS00001006863. The analysis used similar numbers of WGS and WES where WES from germline and BCP-ALL samples are available from EGA (dataset ID: EGAD50000001519). Breakpoints of SVs were analysed using a bespoke Python program, SVs_near_RSSs.py, which creates an analysis window spanning 50 bp either side of each breakpoint (https://github.com/Boyes-Lab/Structural-Variants)[62]. The presence of an RSS within the window is then analysed using the DNAGrep algorithm, via RSSsite[63].

The relative impact of cut-and-run, ESC reintegration and RAG-mediated insertion was determined as follows: SVs were defined as either deletions, insertions, translocations or complex insertions based on the source of the DNA strand on either side of an SV breakpoint. The DNA damage occurring near RSS sites was analysed, as RAG-mediated DSBs are required for both reintegration and cut-and-run-mediated DNA damage. ESC reintegration events are defined by the reinsertion of ESC DNA into the genome. Thus, insertions were considered to be reintegration-derived where the inserted DNA came from immunoglobulin or TCR loci. The proportion of these events near RSS sites was then compared with other types of DNA damage. Open-source scripts to analyse potential cut-and-run events compared to reintegration are provided at https://github.com/Boyes-Lab/Structural-Variants[62].

## Analysis of ecDNAs using AmpliconArchitect[40]
In silico experiments to detect ecDNAs were carried out using AmpliconArchitect[40] according to the AmpliconSuite pipeline (https://github.com/AmpliconSuite/AmpliconSuite-pipeline/blob/master/documentation/GUIDE.md) using WGS BAM files from VIVO Biobank patient samples available from EGA (Accession Code EGAS00001006863).

## Statistics
All statistical analyses were performed using GraphPad Prism v9. Statistical test results are provided as $P$ values in the figures. Detailed descriptions of error bars and the number replicates and/or cells analysed are reported in the figure legends. Biological replicates are shown unless otherwise indicated. Analyses of fold changes between biological replicates were performed using a two-tailed Student's t test or Mann–Whitney $U$ test (depending on data distribution) where *$P < 0.05$, **$P < 0.01$, ***$P < 0.001$, ****$P < 0.0001$. The 95% confidence intervals are given in the figure legends where possible. The 95% confidence intervals for the difference of mean gene expression (H-relapse versus non-relapse) in Fig 3e: *PCNA*: −2.007 to −0.1691; *POLE3*: −2.206 to −0.1488; *POLE4*: −4.910 to −0.8914; *RBX1*: −2.206 to −0.6157. The Kolmogorov–Smirnov test was used to determine whether the data followed a Gaussian distribution. GSEA uses the statistical test described[53,54] with corrections for multiple comparisons. DeSeq2 analysis used the Wald statistical test with the Benjamini–Hochberg correction for multiple testing. Statistical analyses with two categorical variables were performed using a two-way ANOVA. Statistical analyses of the proportion of ESCs above the threshold in patients who relapse versus those who do not were determined using a two-tailed Fisher's exact test. The significance of the difference between matched ESCs in relapse and non-relapse groups was determined using a two-tailed Wilcoxon signed-rank test. Pearson correlation coefficients ($r$ values) were computed for scatter plots and tested (null $r = 0$) with a standard two-tailed test. The significance of SVs involving single cRSSs occurring more frequently at relapse-associated genes was calculated using the hypergeometric distribution (Fig. 5b and Extended Data Fig. 9d) to analyse the number of SVs involving single cRSSs within relapse-associated genes compared to SVs involving single cRSSs in the whole dataset. The significance of the co-localization of

breaks detected by LAM-HTGTS and genes that are frequently mutated in relapsed ALL (Fig. 5c) was calculated by using the hypergeometric distribution (implemented in the R software, https://www.r-project.org) to test whether the number of genes which were more commonly mutated with SJ partner versus 12-RSS or 23-RSS partners (controls) occurred more frequently in the relapse-associated genes versus the whole gene list. A power calculation, taking an alpha value of 0.05 and a desired power of 80% was used to initially estimate the sample size in which to analyse ESC levels.

## Reporting summary

Further information on research design is available in the Nature Portfolio Reporting Summary linked to this article.

## Data availability

WGS datasets from 61 patients with *ETV6–RUNX1* BCP-ALL were downloaded from the European Genome-phenome Archive (EGA) (dataset ID: EGAD00001000116). The human genome sequence hg19, (GCA_000001405.14) release GRCh37.p13, was downloaded from https://hgdownload.soe.ucsc.edu/goldenPath/hg19/bigZips/. SV data for patients with BCP-ALL at diagnosis and relapse were downloaded from Complete Genomics (CGI, from within the TARGET database; dbGaP sub-study ID: phs000464). RNA-seq data of patients with BCP-ALL at diagnosis were downloaded from the TARGET database (dbGaP sub-study ID: phs000464). RNA-seq and WGS datasets from patients with BCP-ALL in the VIVO Biobank cohort were downloaded from EGA under the accession code EGAS00001006863. Raw LAM-ESC and LAM-recombination sequences are available from the European Genome-phenome Archive (EGA) under the dataset ID EGAD50000000597. The extracted recombination junctions and ESCs are given in Supplementary Tables 2 and 3. WES data from patients with BCP-ALL in the VIVO Biobank cohort are available from EGA via the dataset ID EGAD50000001519. Amplicon sequencing data of the recombination junctions used for clonotype analysis are available from EGA under the dataset ID EGAD50000001518. Source data for all graphs is available as Excel spreadsheets for main figures, extended data and supplementary figures. FISH data are available via Research Data Leeds (https://doi.org/10.5518/1693 (ref. 64)). The sample cohort used for each figure is given in Supplementary Table 6. Source data are provided with this paper.

## Code availability

A custom Python script to analyse LAM-ESC and LAM-recombination data, together with a custom database is available at https://github.com/Boyes-Lab/LAM-ESC-Recombination[33]. The script automates BLAST searches against a custom BLAST database, consisting of all V-J recombination events or all head-to-head RSS combinations from the immunoglobulin kappa and lambda loci, for recombination and SJ libraries, respectively. A custom Python script to analyse SJs in WGS datasets is available at https://github.com/Boyes-Lab/NGS-Analysis[48]. A bespoke Python programme to analyse breakpoints of SVs, SVs_near_RSSs.py, is available at https://github.com/Boyes-Lab/Structural-Variants[62]. This creates an analysis window spanning 50 bp either side of each breakpoint; the presence of an RSS within the window is then analysed using the DNAGrep algorithm, via RSSsite. A custom Python script to analyse potential cut-and-run events compared to reintegration events is provided at https://github.com/Boyes-Lab/Structural-Variants[62]. The programme to determine the clonotypes present in sequencing reads compared to the reference motifs near the breakpoint junction of interest, Clonotype_analysis.py, is available at https://github.com/Boyes-Lab/NGS-Analysis[48].

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

**Acknowledgements** The authors acknowledge funding from the Little Princess Trust and Harley Staples Cancer Research Trust, via the Children's Cancer and Leukaemia Group (CCLGA 2019 15; CCLGA 2020 12 and CCLGA 2023 06), Blood Cancer UK (grant 15042), a Wellcome Trust ISSF award (to J.N.F.S.; institutional grant 204825/Z/16/Z), a China Scholarship Council studentship (to X.W.), a Mary and Alice Smith Research Scholarship (to M.P.E.) and a Leeds Doctoral Scholarship (to D.C.). D.R.W. is supported in part by the National Institute for Health and Care Research (NIHR) Leeds Biomedical Research Centre (BRC) (NIHR203331). The views expressed are those of the authors and not necessarily those of the NHS, the NIHR or the Department of Health and Social Care. Primary childhood leukaemia samples and data were provided by VIVO Biobank supported by Cancer Research UK & Blood Cancer UK (grant CRCPSC-Dec21\100003). Further samples were provided by J. Zuna. We also gratefully acknowledge E. Eichler for fosmid clones. The results presented here are in part based upon data generated by the Therapeutically Applicable Research to Generate Effective Treatments (https://www.cancer.gov/ccg/research/genome-sequencing/target) initiative, phs000218; the data used for this analysis are available at the Genomic Data Commons (https://portal.gdc.cancer.gov). We thank the Sanger Cancer Genome Project, primarily funded by the Wellcome Trust, for making available the EGA dataset (ID: EGAD00001000116). Those who carried out the original analyses and collection of dataset ID EGAD00001000116 bear no responsibility for the data presented here. We also thank W. Bickmore, K. Purshouse and S. Boyle, A. Corcoran, C. Bassi and S. Horsley for advice on FISH experiments; staff in the flow cytometry and bioimaging facility, University of Leeds, for advice on microscopy; and J. Ladbury for comments on the manuscript.

**Author contributions** Z.G. performed the LAM-ESC experiments, qPCR analyses of SJ and recombination junctions, DESeq2, GSEA analyses, in depth analyses of LAM-ESC and LAM-recombination data, FISH experiments (with R.d.T., C.C. and L.J.R.), WES, analysis of the VIVO Biobank patient WGS and WES (with data from S.R. and A.V.M.), and with M.P.E., RT–qPCR analyses of gene expression levels in patients and bioinformatic analyses of SVs in the TARGET database. J.N.F.S. developed the LAM-ESC and LAM-recombination assays, performed the mouse RecBCD experiments as well as bioinformatic analyses of SJs in patients with *ETV6–RUNX1* BCP-ALL using the EGA data and, with A.M.F., C.C. and R.d.T., analysed recombination events and SJs by PCR plus amplicon sequencing. M.P.E. performed the LAM-recombination experiments, clonotype analyses, amplification of ESCs by Phi29 and examined SJ and recombination levels by ddPCR. ddPCR experiments to measure SJ and recombination levels for matched ESCs in patients who relapsed and those who did not relapse were performed by M.P.E. and D.C. D.C. measured high copy SJ levels by ddPCR, analysed SJs corresponding to recent recombination events, determined mouse recombination in *Gapdh* ratios, performed control ddPCR and qPCR assays, identified SJs that are frequently present at higher copies in LAM-ESC data and compared mapped replication initiation sites with ESC sequences. X.W. determined the SJ/recombination junction ratio at different stages of mouse B cell development and analysed *Rag1* expression in mouse B cells by RT–qPCR. A.D.G. wrote the code to analyse SVs with cRSSs at their breakpoints. J.B. performed RecBCD experiments for BCP-ALL samples with C.C. and R.d.T. A.V.M. provided detailed genomic subtype annotation for patient samples. Z.G. produced all figures and schematic images. D.R.W. advised on statistics and bioinformatic analyses. All authors analysed the data. J.B., D.R.W. and A.M.F. devised the overall study and wrote the manuscript with Z.G. and J.N.F.S.

**Competing interests** The authors declare no competing interests.

**Additional information**
**Correspondence and requests for materials** should be addressed to Joan Boyes.

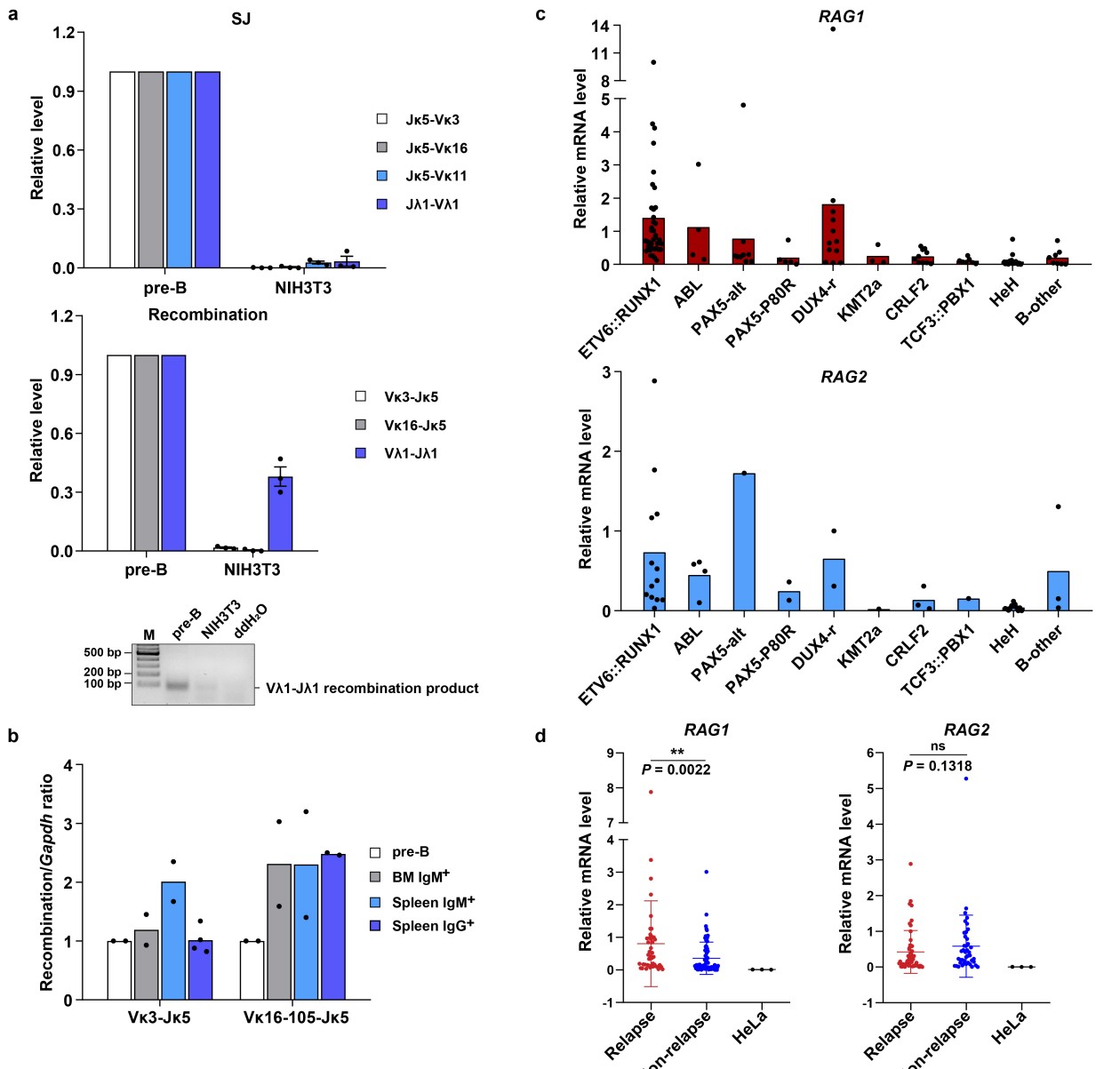

**Extended Data Fig. 1 | Elevated *RAG1* expression at diagnosis correlates with subsequent relapse. a**, Control qPCR experiments showing amplification of recombination and SJ junctions using pre-B and NIH3T3 (non-RAG-expressing) DNA. $n$ = 3 samples. Igλ recombination accounts for only ~5% of mouse IgL recombination[65]; this reduces the difference between specific Vλ1-Jλ1 amplification using pre-B DNA and non-specific amplification (as determined by melt curve analysis) using NIH3T3 DNA. Gel electrophoresis (lower) confirms only minimal amplification of the correct product in NIH3T3 compared to pre-B DNA. For source gel data, see Supplementary Fig. 2a. Mean values are shown ± s.d. **b**, Recombination junction levels were determined by qPCR and normalised to genomic *Gapdh* in mouse B cells at the stages of development shown. The ratio in pre-B cells was set at one and other ratios are plotted against this. Mean values are plotted. $n$ = 2 samples. **c**, Levels of *RAG1* (upper)

and *RAG2* (lower) expression in different BCP-ALL subtypes using samples taken at diagnosis and determined by RT-qPCR. Data are normalised to *HPRT* expression levels. Mean values are plotted; black dots indicate values for individual patients. $n$ = 121 (*RAG1*) and $n$ = 92 (*RAG2*). Samples provided by VIVO Biobank. **d**, Analysis of *RAG1* (upper) and *RAG2* (lower) expression by RT-qPCR in BCP-ALL patient samples (from VIVO Biobank). Samples from patients who subsequently relapse are shown in red whereas those who are not known to have relapsed are in blue. Data are normalized to *HPRT* expression. For *RAG1* expression, $n$ = 44 (Relapse) and 77 (Non-relapse). For *RAG2* expression, $n$ = 50 (Relapse) and 42 (Non-relapse). HeLa cDNA is a negative control. Mean values are shown ± s.d. The significance of the difference in expression between patients who later relapse and those who do not was determined by a two-tailed Mann-Whitney $U$ test.

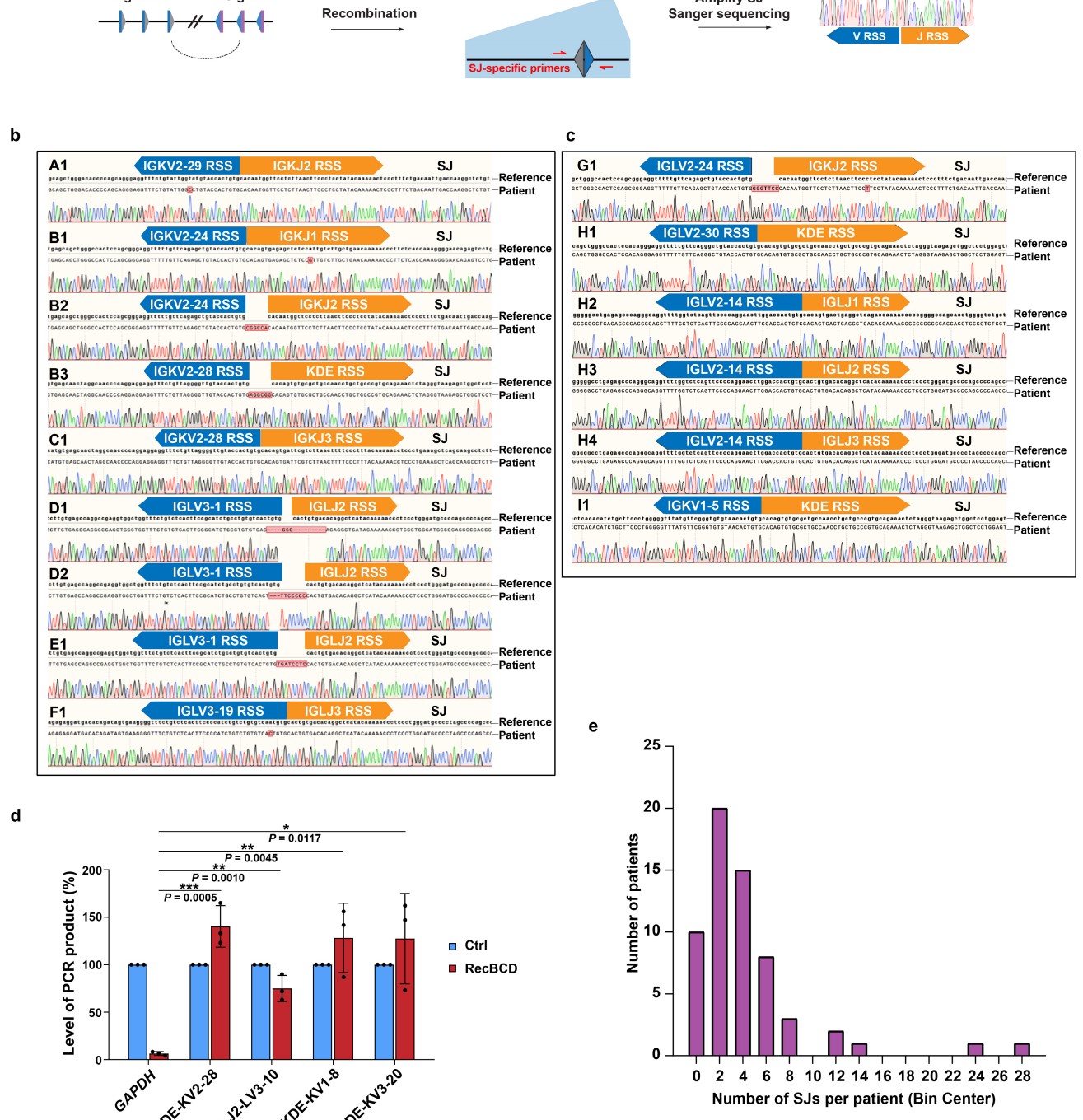

**Extended Data Fig. 2 | ESCs are present as non-chromosomal DNA in BCP-ALL patients. a**, Sequences of recombination events in each *ETV6::RUNX1* + BCP-ALL sample were determined via PCR (using degenerate primers from the BIOMED-2 consortium[31]) and sequencing of the amplified products. The SJ of the corresponding ESC was amplified and sequenced. **b**, Sequences of SJs from *ETV6::RUNX1* + BCP-ALL patients (lower), aligned to the sequences surrounding the respective 12- and 23-RSS in the genome (upper, bold). Individual patients (*n* = 6) are denoted by letters A-F. The SJs detected per patient are numbered. All SJs result from deletional recombination. Samples from a hospital in the Czech Republic with informed consent[19]. **c**, SJs are present in different BCP-ALL subtypes. The SJs in low-hypodiploid (patient G), *CRLF2*-r (patient H) and *BCR::ABL1* (patient I) BCP-ALL samples were detected as above. The SJs detected per patient are numbered 1, 2, 3, etc. *n* = 3 patients. Patient samples provided,

with consent, by the Haematological Malignancy Diagnostic Service, Leeds. **d**, ESCs persist as circles in BCP-ALL samples. The level of undigested DNA following RecBCD treatment was determined by ddPCR; *GAPDH* (linear, genomic DNA) is a negative control. Untreated DNA is a further control (Ctrl). *n* = 3 BCP-ALL samples. Mean values are shown ± s.d. Black dots show the mean of 3-4 technical repeats. The significance of the difference between RecBCD-treated *GAPDH* and SJs was determined by an unpaired, two-tailed Student's t-test. 95% CI: *GAPDH* vs KDE-KV2-28: 98.59 to 169.2; *GAPDH* vs LJ2-LV3-10: 46.30 to 90.86; *GAPDH* vs KDE-KV1-8: 63.22 to 180.3; *GAPDH* vs KDE-KV3-20: 44.64 to 197.5. **e**, The total number of SJs identified in WGS data of 61 *ETV6::RUNX1* + BCP-ALL patients. The *x*-axis shows the number of SJs identified per patient whereas the *y*-axis shows the number of patients with that number of SJs. NGS data from the European Genome-phenome Archive (EGAD00001000116).

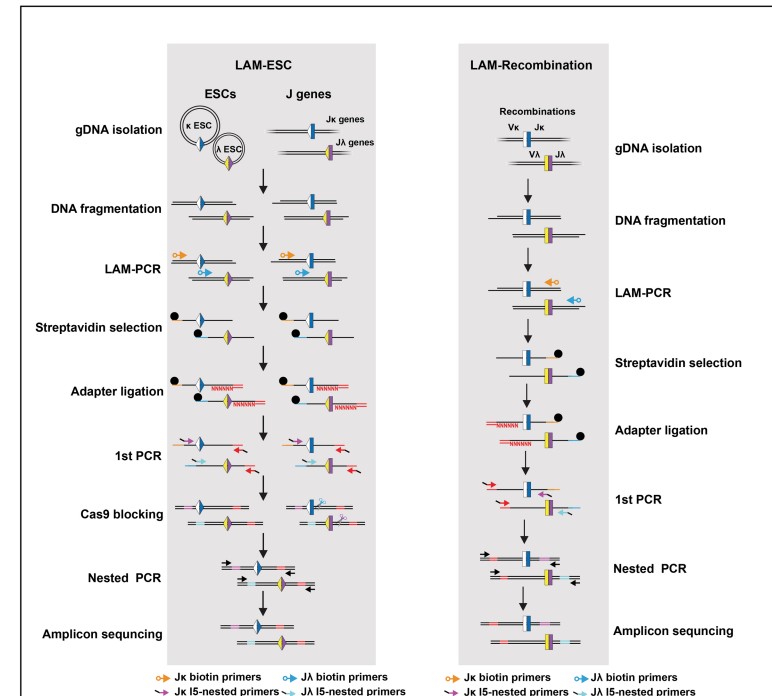

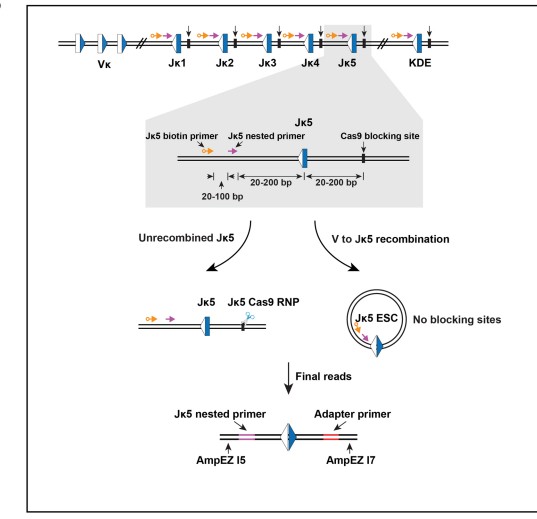

**Extended Data Fig. 3 | Schematic of LAM-ESC and LAM-recombination methods. a**, LAM-ESC and LAM-recombination are based on LAM-HTGTS that was originally described by Hu et al.[32]. LAM-ESC and LAM-recombination directly amplify SJ and coding junctions from sonicated genomic DNA using linear amplification-mediated PCR, followed by bead enrichment and on-bead bridge adapter ligation. This enables exponential PCR amplification and labelling of the enriched PCR products with MiSeq (AmpEZ I5 and I7) adapters. Cas9-mediated blocking is introduced to remove germline sequences from amplified LAM-ESC products. The purified Cas9/sgRNA (RNP) complex is targeted by sgRNAs specific to the *IGK* and *IGL* loci. **b**, Schematic of the designed biotinylated oligo,

nested primer and a Cas9 targeting site for LAM-ESC for a given J gene segment. The amplified libraries are sequenced via Amplicon sequencing and the reads mapped to the *IGL* and *IGK* loci using a custom Python script (https://github.com/Boyes-Lab/LAM-ESC-Recombination)[33]. ESCs resulting from inversional recombination events are excluded from our analyses. Likewise, ESCs from intra-KV recombination, between *bona fide* 12-RSSs and flipped 23-RSSs at a subset of KV gene segments[34], are not detected, since LAM-ESC involves amplification from J regions. The specificity of the pipeline was verified via control experiments with HeLa genomic DNA that generated only negligible background reads for LAM-ESC or LAM-recombination.

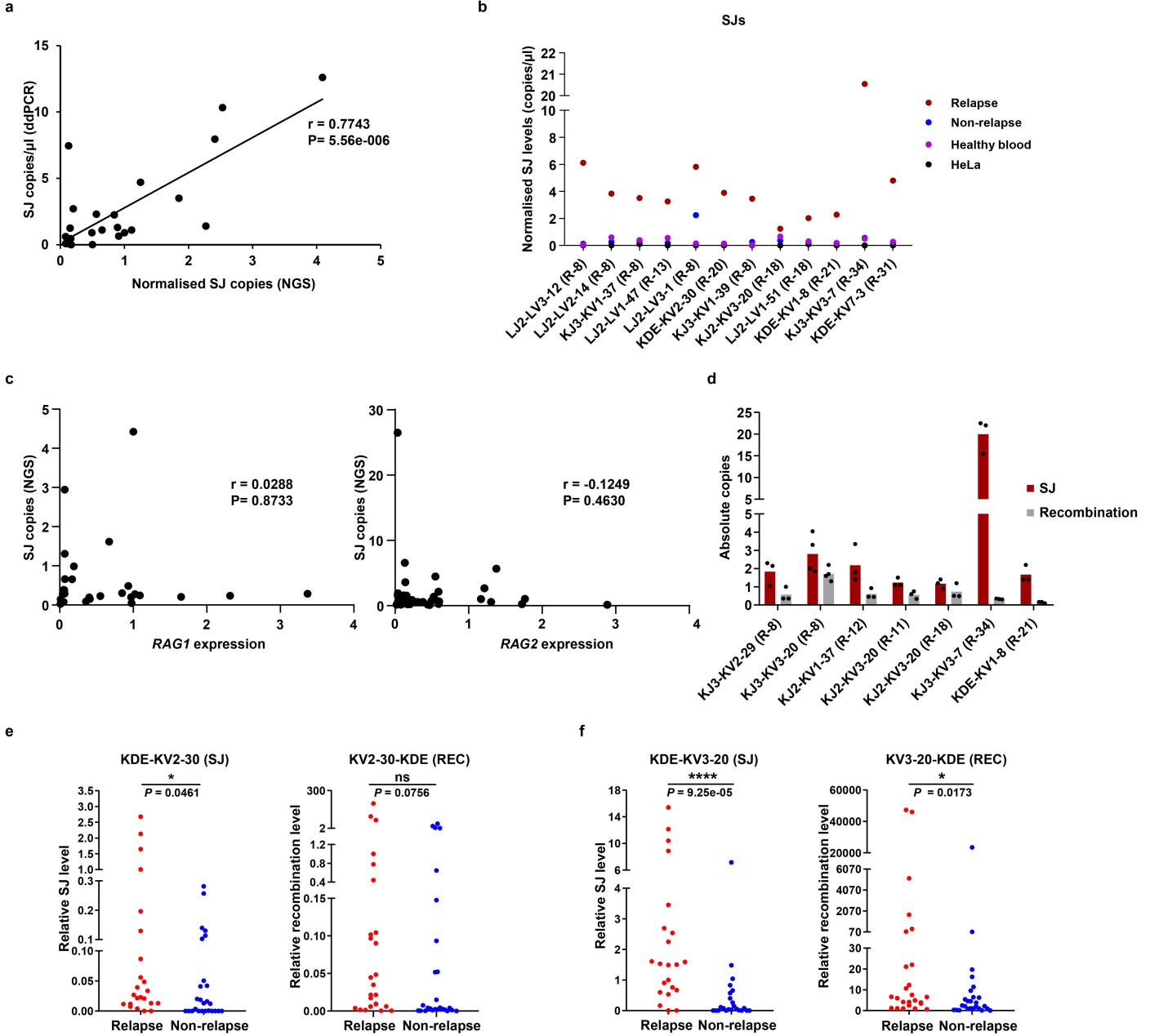

**Extended Data Fig. 4 | Comparison of SJ, RAG gene expression and recombination levels. a**, The levels of 27 SJs that are frequently detected by LAM-ESC were determined by ddPCR using samples from 11 patients and plotted against the normalised levels of the respective SJs determined by LAM-ESC. A strong positive correlation (Pearson, *r*) is observed between the normalised SJ reads determined by NGS and the absolute SJ copies/µl as determined by ddPCR (statistical significance from a two-sided test). **b**, The levels of 12 SJs found above the threshold in healthy blood (Fig. 2a) were measured by ddPCR. Levels in patients who later relapse (*n* = 7) are shown in red; the levels of the same ESCs in patients who did not relapse (*n* = 10) are shown in blue. SJs in healthy blood (*n* = 2) are shown in pink whereas control amplifications with HeLa DNA (*n* = 2) are shown in black. Levels were normalised to the *GAPDH* gene. Complementary clonotype analyses of SJs shown in Fig. 2a are given in Supplementary Fig. 3. **c**, The number of SJ copies that are above the threshold found in healthy blood are plotted per patient against the corresponding expression levels of *RAG1* (*n* = 26) or *RAG2* (*n* = 36). No significant correlation is found between *RAG1* (left) or *RAG2* (right) expression and high SJ

copies per patient following calculation of a two-tailed Pearson correlation coefficient. **d**, Absolute copies of the SJ and corresponding recombination junction determined by ddPCR in cases where the normalised SJ reads exceed the normalised recombination reads in LAM-ESC and LAM-recombination experiments. Black dots indicate technical repeats. Mean values are plotted. *n* = 3 technical repeats for six patients. **e, f**, qPCR analysis of the levels of (**e**) KDE-KV2-30 SJ and the corresponding recombination event (REC) and (**f**) KDE-KV3-20 SJ and the corresponding recombination event in BCP-ALL patients. Subsequent relapse status is plotted (red – relapse; blue – non-relapse). The levels relative to *GAPDH* (genomic DNA) control are shown. The significance of differences in levels was calculated by a two-tailed Mann-Whitney *U* test. For (e), SJ: *n* = 23 (Relapse) and *n* = 26 (Non-relapse); REC: *n* = 25 (Relapse) and *n* = 26 (Non-relapse). For (f), SJ: *n* = 22 (Relapse) and *n* = 23 (Non-relapse); REC: *n* = 25 (Relapse) and *n* = 26 (Non-relapse). Since recombination to KDE precludes further recombination (Fig. 1a), recombination junctions cannot be lost via secondary recombination but instead, these data imply that there is increased ESC replication in patients who later relapse.

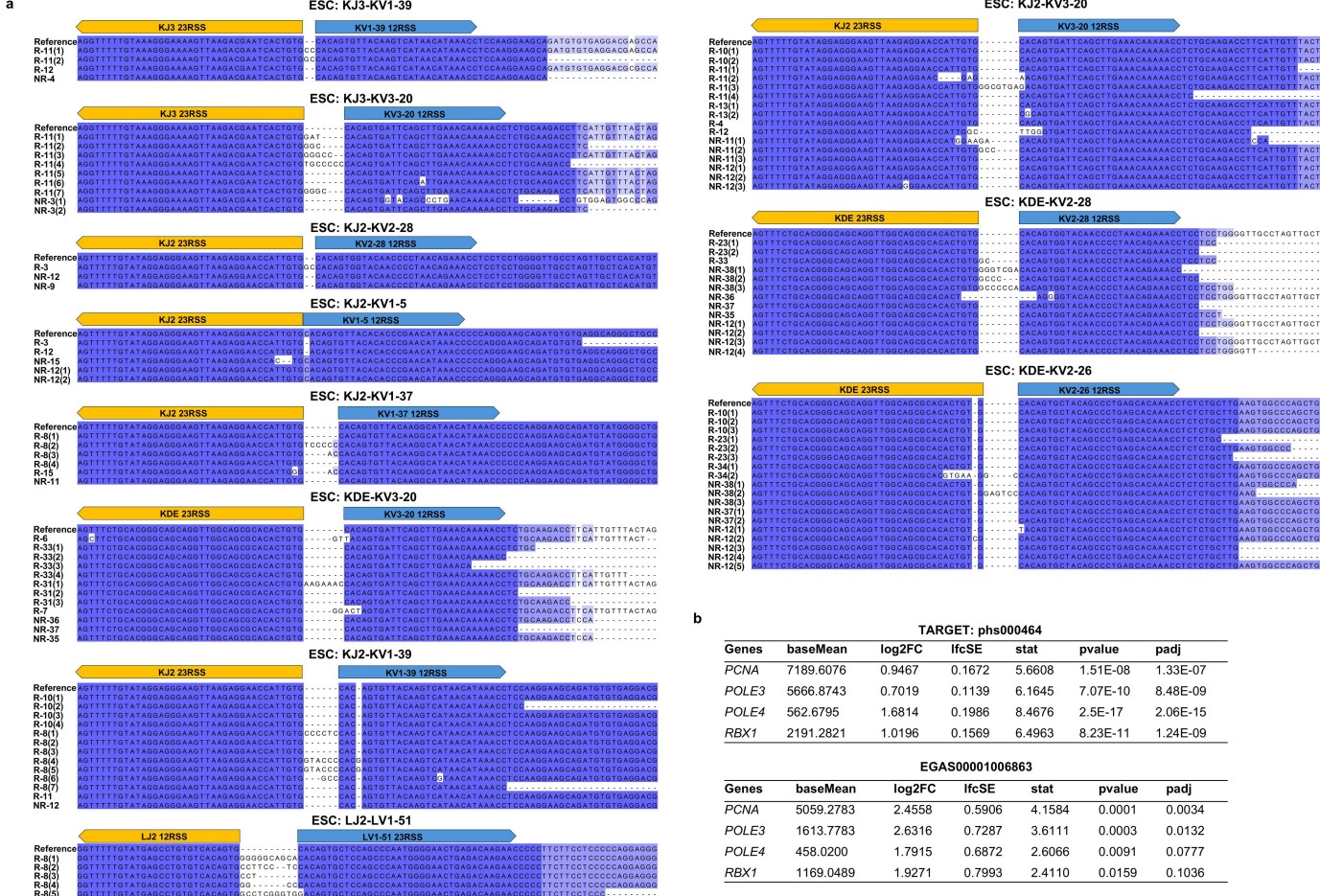

**b**

**TARGET: phs000464**

| Genes | baseMean | log2FC | lfcSE | stat | pvalue | padj |
|---|---|---|---|---|---|---|
| *PCNA* | 7189.6076 | 0.9467 | 0.1672 | 5.6608 | 1.51E-08 | 1.33E-07 |
| *POLE3* | 5666.8743 | 0.7019 | 0.1139 | 6.1645 | 7.07E-10 | 8.48E-09 |
| *POLE4* | 562.6795 | 1.6814 | 0.1986 | 8.4676 | 2.5E-17 | 2.06E-15 |
| *RBX1* | 2191.2821 | 1.0196 | 0.1569 | 6.4963 | 8.23E-11 | 1.24E-09 |

**EGAS00001006863**

| Genes | baseMean | log2FC | lfcSE | stat | pvalue | padj |
|---|---|---|---|---|---|---|
| *PCNA* | 5059.2783 | 2.4558 | 0.5906 | 4.1584 | 0.0001 | 0.0034 |
| *POLE3* | 1613.7783 | 2.6316 | 0.7287 | 3.6111 | 0.0003 | 0.0132 |
| *POLE4* | 458.0200 | 1.7915 | 0.6872 | 2.6066 | 0.0091 | 0.0777 |
| *RBX1* | 1169.0489 | 1.9271 | 0.7993 | 2.4110 | 0.0159 | 0.1036 |

**Extended Data Fig. 5 | ESC sequences are identical regardless of subsequent relapse. a**, Alignment of LAM-ESC reads for the SJs examined in Fig. 3c. The schematic above each plot shows the position of the 12- and 23-RSS; the reference (expected) SJ sequence is shown at the top of each alignment. LAM-ESC sequences from the patients indicated to the left, are aligned and confirm that the RSSs are from identical recombination events. Differences in linear amplification during LAM-ESC results in variability in sequence length.

Processing of the SJ junction is observed for some SJs, potentially due to SJ reopening by RAGs. These small changes are negligible within 20 kb to 1 Mb of the whole ESC. **b**, Differential gene expression of *PCNA*, *RBX1*, *POLE3* and *POLE4* determined by DESeq2 for patients who later relapse with high SJ levels (*n* = 4) and with low SJ levels (*n* = 3) where the SJ levels were determined in Fig. 2a. Significance of the difference determined by a (two-sided) Wald test with the Benjamini-Hochberg correction for multiple testing (padj).

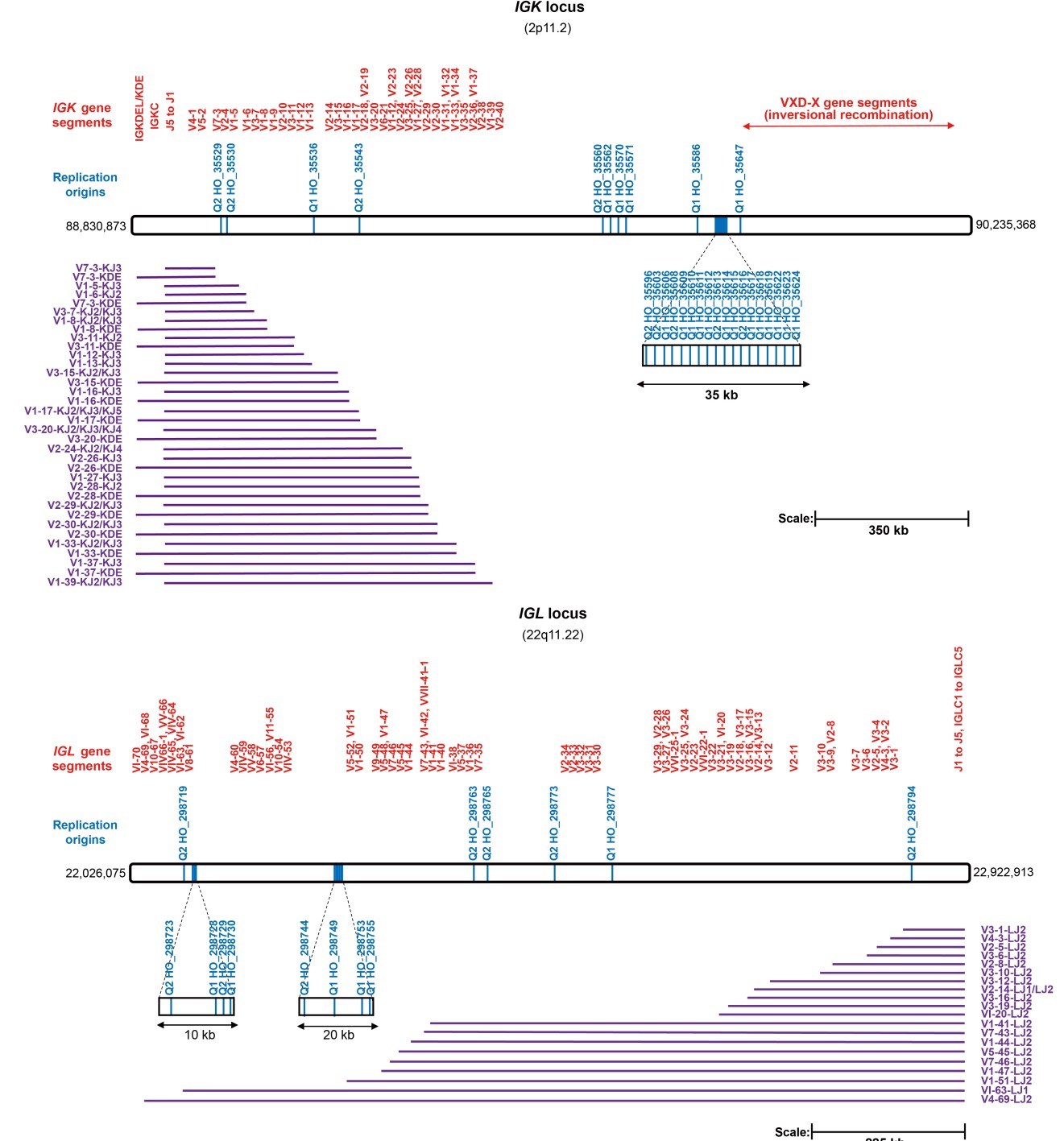

**Extended Data Fig. 6 | Eukaryotic replication origins map to ESC sequences.** Mapping of proposed eukaryotic replication origins and ESC sequences to the *IGK* (upper) and *IGL* (lower) loci. *n* = 35 SJs (*IGK*) and *n* = 20 SJs (*IGL*). Maps of the *IGK* and *IGL* loci are shown with key V and J regions indicated in red. Human replication origins were previously mapped by short nascent strand isolation coupled with next generation sequencing[36]. The activity of these origins was split into 10 quantiles according to mean origin activity where quantile 1 (Q1) represents the top 10% activity. Only Q1 and Q2 replication origins (defined as core replication origins) are shown (light blue) as these account for the majority (70–85%) of replication initiation events. ESCs that were observed at least once above the threshold level found in healthy blood are mapped (in purple) as these ESCs are most likely to have replicated. ESCs below the threshold could have resulted from recent recombination events or have persisted from much earlier recombination events and therefore their level of replication is more difficult to predict.

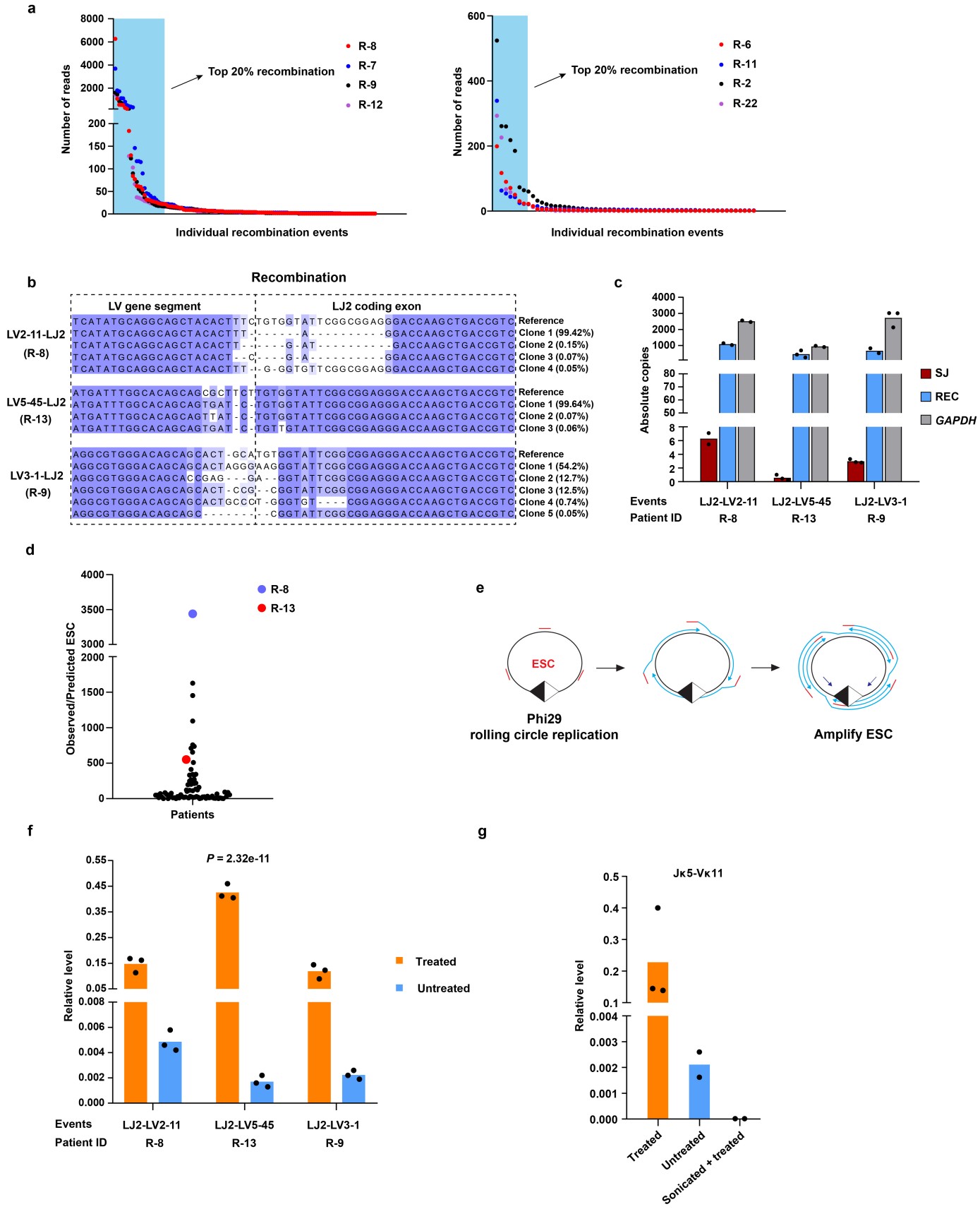

**Extended Data Fig. 7** | See next page for caption.

**Extended Data Fig. 7 | Clonotype analyses imply ESCs persist from early recombination events. a**, To identify major recombination events, all recombination events in a patient (Supplementary Table 2) were ranked by decreasing number of reads and plotted against the individual events. This reveals a point in the distribution of sequencing reads where levels begin increasing rapidly. To define this point, we followed the methodology described[37]. We then found the $x$-axis point for which a line with a slope of -1 was tangent to the curve. We define recombination events above this point to be major recombination events, and those below, minor recombination events. Two sets of plots are shown: Left: Data from four patients where high numbers (left) or lower numbers (right) of sequencing reads were detected. In each case, there is a clear point where the sequencing reads begin to increase rapidly. **b**, SJs that correspond to the primary recombination event (i.e. most sequencing reads) were identified in three patients (R-8 – DUX4-r; R-13 – *ETV6::RUNX1*+ and R-9 – DUX4-r). Here, an identical ESC can be generated only if the second allele undergoes the same recombination event. The extensive addition and deletion of bases upon formation of VJ coding junctions[2,3] means that the chances of two independent recombination events generating exactly the same coding junction are negligible. To determine the number of clonotypes for each SJ, the corresponding recombination junctions were amplified and subjected to Amplicon sequencing. The levels of the distinguishable clonotypes are shown to the right of the respective junction sequences. For LV2-11-LJ2, and LV5-45-LJ2, 99.42% and 99.64%, respectively of coding junctions have the same sequence with only 2-3 distinguishable minor clonotypes. The dominance of a single clone in each case suggests it is very likely that each ESC persisted from the major recombination event. **c**, To test if SJs corresponding to the primary recombination event could have arisen from minor clonotypes, total SJ copies, total recombination junctions and *GAPDH* (genomic DNA) were measured via ddPCR and plotted. The number of recombination copies attributable to each clonotype was calculated by multiplying the absolute copy number of the recombination event by the proportion of each clonotype. From this, the levels of minor clonotypes equate to those expected from 0.81-3.69 relative cell divisions following recombination (see Methods). Mean values are plotted. $n$ = 2 or 3 technical repeats for three patients. **d**, Comparison of the measured SJ copies (b) to those observed for other SJs generated a similar number of cell divisions ago (<6 relative cell divisions – Fig. 3c; $n$ = 77): the observed LJ2-LV2-11 SJ copies are substantially higher than those found for all other recently generated SJs (see Methods), implying that at least some of the observed LJ2-LV2-11 SJs persist from the primary recombination event. **e**, Control experiments to test if ESCs remain circular. Schematic of ESC amplification via rolling circle amplification using Phi29 polymerase. **f**, Samples from patients with known SJs were amplified with Phi29 polymerase whereas the enzyme was omitted from control (untreated) samples. The levels of the SJs shown were determined by ddPCR in treated (amplified; orange bars) and control samples (blue bars). Following normalisation to *GAPDH*, the significance of the increase of the Phi29-treated samples compared to the untreated samples was determined by a two-way Anova test. Mean values are plotted. Black dots indicate technical repeats. $n$ = 3 patients. **g**, Verification that Phi29 amplifies circular DNA. Mouse IgG+ samples with known circular ESCs (Fig. 1d) were treated as in f. Levels of treated (orange bars) and untreated products (blue bars) are shown, following normalisation to *Gapdh* and as determined by ddPCR. Black dots indicate technical repeats. Mean values are plotted. $n$ = 3 IgG+ samples. Brief sonication of DNA prior to Phi29 treatment eliminated ddPCR amplification of the SJ but *Gapdh* amplification was not significantly altered. Amplification of circular DNA by Phi29 implies that SJ persistence cannot be explained by their reintegration.

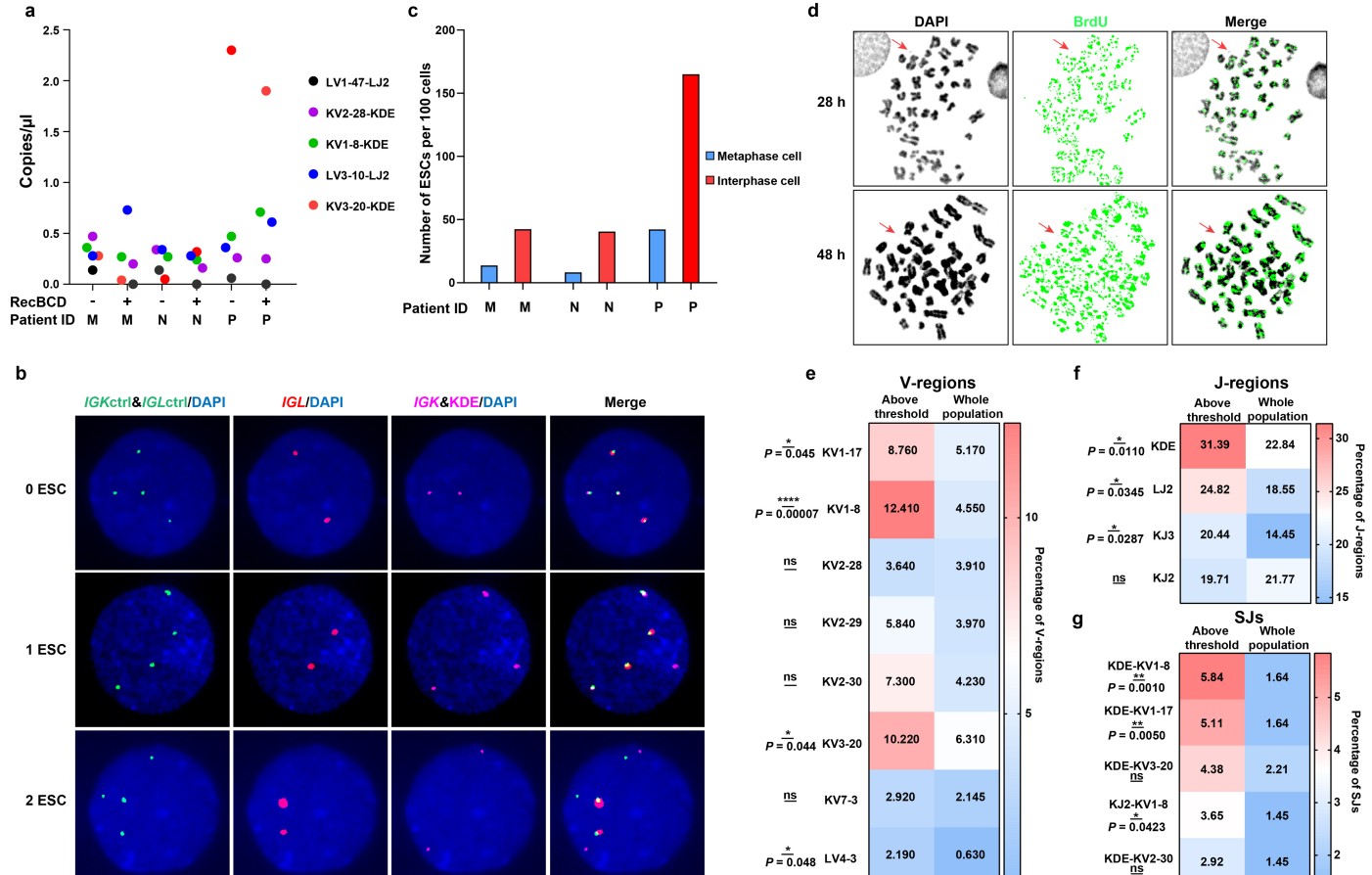

**Extended Data Fig. 8 | ESCs replicate and are non-chromosomal. a**, ddPCR measurements of the levels of five frequently observed SJs, normalised to *GAPDH*, in patients M, N and P, without (-) or with (+) prior RecBCD treatment of the DNA. Individual SJs are shown by distinct colours. Mean values of three or more technical repeats are shown. *n* = 3 patients. The median of normalised, elevated SJ levels in patients who later relapse is 4 copies/μl (Extended Data Fig. 4b); the high copy SJ detected for patient P (normalised value: 2.3 copies/μl) is therefore at the lower end of elevated SJ levels. **b**, Representative interphase FISH images. Control probes against *IGK* and *IGL* are in green; the regions that are excised to generate ESCs are in red (*IGL*) or magenta (*IGK*). Upper: Cell where neither *IGK* nor *IGL* ESCs are present. *n* = 200. The overlapping *IGL* control and ESC signals indicate no *IGL* recombination. The overlapping *IGK* control and ESC signals implies no recombination to KDE but one or both of the *IGK* loci may have recombined to the JK RSSs, with loss of the resulting ESC. Middle: Cell with one ESC. *n* = 99. Lower: Cell with two ESCs. *n* = 27. Controls for FISH probe hybridisation to metaphase chromosome spreads are given in Supplementary Fig. 4. **c**, Total non-chromosomal DNAs detected in DAPI stained metaphase chromosome spreads versus interphase FISH per 100 cells for patients with ESC levels similar to those who later relapse (patient P; *n* = 26 metaphase & 108 interphase images) and those who remain in remission (patients M and N; *n* = 29 & 127 and 24 & 101 images, respectively). Fewer ESCs are detected by DAPI staining compared to interphase FISH due to losses and/ or inefficient DAPI staining of small circular DNAs. **d**, Metaphase chromosomes from cells grown with BrdU for 28 or 48 h, followed by hybridisation of the chromosomes to anti-BrdU antibodies. The labelled non-chromosomal DNA is indicted by a red arrow. Images were deconvoluted using CellSens Dimension software. Magnification of all images is 60x. **e, f, g**, The percentage of times the **(e)** V gene segment **(f)** J gene segment or **(g)** SJ shown is found in the population of SJs above the threshold level in healthy blood (Fig. 2a) compared to the percentage of times it is found in the whole population (*n* = 1585 total SJs from 71 patients). The significance of the difference was calculated by one-sided hypergeometric distribution. *P* = 0.045 (IGKV1-17), 0.00007 (IGKV1-8), 0.044 (IGKV3-20), 0.048 (LV4-3), 0.0110 (KDE), 0.0345 (IGLJ2), 0.0287 (IGKJ3), 0.0010 (IGKV1-8-KDE), 0.0050 (IGKV1-17-KDE), and 0.0423 (IGKV1-8-KJ2).

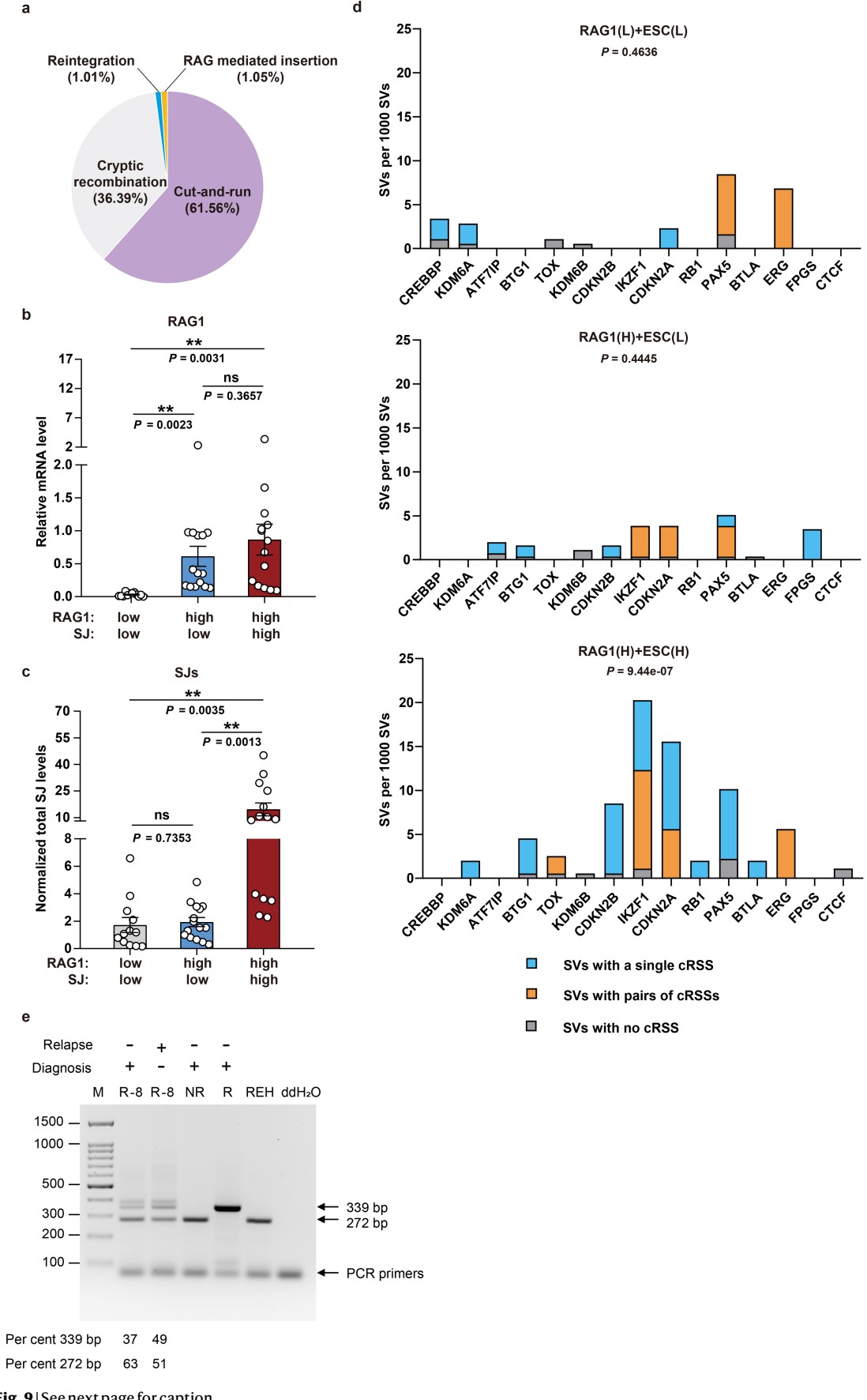

**Extended Data Fig. 9 |** See next page for caption.

**Extended Data Fig. 9 | Increased mutations at relapse-associated genes correlate with high ESC levels. a**, Characterisation of SVs with cRSS(s) at the breakpoint(s). Pie chart showing fraction of SVs that have two cRSSs at the breakpoints (consistent with cryptic recombination[19,47]) or one cRSS at the breakpoint, consistent with cut-and-run[18], reintegration[14–16] or RAG-mediated insertion[49]. Further breakpoint characteristics at single cRSSs (cut-and-run, reintegration or RAG-mediated insertion) were determined using a custom programme available from https://github.com/Boyes-Lab/Structural-Variants[62]. SVs from the TARGET study. $n$ = 150 patients. **b**, *RAG1* expression levels determined by RT-qPCR for BCP-ALL patients shown in Fig. 2a. Three patient groups were identified according to *RAG1* expression and total normalised SJ reads: (i) High *RAG1*, high SJs ($n$ = 14); (ii) High *RAG1*, low SJs ($n$ = 15) and (iii) low *RAG1*, low SJs ($n$ = 12). *RAG* expression data are normalised to *HPRT* expression levels. Data are presented as mean values ± s.d. The significance of the difference in expression levels was determined by a two-tailed Student's t-test. 95% CI: 0.3122 to 1.360 (high *RAG1* expression, high SJs versus low *RAG1* expression, low SJs); 0.2294 to 0.9355 (high *RAG1* expression, low SJs versus low *RAG1* expression, low SJs) **c**, Total normalised SJ levels for the patient groups given in (**b**). Data are presented as mean values ± s.d. The significance of the differences was determined by a two-tailed Student's t-test. 95% CI: 4.675 to 21.12 (high *RAG1* expression, high SJs versus low *RAG1* expression, low SJs); 5.431 to 19.94 (high *RAG1* expression, high SJs versus high *RAG1* expression, low SJs). **d**, SVs at genes that are commonly mutated in BCP-ALL or at relapse with a cRSS on one side of the breakpoint (blue), both sides (orange) or no cRSS (grey) for the patient groups given in (**b**) using WGS or WES compared to the corresponding germline sequence. The significance of increased SVs involving single cRSSs at relapse-associated or frequently mutated genes compared to all SVs involving a single cRSS was calculated using a one-sided hypergeometric distribution; $P$ = 9.44×10$^{-7}$ for the high *RAG1*, high SJ group (bottom panel). **e**, PCR amplification of the *SYK* gene using template DNA from patient samples taken at diagnosis or at relapse (indicated above the gel) as well as the BCP-ALL cell line, REH. R and NR indicate samples taken from patients who did, and who did not, later relapse. ddH$_2$O indicates no template control. A 67 bp insertion in one allele of the *SYK* gene in patient R-8 increases between diagnosis and relapse. Quantification of the 272 and 339 bp bands is shown beneath the gel. For source gel data, see Supplementary Fig. 2b.

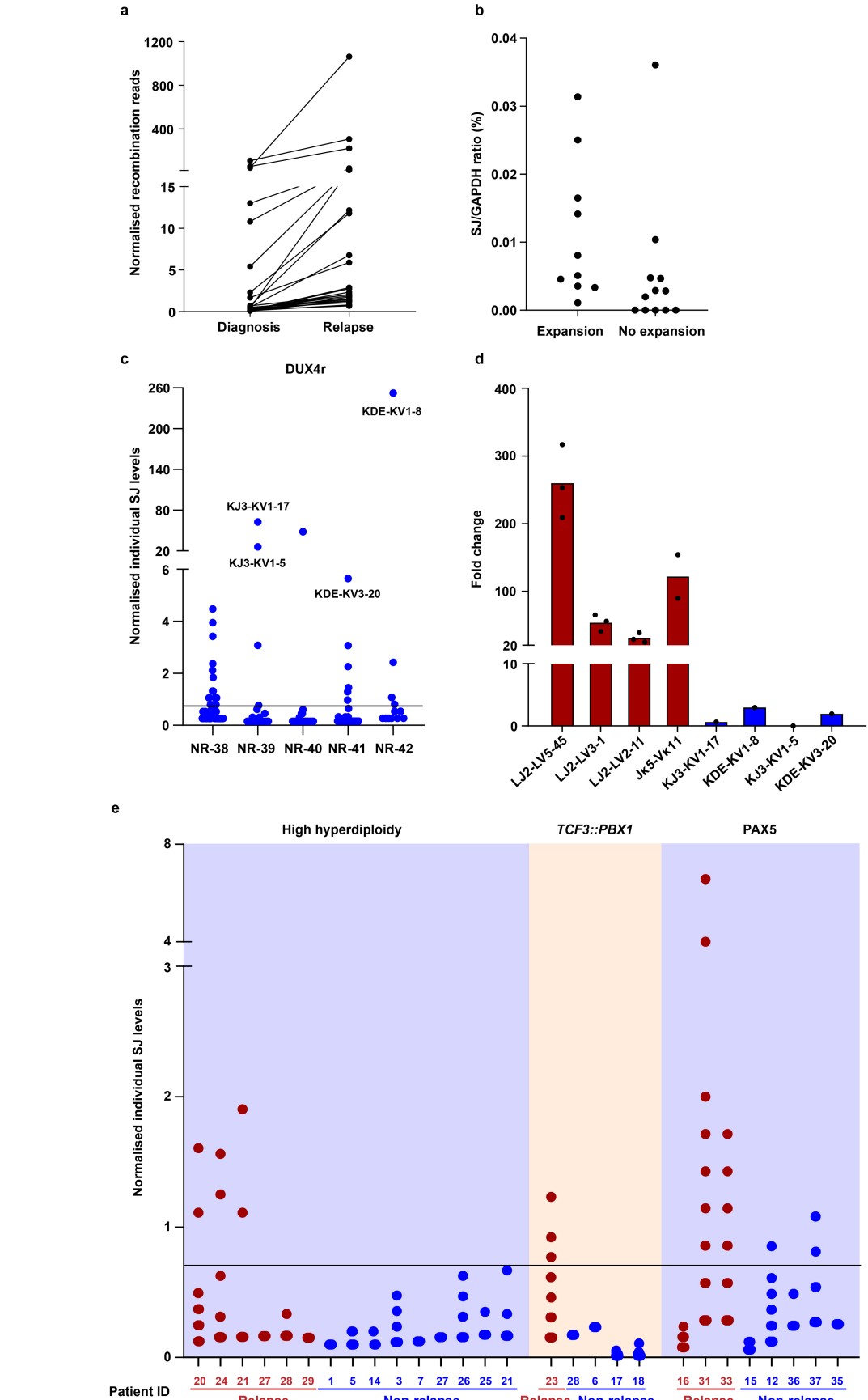

**Extended Data Fig. 10** | See next page for caption.

**Extended Data Fig. 10 | ESCs are associated with clonal expansion between diagnosis and relapse. a**, Changes in normalised LAM-recombination sequencing reads for individual recombination junctions between diagnosis and relapse. $n = 27$. For clarity, only junctions where the levels have increased at relapse are plotted. The changes in all junctions where LAM-recombination reads are detected at both diagnosis and relapse are given in Supplementary Table 2, final tab. **b**, SJ levels at diagnosis when the corresponding recombination junction is increased at relapse (Expansion) or shows no increase between diagnosis and relapse (No expansion). The SJ/GAPDH ratio is shown, using levels determined by ddPCR. $n = 10$ ESCs (Expansion) and $n = 12$ (No expansion), from eight patients. A range of ESC levels correspond to expansion of cells with the corresponding recombination junction. This may be because ESCs can trigger mutations at different times prior to diagnosis and therefore some of the ESCs that triggered damaging mutations may have been diluted; likewise, ESCs from other recombination events may contribute to disease-causing mutations and thus expansion may be influenced by a combination of ESCs. **c**, Total SJ levels plotted for five DUX4-r patients who did not subsequently relapse. The exceptionally high SJ reads in these patients is anomalous and similar to the levels for SJs that are integrated in the genome following inversional recombination events. In three cases, NR-39, NR-40 and NR-42, SJs involving the same sequences, KV1-17/KV1-5 or KV1-8 are present that are 10- to 60-fold higher than the SJs typically observed in patients who later relapse. LAM-ESC reads for KV1-17 and KV1-5 SJs are identical. KV1-8 differs from KV1-17/KV1-5 by just 2 bp. All sequences differ from their corresponding inversional recombination event (KVD1-17/1-5 or KVD1-8) by just a single bp, suggesting that these SJs may have resulted from inversional events. **d**, ddPCR analysis of samples that had, or had not, been treated with Phi29 to amplify circular DNA. The mean fold-change in PCR product is plotted following treatment with Phi29 compared to untreated sample. The plots to the left (in red; $n = 4$ SJs) show fold-change for SJs that appear to be present on ESCs; the plots to the right (in blue; $n = 4$ SJs) show the fold-change for SJ sequences from DUX4-r patients. The low amplification is consistent with these SJs being integrated into the genome. The LAM-ESC data for these five patients were therefore omitted from Fig. 2a. It was not possible to obtain a ddPCR amplification product for the high copy SJ from patient NR-40. **e**, Total SJs in BCP-ALL samples at diagnosis were replotted according to the BCP-ALL subtype: High hyperdiploidy (left; $n = 15$), *TCF3::PBX1* (middle; $n = 5$) and *PAX5* altered/mutated (right; $n = 8$). For each BCP-ALL subtype, the normalised number of copies of each SJ for patients who later relapse (red) and those who remain in remission (blue) is plotted. A threshold is set at the value equivalent to the highest normalised SJ level detected in healthy blood (horizontal line).

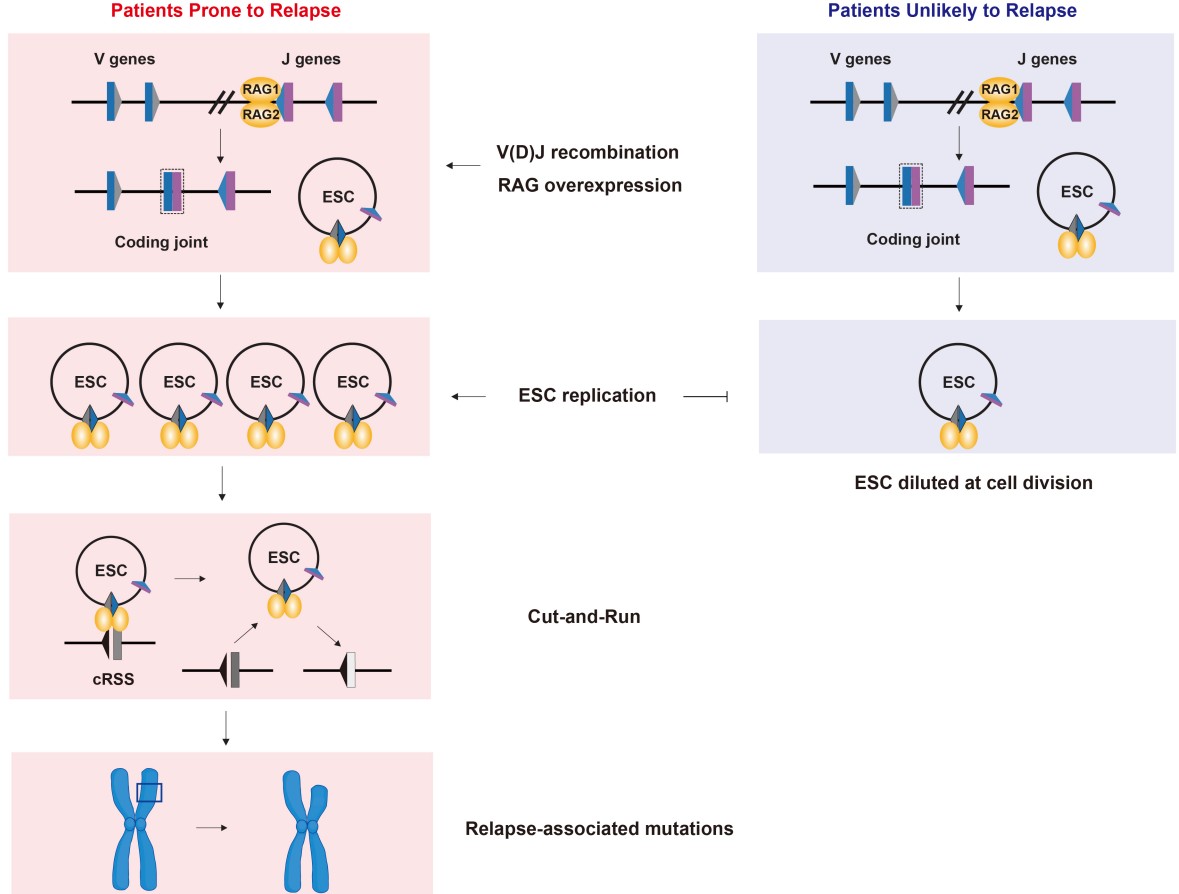

**Extended Data Fig. 11 | Model outlining the impact of increased ESC levels on BCP-ALL progression.** In patients prone to relapse (left), increased *RAG1* expression will lead to increased secondary recombination and the generation of new ESCs. This, combined with cell-intrinsic factors that cause increased ESC replication, results in higher ESC levels. These ESCs can combine with RAG proteins to trigger DSBs at cRSSs in the genome via the cut-and-run reaction. Ongoing mutations, including those at frequently mutated and relapse-associated genes, lead to an increased mutational burden that is inherited by all daughter cells, promoting treatment resistance and relapse. In patients who are unlikely to relapse (right), although new ESCs are generated, the absence of significant ESC replication results in a lower mutational burden and a decreased likelihood of treatment resistance.

# Reporting Summary

## Statistics

For all statistical analyses, confirm that the following items are present in the figure legend, table legend, main text, or Methods section.

| n/a | Confirmed | |
|---|---|---|
| ☐ | ☒ | The exact sample size (*n*) for each experimental group/condition, given as a discrete number and unit of measurement |
| ☐ | ☒ | A statement on whether measurements were taken from distinct samples or whether the same sample was measured repeatedly |
| ☐ | ☒ | The statistical test(s) used AND whether they are one- or two-sided *Only common tests should be described solely by name; describe more complex techniques in the Methods section.* |
| ☒ | ☐ | A description of all covariates tested |
| ☐ | ☒ | A description of any assumptions or corrections, such as tests of normality and adjustment for multiple comparisons |
| ☐ | ☒ | A full description of the statistical parameters including central tendency (e.g. means) or other basic estimates (e.g. regression coefficient) AND variation (e.g. standard deviation) or associated estimates of uncertainty (e.g. confidence intervals) |
| ☐ | ☒ | For null hypothesis testing, the test statistic (e.g. *F*, *t*, *r*) with confidence intervals, effect sizes, degrees of freedom and *P* value noted *Give P values as exact values whenever suitable.* |
| ☒ | ☐ | For Bayesian analysis, information on the choice of priors and Markov chain Monte Carlo settings |
| ☒ | ☐ | For hierarchical and complex designs, identification of the appropriate level for tests and full reporting of outcomes |
| ☐ | ☒ | Estimates of effect sizes (e.g. Cohen's *d*, Pearson's *r*), indicating how they were calculated |

*Our web collection on statistics for biologists contains articles on many of the points above.*

## Software and code

Policy information about availability of computer code

| Data collection | FISH images were acquired using Olympus CellSens Dimension 3.2 (Build 23706) software<br>Flow cytometry data were collected using BDFACSChorus 3.0, version 1.4.3.0<br>ddPCR data was acquired and analysed using the Bio-Rad QuantaSoft v1.7.4 software<br>qPCR data was acquired and analysed using Corbett Rotor-Gene 6000 Series Software (v.1.7, build 87). |
|---|---|
| Data analysis | A custom Python script to analyse SJs in WGS datasets (Extended Data Fig. 2e) is available at (https://github.com/Boyes-Lab/NGS-Analysis; DOI: 10.5281/zenodo.15412502).<br>A custom Python script to analyse LAM-ESC and LAM-recombination data (Fig. 2 and others), together with a custom database is available at https://github.com/Boyes-Lab/LAM-ESC-Recombination; DOI: 10.5281/zenodo.15412315. The script automates BLAST searches against a custom BLAST database, consisting of all V-J recombination events or all head-to-head RSS combinations from the immunoglobulin kappa and lambda loci, for recombination and SJ libraries, respectively.<br>Differentially expressed genes were identified using DESeq2 with \|logFC\| > 0.585 and FDR < 0.05. Gene set enrichment analysis (GSEA) was carried out according to the user guide provided by the BROAD Institute (https://docs.gsea-msigdb.org/#GSEA/GSEA_User_Guide/).<br>In silico experiments to detect ecDNAs were carried out using AmpliconArchitect according to the AmpliconSuite pipeline (https://github.com/AmpliconSuite/AmpliconSuite-pipeline/blob/master/documentation/GUIDE.md).<br>A custom Python script was used to determine the clonotypes present in sequencing reads compared to the reference motifs near the breakpoint junction of interest (Extended Data Fig. 7b). Clonotype_analysis.py, is available at: (https://github.com/Boyes-Lab/NGS-Analysis; DOI: 10.5281/zenodo.15412502).<br>A bespoke Python programme to analyse breakpoints of SVs, SVs_near_RSSs.py, (Fig. 5a, b) is available at (https://github.com/Boyes-Lab/Structural-Variants; DOI: 10.5281/zenodo.15412565). This creates an analysis window spanning 50 bp either side of each breakpoint; the |

presence of an RSS within the window is then analysed using the DNAGrep algorithm, via RSSite.
A custom Python script to analyse potential cut-and-run events compared to reintegration events is provided at https://github.com/Boyes-Lab/Structural-Variants; DOI: 10.5281/zenodo.15412565.
Python version 3.10 was used for all Python scripts.
FISH images were analysed using FIJI 2.16.0 software.
Agarose gels were quantified using FIJI Image J 2, version 2.14.0/1.5f

For manuscripts utilizing custom algorithms or software that are central to the research but not yet described in published literature, software must be made available to editors and reviewers. We strongly encourage code deposition in a community repository (e.g. GitHub). See the Nature Portfolio guidelines for submitting code & software for further information.

# Data

Policy information about availability of data

All manuscripts must include a data availability statement. This statement should provide the following information, where applicable:

- Accession codes, unique identifiers, or web links for publicly available datasets
- A description of any restrictions on data availability
- For clinical datasets or third party data, please ensure that the statement adheres to our policy

WGS datasets from 61 ETV6::RUNX1+ BCP-ALL patients were downloaded from the European Genome-phenome Archive (EGA), dataset ID: EGAD00001000116). The human genome sequence hg19, (GCA_000001405.14) release GRCh37.p13, was downloaded from https://hgdownload.soe.ucsc.edu/goldenPath/hg19/bigZips/. Structural variant data of BCP-ALL patients at diagnosis and relapse were downloaded from the Complete Genomics (CGI, from within the TARGET database: dbGaP Sub-study ID: phs000464). RNA-seq data of BCP-ALL patients at diagnosis were downloaded from the TARGET database (dbGaP Sub-study ID: phs000464). RNA-seq and WGS datasets from BCP-ALL patients in the VIVO Biobank cohort were downloaded from EGA under the Accession Code EGAS00001006863. Raw LAM-ESC and LAM-recombination sequences are available from the European Genome-phenome Archive (EGA), under the dataset ID: EGAD50000000597. The extracted recombination junctions and ESCs are given in Supplementary Tables 2 and 3. Whole exome sequencing data from BCP-ALL patients in the VIVO Biobank cohort are available from EGA via the dataset ID: EGAD50000001519. Amplicon sequencing data of the recombination junctions used for clonotype analysis are available from EGA under the dataset ID: EGAD50000001518. Source data for all graphs is available as three Excel spreadsheets for main, Extended Data and Supplementary Figures, respectively. FISH data are available via Research Data Leeds63: https://doi.org/10.5518/1693. The sample cohort used for each Figure is given in Supplementary Table 6.

# Research involving human participants, their data, or biological material

Policy information about studies with human participants or human data. See also policy information about sex, gender (identity/presentation), and sexual orientation and race, ethnicity and racism.

| Reporting on sex and gender | Sex- or gender-based analyses were not performed as the incidence of BCP-ALL is known to be similar between males and females (1:1.2). The study was initially performed blind. Information provided following data acquisition showed that very similar numbers of samples from males and females were used across both patient groups (i.e. patients known to subsequently relapse and those who remained in remission). |
|---|---|
| Reporting on race, ethnicity, or other socially relevant groupings | No socially constructed or socially relevant categorisation variables were used. Although there is a higher incidence of BCP-ALL in the upper hemisphere, the centres from which the samples were received were all from the upper hemisphere (UK and Czech Republic). Therefore, samples used in the study should not be subject to variables between higher and lower disease incidence areas. |
| Population characteristics | The human research participants are all under age 25 since the peak age group for BCP-ALL is children and young adults. WGS and RNA-seq data are also available for BCP-ALL patients in this age group, via EGA (dataset IDs: EGAD00001000116 and EGAS00001006863) and the TARGET dataset (NIH), allowing comparison of our data with big datasets. Relevant genotypic information regarding BCP-ALL subtype of the human participants is given in Supplementary Table 4. All human participants had been diagnosed with ALL, apart from normal blood samples from two control participants. Information on whether the participants with BCP-ALL suffered relapse disease is given in Supplementary Figure 4. |
| Recruitment | Patients were recruited to donate samples via leaflets/information provided to them by their clinicians. Donations to VIVO Biobank were from throughout the UK, typically from larger centres participating in ALL trials. Samples were donated to HMDS from throughout Northern England whereas those in the Czech Republic were from patients in one of the larger ALL treatment centres. Informed consent was obtained from the parents or legal guardians of children with ALL, eliminating self-selection bias. |
| Ethics oversight | Patient samples, taken as part of routine diagnostics, were supplied by VIVO Biobank, HMDS or a hospital in the Czech Republic. Collection and use of patient samples were approved by the appropriate institutional review board (IRB). Each organisation obtained informed patient consent for anonymised samples to be used by third parties for research. The use of surplus diagnostic material for research by HMDS and collaborators was approved by the Health Research Authority (HRA): 04_Q1205_125. Local ethics approval was obtained from the Biological Sciences Research Ethics Committee, University of Leeds: BIOSCI 18-031, 2308 and CCR 2285, Royal Marsden Hospital NHS Foundation Trust. |

Note that full information on the approval of the study protocol must also be provided in the manuscript.

# Field-specific reporting

Please select the one below that is the best fit for your research. If you are not sure, read the appropriate sections before making your selection.

☒ Life sciences ☐ Behavioural & social sciences ☐ Ecological, evolutionary & environmental sciences

For a reference copy of the document with all sections, see nature.com/documents/nr-reporting-summary-flat.pdf

# Life sciences study design

All studies must disclose on these points even when the disclosure is negative.

| Sample size | An initial power calculation was performed, based on preliminary data, to determine the number of samples that should be analysed to determine if the link between higher ESC levels and subsequent relapse, is statistically significant. Based on this, one hundred samples were requested from VIVO Biobank. Although sample quality precluded analysis of some samples, as many samples as feasible were analysed. Statistical tests were applied to the data to determine if sufficient samples had been tested. |
|---|---|
| Data exclusions | For LAM-ESC data, signal joints (SJs) involving KV gene segments that are known to undergo inversional recombination were excluded from further analyses. This is because inversional recombination results in SJs that are retained in the genome and therefore such SJs do not represent extra-chromosomal (ESC) DNA. This is stated in the legend to Supplementary Table 3 that shows the LAM-ESC data. |
| Replication | Reproducibility was confirmed by performing three technical repeats for each experiment. Three biological repeats were performed with mouse samples where this number of biological repeats is possible. For patient samples, reproducibility was confirmed using samples from at least three patients and often many more, depending on the number of samples that fitted the criteria of the experiment (such as ESCs at a given level or that appeared to persist from the primary recombination event). All attempts at replication were successful. |
| Randomization | Samples from BCP-ALL patients, taken at diagnosis, were provided with information about whether the patient subsequently relapsed. Otherwise, analyses were initially blinded. Experiments were performed in exactly the same way for all samples; data were analysed by comparing patients who later relapsed with those who did not. Further analyses of covariates once additional information was made available showed roughly similar numbers of males and females in each group and representatives of different BCP-ALL subtypes in each group. A large sample size was used to reduce the impact of other variables. |
| Blinding | Anonymised patient samples were blinded for all variables apart from whether or not the patient subsequently relapsed. |

# Reporting for specific materials, systems and methods

We require information from authors about some types of materials, experimental systems and methods used in many studies. Here, indicate whether each material, system or method listed is relevant to your study. If you are not sure if a list item applies to your research, read the appropriate section before selecting a response.

## Materials & experimental systems

| n/a | Involved in the study |
|---|---|
| ☐ | ☒ Antibodies |
| ☐ | ☒ Eukaryotic cell lines |
| ☒ | ☐ Palaeontology and archaeology |
| ☐ | ☒ Animals and other organisms |
| ☒ | ☐ Clinical data |
| ☒ | ☐ Dual use research of concern |
| ☒ | ☐ Plants |

## Methods

| n/a | Involved in the study |
|---|---|
| ☒ | ☐ ChIP-seq |
| ☐ | ☒ Flow cytometry |
| ☒ | ☐ MRI-based neuroimaging |

# Antibodies

| Antibodies used | FITC anti-CD19 (BD Pharmingen, #553785) Clone 1D3<br>PE anti-CD43 (BD Pharmingen, #553271). Clone S7. Lot 7172765<br>FITC anti-IgM (BD Pharmingen, #553408) Clone R6-60.2. Lot 31529<br>PE anti-IgG (eBioscience, #12-4010-82). Polyclonal. Lot 4315025<br>Anti-BrdU (BD Pharmingen #555627). Clone 3D4(RUO). Lot 4015735<br>Goat Anti-Mouse IgG (Jackson ImmunoResearch #115-001-003). Polyclonal. Lot 92114<br>Alexa Fluor 488 donkey anti-goat (ThermoFisher #A32814). Polyclonal. Lot VK308431 |
|---|---|
| Validation | Validation and references are provided on the manufacturer's websites:<br><br>FITC anti-CD19 (BD Pharmingen, #553785) Clone 1D3<br>https://www.bdbiosciences.com/en-us/products/reagents/flow-cytometry-reagents/research-reagents/single-color-antibodies-ruo/fitc-rat-anti-mouse-cd19.553785 |

PE anti-CD43 (BD Pharmingen, #553271). Clone S7.
https://www.bdbiosciences.com/en-eu/products/reagents/flow-cytometry-reagents/research-reagents/single-color-antibodies-ruo/pe-rat-anti-mouse-cd43.553271

FITC anti-IgM (BD Pharmingen, #553408) Clone R6-60.2
https://www.bdbiosciences.com/content/dam/bdb/products/global/reagents/flow-cytometry-reagents/research-reagents/single-color-antibodies-ruo/553xxx/5534xx/553408_base/pdf/553408.pdf

PE anti-IgG (eBioscience, #12-4010-82). Polyclonal
https://www.thermofisher.com/antibody/product/Goat-anti-Mouse-IgG-H-L-Secondary-Antibody-Polyclonal/12-4010

Anti-BrdU (BD Pharmingen #555627). Clone 3D4(RUO).
https://www.bdbiosciences.com/en-gb/products/reagents/flow-cytometry-reagents/research-reagents/single-color-antibodies-ruo/purified-mouse-anti-brdu.555627?tab=product_details

Goat Anti-Mouse IgG (Jackson ImmunoResearch #115-001-003).
https://www.jacksonimmuno.com/catalog/products/115-001-003

Alexa Fluor 488 Donkey anti-goat (ThermoFisher #A32814). Polyclonal.
https://www.thermofisher.com/antibody/product/Donkey-anti-Goat-IgG-H-L-Highly-Cross-Adsorbed-Secondary-Antibody-Polyclonal/A32814

# Eukaryotic cell lines

Policy information about cell lines and Sex and Gender in Research

| Cell line source(s) | hTERT-RPE-1: ATCC. Derived from a diploid human female<br>NIH3T3: A gift from Professor Constanze Bonifer, University of Leeds. Fibroblast cell line derived from male mice.<br>HeLa: A gift from Professor Tariq Enver, ICR, London. Human cervical cancer cell line derived from a female. |
|---|---|
| Authentication | hTERT-RPE-1: Authenticated by ATCC by morphology, STR profiling and karyotyping; Lot number 70043063<br>NIH3T3: Authenticated by amplification with mouse-specific PCR primers<br>HeLa: Authenticated by amplification with human-specific PCR primers. |
| Mycoplasma contamination | hTERT-RPE-1: Tested by ATCC. Mycoplasma negative by Hoechst DNA staining, agar culture and a PCR-based assay. Not subsequently tested for mycoplasma<br>NIH3T3: Confirmed negative for mycoplasma using MycoAlert® Mycoplasma Detection Kit, Catalog No. LT07-318 in 2025<br>HeLa: Confirmed negative for mycoplasma using MycoAlert® Mycoplasma Detection Kit, Catalog No. LT07-318 in 2025 |
| Commonly misidentified lines<br>(See ICLAC register) | HeLa cells have been misidentified as liver cells (Chang liver) where HeLa cells appear to have contaminated the liver cell line. Similarly, HeLa cell contamination has been reported for other cell lines, including HEp-2, KB and J111. In each case, HeLa cells contaminated the other cell lines. Given that we were only using DNA from HeLa cells as a source of non-B cell human DNA and the DNA hybridised as expected to human-specific PCR primers and the cells have the correct morphology, we did not test if another human cell line had been misidentified as HeLa cells. |

# Animals and other research organisms

Policy information about studies involving animals; ARRIVE guidelines recommended for reporting animal research, and Sex and Gender in Research

| Laboratory animals | CBA/C57BL/6J mice between 5 and 7 weeks old were used |
|---|---|
| Wild animals | The study did not involve wild animals |
| Reporting on sex | Pre-B cells were isolated from the femurs of 12 CBA/C57BL/6J mice whereas spleens were isolated from three mice. All mice were 5-7 weeks old. The sex of the mice was not considered in the study design as this is unlikely to influence the B cells used. Instead, approximately equal numbers of male and female mice were used. These had been housed in a full barrier facility where animals are free from common pathogens. |
| Field-collected samples | The study did not involve samples collected from the field |
| Ethics oversight | Animal procedures were performed under Home Office licence P3ED6C7F8, following review by the University of Leeds ethics committee. |

Note that full information on the approval of the study protocol must also be provided in the manuscript.

# Plants

Seed stocks

No seed stocks were used

Novel plant genotypes

Plants were not used in this study

Authentication

Plants were not used in this study

# Flow Cytometry

## Plots

Confirm that:

☒ The axis labels state the marker and fluorochrome used (e.g. CD4-FITC).

☒ The axis scales are clearly visible. Include numbers along axes only for bottom left plot of group (a 'group' is an analysis of identical markers).

☐ All plots are contour plots with outliers or pseudocolor plots.

☐ A numerical value for number of cells or percentage (with statistics) is provided.

## Methodology

Sample preparation

Mouse femurs and spleens were collected from 5 to 7-week-old mice. Bone marrow (BM) cells were flushed from femurs with PBS whereas splenocytes were prepared by flushing cells from finely diced pieces of spleen with PBS through a 50 μm cell strainer. Following preparation of single cell suspensions in PBS, BM cells and splenocytes were centrifuged at 600 x g for 3 minutes and resuspended in 10 ml of 0.168 M NH4Cl to lyse erythrocytes. After 10 minutes, cells were washed with 40 ml PBS and resuspended in 1 ml staining buffer (2% FCS, 1 mM EDTA, 25 mM HEPES-KOH pH 7.9 in PBS).
Cells were stained with the appropriate antibodies prior to purification by flow cytometry. For BM pre-B cells, 1 ml cell suspensions were stained with 6 μl each of FITC anti-CD19 (BD Pharmingen, #553785) and PE anti-CD43 (BD Pharmingen, #553271). BM or spleen IgM+ cells were stained with 10 μl FITC anti-IgM (BD Pharmingen, #553408 - 1 in 100 dilution) whereas spleen IgG+ cells were stained with 15 μl PE anti-IgG (eBioscience, #12-4010-82 - 3 in 200 dilution). Following incubation at room temperature for 10 minutes, cells were washed with PBS and resuspended in 0.5 ml staining buffer

Instrument

Purification was via a FACSMelody (BD) cell sorter

Software

BDFACSChorus software was used to run the flow cytometer and to collect the post-sort purity data

Cell population abundance

Post-sort sample purity was verified by running a sample of sorted cells through the FACSMelody. Only samples with >90% purity were used for further study.

Gating strategy

Gating strategies were as follows:
Bone marrow pre-B cells: (i) Lymphocytes were first gated on forward scatter (FSC-A) and side scatter (SSC-A). (ii) Single lymphocytes were then gated on side scatter (SSC-H and SSC-W). (iii) Single lymphocytes were further gated on forward scatter (FSC-H and FSC-W). (iv) Pre-B cells (CD19+/CD43-) were gated based on their staining with FITC anti-CD19 (CD19 FITC-A) and anti-CD43 (CD43 PE(YG)-A).
Bone marrow / Spleen IgM+ cells. Gating was as above for steps i-iii. (iv) IgM+ cells were gated based on their staining with FITC anti-IgM (IgM FITC-A).
Spleen IgG+ cells: Gating was as above for steps i-iii. (iv) IgG+ cells were gated (IgG + high) based on their staining with PE anti-IgG (IgG PE (YG)-A).

☒ Tick this box to confirm that a figure exemplifying the gating strategy is provided in the Supplementary Information.

