## [Peer Review File · Nature]

Excised DNA Circles from V(D)J Recombination Promote Relapsed Leukaemia

Corresponding Author: Dr Joan Boyes

Version 0:

Reviewer comments:

Referee #1

(Remarks to the Author)

The manuscript by Gao et al describes results indicating that excised signal circles (ESCs) formed in developing lymphocytes during deletional V(D)J recombination events are long-lived and present in mature B-cells. The authors further show through careful quantitative PCR analysis in mouse cells that the ESCs are likely replication-competent, disagreeing with previous reports that ESCs are diluted out during cell division. The authors propose that the replication and asynchronous inheritance of ESCs present a risk to genomic stability and can drive cancer progression. This follows on previous work by this group that ESCs bound by RAG enzymes can cleave at cryptic RSS sequences in the genome through a 'cut-and-run' mechanism. In parallel work with BCP-ALL patient samples, the authors identified increased levels of signal joints (SJs) expected to be in ESCs in patients that have relapsed versus in non-relapsed patients. This manuscript describes highly significant findings that will be of broad interest to several fields of study including lymphocyte development, genomic stability, and cancer studies.

Strengths of the manuscript include the description of a well-performed and systematic study to bolster the claim of ESCs retention and replication in mature B-cells, and the combination of mechanistic studies to correlate ESC levels, RAG transcript levels, and RAG activity at cryptic RSSs with BCP-ALL relapse potential. Multiple methods are used including modifications of LAM-HTGTS, which nicely demonstrate the presence of rare events in patient samples. Although a compelling study, there are some concerns that the conclusions are based on mostly correlational evidence. Major and minor points are listed below.

Major points:

1. The largest concern is that the authors' claim that ESCs 'promote' relapsed leukaemia (in the title of the manuscript) is based largely on correlative results. RAG1 transcript levels are also increased in relapsed BCP-ALL patient samples, and as RAG enzymes can cleave at off-target sites, this complicates the interpretations made in the manuscript. With that, the authors attempted to address this issue by comparing events at single versus paired cRSSs. However, there are several types of RAG-directed off-target events (doi: 10.1038/nature02355; doi: 10.1016/j.cell.2015.10.016; and doi: 10.1016/j.molcel.2016.06.034; among others). This should be addressed in the manuscript, especially the possibility of DNA breaks at non-RSS sites (see lines 283-286). Overall, it is not clear that the levels of ESCs cause the genomic instability or are just a by-product of higher RAG enzyme activity.

2. Deletional and inversional V(D)J recombination occurs in the Igk locus, as was described by the authors. In addition, intra-Vk events have also been identified as a result of overlapping 12/23 signals at V gene segments (doi: 10.1016/j.celrep.2019.11.088). Further, consecutive recombination events may occur at the same locus. These additional outcomes at the Igk locus should be discussed, and it should be clear that SJs chosen to study can only be formed through deletion events.

3. RecBCD was used to reduce linear DNA in the sample, leaving intact ESCs. It was not stated that this approach was taken with the BCP-ALL samples. Instead, it could be interpreted that the presence of ESCs was inferred by the detection of SJs in genomic DNA samples (untreated with RecBCD). This needs clarification. If not, then the BCP-ALL samples should be treated with RecBCD to confirm that the levels of SJs present are in ESCs in the BCP-ALL samples.

4. Increased expression of DNA repair proteins and replication proteins were detected in patients prone to relapse as compared to non-relapsed patients, which is discussed in relation to ESC replication. While a correlation with ESC levels may be present in some samples, it seems to be overstated in the manuscript, since the increased levels of these transcripts would be expected in relapsed samples that would occur with other cancer-driving mutations (independent of ESCs).
5. Appropriate statistical analyses are used in the results shown in the figures, and are well-described in the Methods section of the manuscript.

Minor points:

1. Figures 3B and 5C are not referred to in the text of the manuscript.
2. Line 91 states that ratio of recombination junction to other genomic regions is maintained throughout B-cell development, but the Vk3-Jk5 data for splenic IgM+ cells (extended data Figure 1) is not consistent with this statement.
3. Line 127 refers to Figure 2B, but this figure is not well annotated to match the statement in the text.
4. The model shown in Extended Data Figure 9 focuses exclusively on the Cut-and-Run mechanism. However, the authors could not rule out re-integration events in the manuscript.

(Remarks on code availability)

The web pages could not be accessed with the URLs provided.

Referee #2

(Remarks to the Author)

In this paper, Gao et al., report that a type of extrachromosomal element generated from VDJ recombination, the excised signal circle (ESC), can be retained through series of cell divisions and accumulate at high copies, correlating with worse prognosis in cancer patients. The authors report that abundant signal joint sequences, which indicate ESCs are likely circular in structure, were detected in mouse B cells and in human leukemia samples. The authors postulate that in patient samples, RAG1 expression is not suppressed and hence facilitates RAG/ESC complex formation, thereby contributing to genome instability. This is a fascinating concept which, if true, is important, and potentially of broad interest.

However, some major concerns are raised, which would need to be fully addressed, and if so, would greatly strengthen the paper and confirm its conclusions. The concerns arise in two areas – biological issues that for which conclusions are in need of some further validation and computational issues that could be addressed. As is, some of the major conclusions rest on relatively limited assays and data. Bolstering these claims through additional data, as described below, will be crucial.

Biological issues:

1. All data provided demonstrating the presence of ESCs were derived from sequencing or PCR approaches. Validating their presence by an imaging approach, such as DNA FISH would greatly strengthen the conclusions. Since at least some of them are reported by the authors to be about the same size as ecDNA, detection by DNA FISH should be very much feasible. This would also give a sense on the copies of ESC per cell at single cell level to support the claim that there are high levels of ESCs.
2. Linear DNA digestion should also be performed in human samples to demonstrate circular structures, rather than only in mouse samples.
3. Wouldn't the paper be greatly strengthened if the authors were to examine ESC levels in BCP-ALL cell lines as well?
3. In ETV6::RUNX1+ BCP-ALL samples, based on different methods, SJs were detected in 6/9 samples (67%) by nested PCR (1-3 distinct SJ per patient). By WGS, and in a separate cohort, SJs were detected in 51/61 BCP-ALL samples (83%) (1-27 distinct SJs per patient). In lines 138-140, the authors claimed that ESCs could also be detected in other ALL, however their prevalence is not documented in these BCP-ALL subtypes. Despite ESCs could be frequently detected in ETV6:RUNX1+ BCP-ALL, the same cannot be concluded for other subtypes by simply showing a few cases evident of ESC.
4. The evidence for high copy of ESCs is limited. A lot of the assays rely on PCR, and the authors said it is not feasible to get quantitative data to gauge copy number from WGS. Fig 3a figure legend first states 'the number of copies of each SJ' was plotted, then later writes 'y axis shows the normalized level of each SJ' LAM-ESC. Further, these samples are normalized to ESC level detected in healthy blood, and the authors pointed out earlier in introduction that ESCs are diluted during cell division and therefore are usually expected to gradually fall to negligible levels. I assume this is the case for healthy blood, meaning there are negligibly low levels of ESCs, or even none. In that case, a 30-fold increase would mean at most 30 copies (the limit of the Y-axis) from an unknown number of cells, and that would mean ESCs copy is very low at both bulk and single cell level instead.
5. Line 178-181: "Given that the number of normalized sequencing reads correlates well with absolute ESC levels, as determined by droplet digital PCR (ddPCR, Extended Data Fig. 3a) these data imply that there is increased ESC replication and/or persistence in patients who subsequently relapse." Similar claims were made throughout the text. Higher copy does not necessarily mean increased replication – are the authors trying to say they are amplified? This statement seems to contradict with the data provided in Line 239-242: 'Whilst numerous ESCs that correspond to "major" recombination events

are detected, implying that ESCs persist, no significant difference is detected in the per cent of such ESCs between patients who later relapse and those who do not (25.85% versus 28.3%, respectively).’

6. Lines 134-137: “Since each recombination event generates just a single ESC and there are millions of cells in these malignancies, the ability to detect SJs in these samples suggests that these ESCs have replicated and persisted through many cell divisions.” Is this true to say so? If each cell from those millions of cells each generates one ESC, there could be millions of copies of ESC too, and does not necessarily mean they replicated and conserved in multiple cell divisions. None of the techniques used so far demonstrated ESCs are accumulated in high copies at the single cell level.

7. The authors say RAG1/2 expression is not correlated with ESC copy (Ext Fig 3b), but the data is obtained from health blood samples, where ESCs are not expected to be found. What is the correlation of RAG1/2 expression with ESC copies in BCP-ALL samples then?

8. Line 182-184: ‘and even in samples where RAG1 expression is very similar, high ESC copies are observed in patients who later relapse but not in those who do not (Fig. 3c).’ I don’t think this is appropriate to say based on just 10 samples.

9. In extended Fig 3c, the authors looked at the levels of recombination junction vs ESCs in patient samples. Provided on the y-axis is absolute copy, and the highest value is only 4. Contrasting to this, in ext fig 3d, the y axis is way wider in REC (6000-300 max) than SJ (3.5-18). Does that mean in most cases the copies of REC are even higher than SJ? Can the authors represent the data for each patient as in Fig 4b?

10. Line 205-206: Notably, of the recently generated ESCs, eleven are 206 identical in both patient groups. Where is the supporting data for this statement? Do the authors have sequencing tracks of the PCR products?

11. The authors claimed “high ESC replication coupled to relapse”, however, what is the correlation between the various cell-intrinsic factors from RNA-seq data and ESC itself? The authors highlighted two information: 1) high levels of ESC could be detected in patients who relapsed, and 2) GSEA enrichment from transcriptome of relapse patients (note – these are an independent cohort) identified various cell intrinsic factors. These are merely parallel observations, and it is not possible to draw correlational conclusions between ESCs and the cell intrinsic factors. The data provided in fig 4e is misleading in a way because the levels of ESCs was not shown to be correlated with the expression of these genes. Provided in 4d and e are at most differences between relapse and non-relapse patients. Perhaps the authors could do a comparison between high ESC and low ESC patients instead.

12. No other sample characteristics are examined when the presence of ESCs is linked to relapse. One alternate hypothesis could be that the samples in which ESCs are detected have a higher burden of disease, and therefore a higher tumor “purity” that allows detection of ESCs. In all analyses tumor purity should be investigated as far as possible and its effect investigated.

13. The analysis regarding differences in gene expression in RNA-seq data from 123 patients at diagnosis is speculative given that ESCs were not detected in this cohort. This analysis could be repeated in a cohort in which ESC detection was performed and RNA-seq is available.

14. Throughout the paper the authors could employ more controls e.g. cell lines/tissue samples that would not be expected to have any RAG activity or signal joint DNA present in them.

Computational issues:

1) It appears that in total data from 340 patients may have been used in various analyses in the study. However, the cohorts and exactly which data is used in which experiments in the study is sometimes unclear e.g. “>85 patient samples shows that...”. A figure/cohort diagram in supplementary outlining exactly which data is used for which analyses each of these would have been helpful. In addition, n numbers in the legends would be helpful throughout.

2) The bespoke computational tools created by the group and used to analyze the NGS data are not published. Furthermore, they are not well-described in the main text, have no dedicated methods sections, and are not accessible via the github links placed in the paper which appear to be dead. Without explanation or code, it’s very difficult to assess the veracity of the claims made.

3) There are no mention of sequencing characteristics e.g. coverage for each dataset or what parameters or thresholds were used.

5) It would be helpful if the authors could comment on whether existing computational tools for inferring ecDNA identify circular DNA in hematological malignancies and the reasons for the rate of detection relative to other cancer types. Have others described the presence of excised signal circles, if not, why not? What results do existing tools produce for samples included in this study.

Referee #3

(Remarks to the Author)

In their manuscript entitled “Excised DNA Circles from V(D)J Recombination Promote Relapsed Leukaemia” Zeqian Gao and colleagues present evidence that excision circles (ESC) in lymphocytes can be maintained during cell division and that through their recombination potential may promote genomic instability in leukemias. I am neither a B-cell nor a leukemia expert, but this seems like a very provocative and fascinating hypothesis, which as far as I can tell has the potential to change our current understanding of how ESC contribute to leukemia development.

Here are some suggestions to that may help further strengthen the manuscripts:

Major comments:

1. ESC were studied using PCR and sequencing. In order to validate their extrachromosomal presence, it would be crucial to perform metaphase FISH.
2. ESC replication is not directly shown. The current evidence is not strong. In vitro growth of such cells with nucleotide analogues, eg. BRDU, and measurements of analogue inclusion in replicated ESC would be necessary.
3. PCRs shown in Fig 1 need negative controls. How specifically do the primers amplify the recombination sequence? This is not clear. Ideally, the authors would perform the PCR also in other cell types (not B-cells) to show the negative signal there.
4. If ESC behave like ecDNA, they should randomly segregate to daughter cells during mitosis. That should lead to differences in ESC copy numbers between cells. This again needs to be tested using FISH.
5. LAM-HTGTS should also be performed in cells that do not express RAG1/2 in order to get a sense for the background signal of recombination between the Ig locus and other chromosomes. Some of these recombination events may represent artifacts of amplification with the degenerate primer. In order to exclude this one would need to include negative controls.
6. The RNA sequencing analysis seems very preliminary and vague without any functional support of the link between the differentially expressed genes and the amount of ESC. I would suggest omitting this part in the manuscript, or to perform perturbation experiments to establish/test functional links.
7. If ESC re-integration is causing disease progression/relapse, shouldn't these integrations also clonally expand? Could the authors use their HT-GTS to test this and verify the impact these integrations have on the expression of tumor suppressor genes? Without one example of such tumor suppressor disruption, the association remains purely speculative. At least one example of such recombinations/integrations should be investigated in depth with further validation and functional readouts to support this claim.
8. Did the authors observe any other ecDNA in leukemias using their methods? Is there evidence of RAG1 recombination signals (RSS) at the junctions of these ecDNA? This may be outside the scope of the manuscript, but would further support the importance of RAG1 recombination in leukemia development.

Minor comments:

1. The publication by Koche et al. Nature Genetics 2020 proposed a similar mechanism through which ecDNA could contribute to cancer progression/relapse. They showed that ecDNA can re-integrate into the genome of cancers and that this occurs at sites of tumor suppressor genes and near oncogenes and that this affects the expression of these genes. The authors seem to have missed citing/acknowledging this paper in their introduction/discussion.
2. The method the authors use to isolate the ESC is similar to Circle-seq. The authors should mention this and acknowledge the work done on this method in the past.

(Remarks on code availability)

I am not an expert to review this code.

Version 1:

Reviewer comments:

Referee #1

(Remarks to the Author)

The authors have addressed in full this reviewer's previous concerns with the addition of new data and analyses, along with improved additions to the text. The figures are clearly presented and appropriate statistical analyses are used throughout the manuscript.

(Remarks on code availability)

Code is provided within each of the three repositories that are useful in the analyses as specified in the Code Availability Statement in the manuscript. Clear and detailed instructions are included in the README files.

Referee #2

(Remarks to the Author)

The authors have done an impressive job of revising the manuscript, greatly strengthening the study with extensive new data. It is a highly compelling revision and will be an important contribution to the literature that may open new areas of insight into mechanisms of leukemia development.

Referee #3

(Remarks to the Author)

Thank you for addressing all of my comments in such detail and with well-designed experiments. I applaud the authors for this work, which I find highly interesting, and I look forward to reading it again once it is published.

(Remarks on code availability)

I don't have the computational expertise to review this code.

Dear Dr. Editor,

Thank you very much for your Email and the opportunity to revise our manuscript. We also thank the Reviewers for their insightful and constructive comments which we believe we have now addressed in full, as outlined point by point below. These revisions have substantially improved the manuscript and we would like to thank you for considering this revised version.

Yours sincerely,

Joan Boyes

Referee #1:

The manuscript by Gao et al describes results indicating that excised signal circles (ESCs) formed in developing lymphocytes during deletional V(D)J recombination events are long-lived and present in mature B-cells. The authors further show through careful quantitative PCR analysis in mouse cells that the ESCs are likely replication-competent, disagreeing with previous reports that ESCs are diluted out during cell division. The authors propose that the replication and asynchronous inheritance of ESCs present a risk to genomic stability and can drive cancer progression. This follows on previous work by this group that ESCs bound by RAG enzymes can cleave at cryptic RSS sequences in the genome through a 'cut-and-run' mechanism. In parallel work with BCP-ALL patient samples, the authors identified increased levels of signal joints (SJs) expected to be in ESCs in patients that have relapsed versus in non-relapsed patients. This manuscript describes highly significant findings that will be of broad interest to several fields of study including lymphocyte development, genomic stability, and cancer studies.

Strengths of the manuscript include the description of a well-performed and systematic study to bolster the claim of ESCs retention and replication in mature B-cells, and the combination of mechanistic studies to correlate ESC levels, RAG transcript levels, and RAG activity at cryptic RSSs with BCP-ALL relapse potential. Multiple methods are used including modifications of LAM-HTGTS, which nicely demonstrate the presence of rare events in patient samples. Although a compelling study, there are some concerns that the conclusions are based on mostly correlational evidence. Major and minor points are listed below.

Major points:

1. The largest concern is that the authors' claim that ESCs 'promote' relapsed leukaemia (in the title of the manuscript) is based largely on correlative results. RAG1 transcript levels are also increased in relapsed BCP-ALL patient samples, and as RAG enzymes can cleave at off-target sites, this complicates the interpretations made in the manuscript. With that, the authors attempted to address this issue by comparing events at single versus paired cRSSs. However, there are several types of RAG-directed off-target events ([doi: 10.1038/nature02355](https://doi.org/10.1038/nature02355);

doi: 10.1016/j.cell.2015.10.016; and doi: 10.1016/j.molcel.2016.06.034; among others). This should be addressed in the manuscript, especially the possibility of DNA breaks at non-RSS sites (see lines 283-286). Overall, it is not clear that the levels of ESCs cause the genomic instability or are just a by-product of higher RAG enzyme activity.

We thank the reviewer for this insightful comment. To address it, we firstly examined the relative contribution of the different RAG-directed off-target events to the structural variants (SVs) in the whole genome sequencing (WGS) data from 150 BCP-ALL patients. Consistent with previous studies (doi: 10.1038/ng.2874), nearly 40% of SVs (38.42%) have a cRSS at the breakpoint, that is typical of erroneous RAG-mediated reactions. Of these, we found that ~62% have a cRSS on just one side of the breakpoint, typical of cut-and-run reactions whereas 36% have cRSSs on both sides of the breakpoint, as would be required for cryptic recombination (off-target RAG-mediated recombination). The contribution of the other documented RAG-directed off-target events (ESC reintegration and RAG-mediated insertion [doi: 10.1084/jem.20161638]) is minimal, as determined by our custom programme to detect insertion and RAG-mediated ESC reintegration reactions (<https://github.com/Boyes-Lab/NGS-Analysis>). These data are shown in the new Extended Data Fig. 8a.

Next, to test if there is a direct link between increased ESC levels and the mutations associated with BCP-ALL and relapsed BCP-ALL, we analysed WGS and whole exome sequencing (WES) data from patients where we had previously determined RAG and ESC levels. Three groups, each of 14-15 patients, were analysed: (A) Patients with high *RAG1* expression plus high ESC levels; (B) those with high *RAG1* expression but low ESC levels and (C) those with low *RAG1* expression and low ESC levels (Extended Data Fig. 8b,c). The types of structural variants and mutated genes were compared. We find significantly more breaks at single cRSS (consistent with cut-and-run) in the patient cohort with both high *RAG1* expression and high ESCs compared to when *RAG1* expression and ESC levels are low. Importantly, we also find that SVs at genes that are frequently mutated in BCP-ALL or relapsed BCP-ALL, are markedly increased in the high *RAG1* expression/high ESC patient group compared to both of the other patient groups (Extended Data Fig. 8d, lower panel compared to top and middle panels). Moreover, there is a significant increase in SVs with a single cRSS at the breakpoint in the patients high *RAG1* expression and high ESCs as would be expected from cut-and-run mediated breaks. This is not observed in the high *RAG1* plus low ESC group, nor the low *RAG1* expression, low ESC group (Extended Data Fig. 8d upper and middle panels). These data therefore link the presence of high ESC levels to mutations at frequently mutated/relapse associated genes in BCP-ALL. The data are inconsistent with significant levels of SVs at relapse-associated genes being triggered by off-target RAG activity at pairs of CAC/heptamers within the same chromatin loop (10.1016/j.cell.2015.10.016; and doi: 10.1016/j.molcel.2016.06.034) nor RAG-mediated DNA insertions (doi: 10.1084/jem.20161638).

RAG cleavage at unusual DNA structures has been documented at major breakpoint region near *BCL-2* (doi: 10.1038/nature02355) resulting in translocation of *BCL-2* into the *IGH* locus. However, our analysis of SVs in the three groups outlined above with different levels of ESCs and *RAG1* expression did not identify SVs near *BCL-2* in any patient group. This contrasts with other frequently mutated genes where SVs are found (Extended Data Fig. 8d) and implies that the contribution of RAG cutting at unusual DNA structures (non-RSS sites) is relatively small compared to the SVs with one or two cRSSs at the breakpoint.

We have added these data as the new Extended Data Fig. 8a-d and describe this in the text (lines 340-347 and 368-378).

2. Deletional and inversional V(D)J recombination occurs in the *Igk* locus, as was described by the authors. In addition, intra-V_k events have also been identified as a result of overlapping 12/23 signals at V gene segments (doi: 10.1016/j.celrep.2019.11.088). Further, consecutive recombination events may occur at the same locus. These additional outcomes at the *Igk* locus should be discussed, and it should be clear that SJs chosen to study can only be formed through deletion events.

We have now added further discussion of the different recombination events that can occur at the *IGK* locus (lines 84-86, 175-176 and 1298-1300) and thank the Reviewer for asking us to do this. We were careful to exclude any SJs formed via non-deletional events from our analyses: Firstly, we excluded *IGK* SJs resulting from known inversional recombination events from the LAM-ESC data. Next, we carefully examined the LAM-recombination data to identify cases where the primary recombination event is inversional to determine if this might result in subsequent inversional recombination events at gene segments that usually undergo deletional recombination. Any such SJs that were likely within the genome were removed from our analyses. Thirdly, we reasoned that intra-KV recombination events will generate ESCs from in between V gene segments. This may increase the number of ESCs present but they will not be detected by our LAM-ESC or LAM recombination assays since the linear PCR primers bind to KJ (and LJ) regions. We have explained this in the text (lines 175-176 and 1298-1300) as well as in the legend to Supplementary Table 3 (lines 1519-1520). Finally, to verify that non-deletional SJs have not skewed our conclusions, we analysed our data using just *IGL* SJs, that are formed only via deletional recombination events. We observe the same phenomena as when SJs from both *IGL* and *IGK* loci are analysed, albeit with fewer SJs.

3. RecBCD was used to reduce linear DNA in the sample, leaving intact ESCs. It was not stated that this approach was taken with the BCP-ALL samples. Instead, it could be interpreted that the presence of ESCs was inferred by the detection of SJs in genomic DNA samples (untreated with RecBCD). This needs clarification. If not, then the BCP-ALL samples should be treated with RecBCD to confirm that the levels of SJs present are in ESCs in the BCP-ALL samples.

We removed linear DNA from mouse genomic DNA for Fig. 1d using RecBCD. In this case, the genomic DNA was prepared from freshly isolated IgM⁺ and IgG⁺ cells using conditions that minimise DNA shearing. For the most part, the SJs within circular ESCs remained intact following RecBCD treatment whereas linear genomic DNA (*Gapdh*) was reduced to low levels (1-3% of untreated DNA).

We received BCP-ALL samples as frozen cell pellets or as frozen DNA samples. Freeze-thawing causes double strand DNA breaks in both cellular and isolated DNA (doi: 10.1038/s41598-020-79670-8; doi: 10.1177/09636897211070239). Given that the ESCs analysed are between 20 kb and 1 Mb, unfortunately, few circular molecules remained unbroken following freeze-thawing and ESCs were at the limits of reliable detection following removal of linear DNA by RecBCD.

To test if ESCs are truly circular in BCP-ALL, we carried out two independent analyses.

Firstly, to test if unbroken ESCs in the original BCP-ALL samples are circular, we performed rolling circle amplification using Phi29 to amplify remaining circular DNA, followed by quantification of the SJ by ddPCR. This showed that SJs are significantly increased compared to untreated samples, for both BCP-ALL samples and mouse IgG⁺ DNA. By contrast, sonication of mouse IgG⁺ DNA eliminated Phi29 amplification of ESCs, verifying that Phi29 is amplifying circular DNA (original Extended Data Fig 6c-e, now d-f).

Secondly, we obtained fresh BCP-ALL patient samples taken at diagnosis and stored only at 4°C. High molecular weight genomic DNA was prepared under conditions that minimise DNA shearing and linear DNA was removed using RecBCD, as requested. Exactly as seen for mouse SJs (Fig. 1d), we find that SJs in BCP-ALL samples are present at very similar levels in RecBCD-treated and untreated DNAs, implying that they are indeed circular; by contrast, RecBCD reduces control (linear) *GAPDH* DNA to ~7% of the untreated level. The new RecBCD data have been added as Fig. 2c and described in the text (lines 142-147).

4. Increased expression of DNA repair proteins and replication proteins were detected in patients prone to relapse as compared to non-relapsed patients, which is discussed in relation to ESC replication. While a correlation with ESC levels may be present in some samples, it seems to be overstated in the manuscript, since the increased levels of these transcripts would be expected in relapsed samples that would occur with other cancer-driving mutations (independent of ESCs).

We have now compared expression of the DNA replication and repair proteins in patients with (a) high ESCs who later relapse, (b) low ESCs who later relapse and (c) low ESCs who do not later relapse, where low ESC levels are defined as being similar to those found in healthy blood (Fig. 3a). A significant increase in expression of *PCNA*, *POLE3*, *POLE4* and *RBX1* is observed in patients with high ESCs who later relapse but not in patients with low ESCs, regardless of whether the latter patients relapse. This suggests that increased expression of these genes is linked to higher ESC levels, rather than relapse.

By contrast, we find that *MLST8* and *RPA2* expression is higher in patients prone to relapse compared to those who do not, regardless of ESC levels, suggesting that increased expression of these genes may be linked to relapse. We have therefore removed these genes from the Figure and have removed discussion of MTORC1 from the manuscript and thank the Reviewer for asking us to do this important control.

The new data for *PCNA*, *POLE3*, *POLE4* and *RBX1* expression that better link expression of these genes with high ESC levels are now shown as the revised Fig. 4e and are described in the text (lines 233-242).

In response to Reviewer 2, we have also examined RNA-seq data for patients where we had measured ESC levels. RNA-seq is available (doi: 10.1038/s41375-022-01806-8) for four patients with high ESC levels and three with low ESC levels. A significant increase in expression of the genes encoding DNA repair and replication proteins (*PCNA*, *POLE3*, *POLE4* and *RBX1*) is observed in the patients with high ESC levels. Further GSEA studies show a highly significant increase in these groups of genes in the patients with high ESC levels. These data further strengthen the link between increased expression of DNA replication and repair genes with high ESC levels and have been added as the new Extended Data Fig. 4b and Fig. 4f and are described in the text (lines 233-242).

5. Appropriate statistical analyses are used in the results shown in the figures, and are well-described in the Methods section of the manuscript.

Minor points:

1. Figures 3B and 5C are not referred to in the text of the manuscript.

These were previously referred to (lines 176 and 259 in the original manuscript, now lines 184 and 276). We have rewritten the text to make it clearer where we refer to Fig. 3b.

2. Line 91 states that ratio of recombination junction to other genomic regions is maintained throughout B-cell development, but the Vk3-Jk5 data for splenic IgM⁺ cells (extended data Figure 1) is not consistent with this statement.

We have now analysed different aliquots of the same original DNA samples and show revised data where a higher (more consistent) recombination level is found for the splenic IgM⁺ cells for the Vk3-Jk5 recombination event (revised Extended Data Fig. 1a, now b).

3. Line 127 refers to Figure 2B, but this figure is not well annotated to match the statement in the text.

We have now reannotated Fig. 2B and Extended Data Fig. 1d and amended the text (line 133) and hope that this is clearer.

4. The model shown in Extended Data Figure 9 focuses exclusively on the Cut-and-Run mechanism. However, the authors could not rule out re-integration events in the manuscript.

As described in response to major point 1, we analysed the SVs in the WGS from 150 BCP-ALL patients and found that of the SVs with cRSSs at the breakpoint, >61% have a single cRSS, that could have been generated by cut-and-run or reintegration. We then ran our in-house pipeline (<https://github.com/Boyes-Lab/NGS-Analysis>) to measure the levels of re-integration and find that cut-and-run occurs >60-fold more frequently than re-integration, which is consistent with the >10-fold higher level of cut-and-run predicted by our *in vitro* analyses (doi: 10.1016/j.molcel.2019.02.025). These data have now been added at the new Extended

Data Fig. 8a and thus support our model in Extended Data Fig. 9 (now 10) that focusses on cut-and-run.

Referee #1 (Remarks on code availability):

The web pages could not be accessed with the URLs provided.

We apologize that the original Github token didn't work; a new (working) token has now been supplied that will remain active until 31/12/25. Reviewers may access the code directly by cloning the GitHub repo via their command-line using the command "git clone https://<username>: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@github.com/Boyes-Lab/<REPO_name>.git" where <username> is replaced with an active GitHub account name and <REPO_name> is replaced with one of the repositories, namely NGS-Analysis, Structural-Variants, and LAM-ESC-Recombination.

Referee #2:

In this paper, Gao et al., report that a type of extrachromosomal element generated from VDJ recombination, the excised signal circle (ESC), can be retained through series of cell divisions and accumulate at high copies, correlating with worse prognosis in cancer patients. The authors report that abundant signal joint sequences, which indicate ESCs are likely circular in structure, were detected in mouse B cells and in human leukemia samples. The author postulate that in patient samples, RAG1 expression is not suppressed and hence facilitates RAG/ESC complex formation, thereby contributing to genome instability. This is a fascinating concept which, if true, is important, and potentially of broad interest.

However, some major concerns are raised, which would need to be fully addressed, and if so, would greatly strengthen the paper and confirm its conclusions. The concerns arise in two areas – biological issues that for which conclusions are in need of some further validation and computational issues that could be addressed. As is, some of the major conclusions rest on relatively limited assays and data. Bolstering these claims through additional data, as described below, will be crucial.

Biological issues:

1. All data provided demonstrating the presence of ESCs were derived from sequencing or PCR approaches. Validating their presence by an imaging approach, such as DNA FISH would greatly strengthen the conclusions. Since at least some of them are reported by the authors to be about the same size as ecDNA, detection by DNA FISH should be very much feasible. This would also give a sense on the copies of ESC per cell at single cell level to support the claim that there are high levels of ESCs.

We have performed FISH analyses as requested. Since the samples used previously are only available as DNA or non-viable frozen cells, they unsuitable for metaphase FISH. We therefore prepared metaphase chromosome spreads from recently isolated patient samples that were surplus to diagnostic requirements and that had been expanded briefly in culture. We verified that ESC levels are similar between these samples and those used previously by ddPCR and confirmed that the ESCs are present as circular DNA, using RecBCD (Extended Data Fig. 7a). We furthermore verified that our FISH probes are specific using a control human cell line with a normal karyotype (Supplementary Fig. 4). Chromosome spreads from patient samples were then hybridised with locus-specific probes against regions that are normally excised upon ESC formation (KV2-4 to KJ1 & KJ5 to KDE for primary and secondary *IGK* recombinations, respectively and LV4-3 to LJ1 for *IGL* ESCs) as well as control regions that lie adjacent to *IGK* and *IGL* loci and remain chromosomal (Fig. 5f). DAPI staining of metaphase spreads confirmed the presence of extrachromosomal DNAs (Fig. 5d). EcDNAs are not detectable in BCP-ALL patient samples (doi: 10.1016/j.neo.2024.101025 and "Computing issues", point 5),

implying that these extrachromosomal DNAs are ESCs. However, most ESCs are smaller than ecDNAs and unfortunately, the DAPI-stained “ESCs” are lost during the high temperature washes required to obtain specific DNA FISH signals. Therefore, to investigate if the extrachromosomal entities are indeed ESCs, we compared the number of DAPI-stained extrachromosomal DNAs in metaphase spreads with ESC FISH signals in interphase cells. ESC levels in interphase cells were determined from the ESC-specific FISH signals (red or far red, Fig. 5f,g and Extended Data Fig. 7b) that do not overlap with the control regions that remain chromosomal (green signal). In patient P, with an ESC at similar levels to “high copy” ESCs found in patients who later relapse (albeit at the lower end of the range; compare Extended Data Fig. 7a with Extended Data Fig. 3b), up to seven ESCs (Fig. 5i; average 1.65) are observed in ~50% of the cells (Fig. 5e). By contrast, in patients M and N, with ESC levels similar to those who do not subsequently relapse (Extended Data Fig. 7a), only one (occasionally two) ESC(s) are observed in ~35% of cells (Fig. 5e-g and Extended Data Fig. 7b). These may be an underestimate due to the small size of some ESCs. Nonetheless, the numbers correspond well to the relative levels of DAPI-stained extrachromosomal DNAs in metaphase spreads (Extended Data Fig. 7c), implying that the latter are indeed ESCs. Moreover, the concentration of 3-7 ESCs in some cells is intriguing and is discussed further in terms of ESC replication in point 5 below and lines 302-305 in the manuscript.

Although the ESC numbers per cell are lower than ecDNAs, ESCs and ecDNAs cause cell dysregulation in different ways. EcDNAs frequently encode oncogenes and here, higher ecDNA copies confer a greater growth advantage. By contrast, ESCs trigger DSBs when complexed with RAG proteins, leading to mutations (doi: 10.1016/j.molcel.2019.02.025). Crucially, once ESC-mediated damage has occurred, that damage will be present in all daughter cells even if the daughter cells do not inherit the ESC. Not only this, but (a) ESC-mediated damage can accumulate through time by ongoing action of the RAG/ESC complex and this additional damage also will be passed onto daughter cells, regardless of whether the ESC is inherited (b) new ESCs can be generated in parallel in different cells, leading to damage in those cells and (c) different ESCs can trigger additional mutations in the same cell at different stages of cancer progression. Consequently, ESCs can confer a cumulative and lasting growth advantage at low numbers, consistent with the observed expansion of cells with recombination events corresponding to ESCs (Fig. 6d,e).

The new FISH data are now added as Fig. 5d-i and Extended Data Fig. 7a,b,c and described in the text lines 280-305 whereas the different modes of action of ecDNAs versus ESCs is discussed in lines 422-435.

2. Linear DNA digestion should also be performed in human samples to demonstrate circular structures, rather than only in mouse samples.

As described in response to a similar comment from Reviewer 1:

We removed linear DNA from mouse genomic DNA for Fig. 1d using RecBCD. In this case, the genomic DNA was prepared from freshly isolated IgM⁺ and IgG⁺ cells using conditions that minimise DNA shearing. For the most part, SJs within circular ESCs remained intact following RecBCD treatment whereas linear genomic DNA (*Gapdh*) was reduced to low levels (1-3% of untreated DNA).

We received BCP-ALL samples as frozen cell pellets or as frozen DNA samples. Freeze-thawing causes double strand DNA breaks in both cellular and isolated DNA (doi: 10.1038/s41598-020-79670-8; doi: 10.1177/09636897211070239). Given that the ESCs analysed are between 20 kb and 1 Mb, unfortunately, few circular molecules remained unbroken following freeze-thawing and ESCs were at the limits of reliable detection by qPCR, following removal of linear DNA by RecBCD.

To test if ESCs in BCP-ALL are truly circular, we carried out two independent analyses.

Firstly, to test if unbroken ESCs in the original BCP-ALL samples are circular, we performed rolling circle amplification using Phi29 to amplify remaining circular DNA, followed by quantification of the SJ by ddPCR. This showed that SJs are significantly increased compared to untreated samples, for both BCP-ALL samples and mouse IgG⁺ DNA. By contrast,

sonication of mouse IgG⁺ DNA eliminated Phi29 amplification of ESCs, verifying that Phi29 is amplifying circular DNA (original Extended Data Fig 6c-e, now d-f).

Secondly, we obtained fresh BCP-ALL patient samples taken at diagnosis and stored only at 4°C. High molecular weight genomic DNA was prepared under conditions that minimise DNA shearing and linear DNA was removed using RecBCD, as requested. Exactly as seen for mouse SJs (Fig. 1d), we find that SJs in BCP-ALL samples are present at very similar levels in RecBCD-treated and untreated DNAs, implying that they are indeed circular; by contrast, RecBCD reduces control (linear) *GAPDH* DNA to ~7% of the untreated level. These new RecBCD data have been added as Fig. 2c and described in the text (lines 142-147).

3. Wouldn't the paper be greatly strengthened if the authors were to examine ESC levels in BCP-ALL cell lines as well?

We agree and have looked for ESCs in two BCP-ALL cell lines (NALM-6 and REH) but unfortunately only negligible levels were detected. Specifically, we looked in both cell lines via ddPCR for the most commonly observed ESCs but found that ESCs are present at only negligible/background levels, similar to those found in HeLa cells where ESCs are not expected. We also used LAM-ESC to detect all possible light chain ESCs in REH cells but positive ESC reads were not obtained.

The lack of detectable ESCs in cell lines may be because the absence of a centromere results in imperfect ESC inheritance at mitosis. Most BCP-ALL cell lines were generated between 1973 and 1984; the large number of passages in culture since the original recombination event appear to have resulted in the original ESCs being lost, perhaps due to losses at mitosis gradually outstripping ESC replication.

In patients, secondary recombination from ongoing RAG activity (doi: 10.1038/leu.2016.142; doi: 10.1038/leu.2014.322) generates new ESCs. In most BCP-ALL cell lines, however, RAG1 or RAG2 proteins are at low to undetectable levels, substantially reducing the generation of new ESCs. Only REH cells have RAG levels close to those in patients but even here, our new control experiments showed that ESCs were not detected by LAM-ESC. Nanopore sequencing of REH DNA has shown *IGK* rearrangement to KDE and other chromosomal rearrangements (doi: 10.26508/lsa.202302481) that will substantially limit the number of possible secondary recombinations. Therefore, unfortunately, examining endogenous ESCs in cell lines is not feasible.

3. In ETV6::RUNX1+ BCP-ALL samples, based on different methods, SJs were detected in 6/9 samples (67%) by nested PCR (1-3 distinct SJ per patient). By WGS, and in a separate cohort, SJs were detected in 51/61 BCP-ALL samples (83%) (1-27 distinct SJs per patient). In lines 138-140, the authors claimed that ESCs could also be detected in other ALL, however their prevalence is not documented in these BCP-ALL subtypes. Despite ESCs could be frequently detected in ETV6:RUNX1+ BCP-ALL, the same cannot be concluded for other subtypes by simply showing a few cases evident of ESC.

Lines 138-140 refer to three BCP-ALL samples where we looked for ESCs via nested PCR, followed by cloning and Sanger sequencing (Extended Data Fig. 1d). In addition to these experiments, we detected ESCs via LAM-ESC in 71 other patients (Fig. 3), spanning 12 different BCP-ALL subtypes (Supplementary Table 4) including 16 *ETV6::RUNX1+*, 15 high hyperdiploid, 8 *PAX5*-altered, 7 *CRLF2* and 6 *KMT2A*. In fact, ESCs were present in every BCP-ALL subtype examined. We therefore believe it is reasonable to state that ESCs are found in other BCP-ALL subtypes. We have now clarified that ESCs in different subtypes were examined as part of the LAM-ESC experiments (lines 140-141).

4. The evidence for high copy of ESCs is limited. A lot of the assays rely on PCR, and the authors said it is not feasible to get quantitative data to gauge copy number from WGS. Fig 3a figure legend first states 'the number of copies of each SJ' was plotted, then later writes 'y axis shows the normalized level of each SJ' LAM-ESC. Further, these samples are normalized to ESC level detected in healthy blood, and the authors pointed out earlier in introduction that ESCs are diluted during cell division and therefore are usually expected to gradually fall to

negligible levels. I assume this is the case for healthy blood, meaning there are negligibly low levels of ESCs, or even none. In that case, a 30-fold increase would mean at most 30 copies (the limit of the Y-axis) from an unknown number of cells, and that would mean ESCs copy is very low at both bulk and single cell level instead.

We apologise for not originally specifying ESC numbers and have now addressed this, as described from point (d) onwards, below. Firstly, however, the Reviewer's comments have highlighted that we did not make some things sufficiently clear in the original manuscript. We have now amended the text to clarify the points below.

a) We show in Extended Data Fig. 3a that there is a good correlation between LAM-ESC reads and SJ copies measured by ddPCR. Therefore, contrary to the Reviewer's statement above, it is possible to gauge relative ESC levels from WGS, albeit not as accurately as by ddPCR (amended lines 186-189).

b) We did not normalise the SJ levels to those in healthy blood but instead compared the number of normalised SJ reads from LAM-ESC experiments between patient samples and those in healthy blood. Normalised SJ reads refers to the fact that we divided the reads for each SJ in a LAM-ESC experiment by the total number of sequencing reads for that experiment (now made clearer - lines 1162-1163). We also thank the reviewer for pointing out the anomaly in the legend to Fig. 3a regarding the plotting of normalised SJ reads; we have now corrected this (lines 1159-1162).

c) We find SJs are present at detectable levels via LAM-ESC in healthy human blood (Fig. 3a). Notably, 60-70% of B cells in healthy blood are naïve B-cells (doi: 10.1002/cyto.a.24507) that have undergone relatively few cell divisions following ESC formation and therefore large ESC losses are not expected. We have now performed ddPCR experiments that confirm the presence of SJs in healthy blood (new Extended Data Fig. 3b).

d), In response to the Reviewer's comments to better understand the numbers of ESCs present, we have performed ddPCR to compare ESCs levels in BCP-ALL samples and healthy blood. Exactly as found by normalised LAM-ESC reads, we find that the "high copy" ESCs (above the threshold shown in Fig. 3a) in patients prone to relapse are between 3- and 125-fold higher than the levels found in healthy blood. By contrast, ESC levels in patients who do not relapse are present at levels very similar to those found in healthy blood. These data have been added as the new Extended Data Fig. 3b and described in the text on lines 188-190.

e), It is not possible to accurately estimate total ESC copies at the bulk level by PCR as patients have multiple ESCs (an average 31 different ESCs for patients with ESCs above the threshold in healthy blood) and the poor conservation of RSSs means that it is not possible to generate probes that accurately measure all *IGK* and *IGL* ESCs by ddPCR or qPCR. To gain an insight into the number of individual ESCs, ESC levels were determined by ddPCR and compared with the total cell number, as determined by amplification of *GAPDH*. This showed that "high copy" ESCs, with levels above the threshold in healthy blood, are equivalent to an average of 0.68% of cells (range 0.25% to 2.74%) compared to 0.066% for individual ESCs in patients who do not relapse and 0.04% in healthy blood. Clearly these ESC numbers are low but as stated above, this will be higher once the multiple different ESCs per patient is considered.

f) Finally, ESC copies at the single cell level were measured by FISH. In patient P, with ESC levels similar to those who later relapse (albeit towards the lower end of the range of high copy ESCs; Extended Data Fig. 7a), between one and seven ESCs are observed per cell in ~50% of cells (Fig. 5e). By contrast, only one ESC is typically observed in ~35% of cells in patients with "non-relapse" ESC levels (Extended Data Fig. 7a). As noted in point 1, this could be an underestimate if the signal from small ESCs is weak and/or ESCs are occluded in the interphase nuclei. Nonetheless, these numbers are consistent with the estimates of bulk ESC levels from ddPCR measurements of individual ESCs (above).

Whilst ESC numbers overall are considerably lower than ecDNAs, the catalytic way in which ESCs trigger cell dysregulation, and the fact that mutations will accumulate over time (discussed in response to point 1), means that ESCs nonetheless impact on cancer progression. We have now added information about ESC numbers (lines 289-296) and discussed how this can impact on ALL progression (lines 422-435). In light of the Reviewer's

comments, we have also substituted “elevated” for “high” ESC levels where possible in the manuscript.

5. Line 178-181: “Given that the number of normalized sequencing reads correlates well with absolute ESC levels, as determined by droplet digital PCR (ddPCR, Extended Data Fig. 3a) these data imply that there is increased ESC replication and/or persistence in patients who subsequently relapse.” Similar claims were made throughout the text. Higher copy does not necessarily mean increased replication – are the authors trying to say they are amplified? This statement seems to contradict with the data provided in Line 239-242: ‘Whilst numerous ESCs that correspond to “major” recombination events are detected, implying that ESCs persist, no significant difference is detected in the per cent of such ESCs between patients who later relapse and those who do not (25.85% versus 28.3%, respectively).’

A number of pieces of evidence suggest that ESCs have indeed replicated.

Firstly, our studies in mice showed that ESCs remain constant during normal B-cell development. New ESCs are not generated after the pre-B cell stage as RAGs are no longer expressed (doi: 10.1016/1074-7613(95)90131-0). In the absence of ESC replication, ESCs would be expected to be diluted by approximately half at each cell division. B-cells undergo at least six cell divisions from the pre-B stage (when the light chain ESCs are generated) to the IgG⁺ stage (doi: 10.1084/jem.184.1.277). Without replication, ESCs would be expected to fall to 1.56%, or less, of their starting level by IgG⁺ cells. We find instead that the ratio of the ESC to its corresponding recombination junction (that is replicated as part of the genome) stays roughly constant (Fig. 1c), implying that the ESCs have replicated.

Secondly, in BCP-ALL samples, there are examples where ESCs are present at higher levels than the recombination event from which they were generated. This is shown in Extended Data Fig. 3d), including three new examples that were identified in response to point 4 above. Two ESCs are present at >9-fold their corresponding recombination event, one of which results from recombination to KDE. Once an allele has recombined to KDE, further secondary *IGK* recombination on that allele is not possible. Therefore, the increased ESC copies compared to recombination cannot be explained by loss of the original recombination event via secondary recombination. Instead, the data strongly support the idea that the ESC has replicated. We have now added these data to Extended Data Fig. 3d and describe it in the text (lines 198-200).

Thirdly, as discussed in greater detail in response to comment 6 (below), clonotype analyses show that the number of ESCs in a sample is greater than the number of different recombination events that could have generated that specific ESC. This is the case for 23 different ESCs (42.6% of those analysed), primarily but not exclusively for the higher copy ESCs in patients who later relapse. This clonotype analysis also implies that ESCs have replicated.

Fourth, FISH analyses show multiple ESCs per cell in some cells. Recombination to an individual KJ, LJ or KDE can only occur twice in any given cell – once on each allele. In the new Fig. 5i, we show FISH analysis of ESCs resulting from recombination to KDE; a maximum of two FISH signals involving recombination to KDE are possible per cell. Instead, using FISH probes that detect only KDE ESCs, we observe at least 7 ESCs (new Fig 5i). This is very difficult to explain without ESC replication.

Finally, in the new Extended Data Fig. 7d, we show bromodeoxyuridine (BrdU) incorporation into small extrachromosomal entities. Given that ecDNAs are not found in BCP-ALL (as discussed under “Computational issues, point 5” below), these data support the idea that ESCs indeed replicate.

The statement that ESCs replicate is not contradictory to the statement in (original) lines 239-242 but we agree that the latter statement could have been clearer. We see no difference in the number of major recombination events with corresponding ESCs in patients who later relapse versus those who do not. However, because ESCs are likely amplified/replicated in patients who later relapse, the **total number** of ESCs is greater in these patients. We have now re-written this to improve clarity (lines 255-258).

6. Lines 134-137: "Since each recombination event generates just a single ESC and there are millions of cells in these malignancies, the ability to detect SJs in these samples suggests that these ESCs have replicated and persisted through many cell divisions." Is this true to say so? If each cell from those millions of cells each generates one ESC, there could be millions of copies of ESC too, and does not necessarily mean they replicated and conserved in multiple cell divisions. None of the techniques used so far demonstrated ESCs are accumulated in high copies at the single cell level.

We believe this statement is true for a number of reasons:

Firstly, the new FISH data do show that ESCs accumulate at higher copies in some single cells than can be explained by recombination alone (new Fig. 5h,i) This strongly suggests that ESCs have replicated, as discussed above.

Secondly, each recombination event generates just a single ESC. In the absence of replication, this ESC is expected to be diluted by roughly half at each cell division (doi: 10.1084/jem.20060964). Therefore, after 10 cell divisions, in the absence of replication, the ESC would be diluted >1000-fold; after 20 cell divisions, this would be >1 million-fold. It would be very hard to detect an ESC after this level of dilution in the background of genomic DNA. However, we can detect ESCs that appear to correspond to a single recombination event, based on clonotype analysis (below), suggesting that the ESC has replicated.

Thirdly, each time V(D)J recombination generates an ESC, there is a permanent change in the genome where the V, (D) and J gene segments become joined. To generate as much antigen receptor diversity as possible, bases are added and deleted at random during the joining of the gene segments and this generates a unique sequence (clonotype) for each recombination event. Therefore, the number of times that a given ESC has been generated can be measured by the number of different clonotypes of the corresponding recombination event. If there are more ESCs than there are clonotypes for any given ESC/recombination, then this implies that the ESCs have replicated.

We investigated the ESC:clonotype ratio in BCP-ALL patient samples by capitalising on the LAM-recombination and LAM-ESC data where sequencing reads in the former can be interrogated to determine the number of different clonotypes. These data are plotted in the new Supplementary Fig. 3).

For the ESCs present above the threshold found in healthy blood, we find that there are more ESCs than clonotypes in 65% of cases. This implies that those ESCs have replicated. Indeed, there are two instances where the ESC is 20-100-fold higher than the number of clonotypes. There are also four instances where recombination is to KDE and where secondary recombination cannot have removed the original recombination event, further supporting the idea that ESCs above threshold levels in healthy blood, have replicated.

By contrast, when we examine ESCs below the threshold, with levels similar to those in healthy blood, ESC reads exceed the number of clonotypes in only 22% of cases (Supplementary Fig. 3). Therefore, these lower level ESCs appear to replicate less.

Fourth, we considered how often the same ESC is likely to be generated within the cell populations examined in our experiments. There are approximately 76 human KV gene segments that can recombine with 5 KJ gene segments or KDE, giving $76 \times 6 = 456$ combinations. For human *IgL*, there are 74 LV gene segments that primarily recombine with LJ1-3, giving $74 \times 3 = 222$ combinations. Thus, the chances of any given light chain recombination event/ESC being generated is 1 in 678 (excluding preferential RSS use).

We then examined our ddPCR data using *GAPDH* as a measure of cell number and find that typically ~750 cells were used per μ l. Therefore, per μ l we would expect $750/678 = 1.1$ recombination events (and corresponding ESCs) will involve the same V and J sequences. Of the 12 (relapse) cases examined where the ESCs are present above the threshold found in healthy blood, the ESC level per μ l (measured by ddPCR) is higher in eleven cases and at the expected level in the last case. However, when those same ESCs were examined in patients who did not go on to relapse, ESC levels per μ l were all below the expected level. Given that

only two of the ESCs involved V regions that are frequently used (Extended Data Fig. 7e), these data are consistent with replication of the ESC in patients who are prone to relapse. Collectively, these data point to ESC replication rather than the same ESC being generated multiple times.

7. The authors say RAG1/2 expression is not correlated with ESC copy (Ext Fig 3b), but the data is obtained from health blood samples, where ESCs are not expected to be found. What is the correlation of RAG1/2 expression with ESC copies in BCP-ALL samples then?

The data in Extended Data Fig. 3b (now c) show the correlation between ESC copies and RAG1/2 expression in BCP-ALL samples, exactly as requested by the Reviewer. We have rewritten the text to make this clearer (line 190-192) and apologise for not making this clear initially.

8. Line 182-184: 'and even in samples where RAG1 expression is very similar, high ESC copies are observed in patients who later relapse but not in those who do not (Fig. 3c).' I don't think this is appropriate to say based on just 10 samples.

We analysed 11 samples that have similar RAG1 expression levels. Of these four of the five patients who went on to relapse show elevated ESC levels whereas none of the six patients who remained in remission showed elevated ESC levels. This difference is significant ($P = 0.0156$) as determined by a Fisher's exact test and implies that ESC levels are not directly linked to the levels of RAG1 expression. This idea is supported by the data shown in Extended data Fig. 3b (now c) where no correlation is observed between the expression levels of RAG1 or RAG2 and ESC levels across ≥ 26 samples. We have now added the P value to Fig. 3c (lines 1172-1173) and hope that this, plus our explanation here, is acceptable.

9. In extended Fig 3c, the authors looked at the levels of recombination junction vs ESCs in patient samples. Provided on the y-axis is absolute copy, and the highest value is only 4. Contrasting to this, in ext fig 3d, the y axis is way wider in REC (6000-300 max) than SJ (3.5-18). Does that mean in most cases the copies of REC are even higher than SJ? Can the authors represent the data for each patient as in Fig 4b?

The level of recombination is an indicator of the number of cell divisions since the recombination event in question occurred. For Fig. 4b, we deliberately looked at SJs that correspond to recent recombination events to avoid complications from SJ losses at mitosis. Hence, the recombination levels are low. For Extended Data Fig. 3c (now 3d), the SJs examined also correspond to recent recombination events and hence the absolute recombination levels, as measured by ddPCR, are also low. By contrast, in Extended Data Fig. 3d and e (now e and f) we examined the levels of KV3-20-KDE and KV2-30-KDE recombination events and their corresponding SJs in 24 patients. Hence, there is a very broad range of recombination levels – from those that took place many years ago to those that have just occurred.

When the recombination event that generated the ESC took place many generations ago, the SJ:recombination ratio is lower than for recent recombination events due to SJ losses at mitosis and here, recombination is often higher than the SJ. Due to the complication that the SJ:recombination ratio is a measure of SJ replication, SJ losses at mitosis and the amount the cell has divided, we do not think it would be informative to plot the data in Extended Data Fig. 3e and f in this way. Instead, we compare the SJ and recombination levels (as measured by qPCR and normalised to *GAPDH*) between patients who go on to relapse and those who do not and find a significant increase in ESC levels, but not recombination, in the patients who go on to relapse.

Replotting Extended Data Fig. 3d in the way requested by the reviewer would lose the information about individual patient values. We therefore think it is more informative to keep this figure as is.

10. Line 205-206: Notably, of the recently generated ESCs, eleven are 206 identical in both patient groups. Where is the supporting data for this statement? Do the authors have sequencing tracks of the PCR products?

We now show sequencing tracks of the corresponding LAM-ESC data (new Extended Data Fig. 4a) that demonstrate that the same sequences are present in the patients who do and who do not later relapse. To complement this, we verified that the primers and probes used in the ddPCR experiments (shown in Fig. 4c) are highly specific via new control experiments that are shown in the revised Fig. 4c. This is described further in response to point 14 below and Reviewer 3 (point 3).

Minor processing of the ESC junction is apparent in the sequencing tracks for some ESCs (Extended Data Fig. 4a), but this processing accounts for just a few bp within 20 kb to 1 Mb of the ESC. Importantly, the same RSSs and adjacent sequences are amplified in the relapse and non-relapse samples studied in Fig. 4c.

11. The authors claimed “high ESC replication coupled to relapse”, however, what is the correlation between the various cell-intrinsic factors from RNA-seq data and ESC itself? The authors highlighted two information: 1) high levels of ESC could be detected in patients who relapsed, and 2) GSEA enrichment from transcriptome of relapse patients (note – these are an independent cohort) identified various cell intrinsic factors. These are merely parallel observations, and it is not possible to draw correlational conclusions between ESCs and the cell intrinsic factors. The data provided in fig 4e is misleading in a way because the levels of ESCs was not shown to be correlated with the expression of these genes. Provided in 4d and e are at most differences between relapse and non-relapse patients. Perhaps the authors could do a comparison between high ESC and low ESC patients instead.

We have now done further comparisons, including between patients with high and low ESC levels, as requested. Specifically, we compared expression of the DNA replication and repair genes in patients with (a) high ESCs who later relapse, (b) low ESCs who later relapse and (c) low ESCs who do not later relapse. A significant increase in expression of *PCNA*, *POLE3*, *POLE4* and *RBX1* is observed in patients with high ESCs who later relapse but not in patients with low ESCs, regardless of whether the latter patients relapse. This suggests that increased expression of these genes is linked to higher ESC levels, rather than relapse.

By contrast, we find that *MLST8* and *RPA2* expression is higher in patients prone to relapse compared to those who do not, regardless of ESC levels, suggesting that increased expression of these genes may be linked to relapse. We have therefore removed these genes from the Figure and have removed discussion of *MTORC1* from the manuscript and thank the Reviewer for asking us to do this important control.

The new data for *PCNA*, *POLE3*, *POLE4* and *RBX1* expression that better link expression of these genes with high ESC levels are now shown as the revised Fig. 4e and are described in the text (lines 233-242).

12. No other sample characteristics are examined when the presence of ESCs is linked to relapse. One alternate hypothesis could be that the samples in which ESCs are detected have a higher burden of disease, and therefore a higher tumor “purity” that allows detection of ESCs. In all analyses tumor purity should be investigated as far as possible and its effect investigated. We have now added the per cent leukaemic blasts, which is a measure of tumour purity, to Supplementary Table 4 that lists all patients and their BCP-ALL subtype. The vast majority of bone marrow samples analysed have a high tumour infiltration of >90%. Importantly, there is no significant difference between the % blasts at diagnosis in patients who later relapse (92.1%) and those who did not (93.4%). This is now described in the text (lines 194-195).

13. The analysis regarding differences in gene expression in RNA-seq data from 123 patients at diagnosis is speculative given that ESCs were not detected in this cohort. This analysis could be repeated in a cohort in which ESC detection was performed and RNA-seq is available.

We have now examined RNA-seq data for patients where ESC levels are known. RNA-seq is available (doi: 10.1038/s41375-022-01806-8) for four patients with high ESC levels and three with low ESC levels. A significant increase in expression of the genes encoding DNA repair and replication proteins is observed in the patients with high ESC levels (new Extended Data Fig. 4b). Further GSEA studies show a highly significant increase in these groups of genes in the patients with high ESC levels. These data further strengthen the link between expression of DNA replication and repair genes with high ESC levels and have been added as the new Fig. 4f.

14. Throughout the paper the authors could employ more controls e.g. cell lines/tissue samples that would not be expected to have any RAG activity or signal joint DNA present in them.

We have now included controls for our PCR reactions by using DNA and cDNA from mouse NIH 3T3 cells and human HeLa cells. Specifically, we have compared the levels of PCR products using mouse pre-B cell versus NIH 3T3 DNA/cDNA for Fig. 1b (added as the new Extended Data 1a) and 1e (added to the original Figure). We compared values obtained using HeLa cDNA versus cDNA from patient samples for Fig. 2a (added to the original Figure). We furthermore used HeLa DNA as a negative control for ddPCR primers and probes and have added these values to Extended Data Fig. 3b and Fig. 4c.

Computational issues:

1) It appears that in total data from 340 patients may have been used in various analyses in the study. However, the cohorts and exactly which data is used in which experiments in the study is sometimes unclear e.g. “>85 patient samples shows that...”. A figure/cohort diagram in supplementary outlining exactly which data is used for which analyses each of these would have been helpful. In addition, n numbers in the legends would be helpful throughout.

We now provide Supplementary Table 6 that shows which dataset was used for which analyses and have highlighted this in the text (lines 848-849). We have also checked the Figure legends and ensured that any missing N values have been added.

2) The bespoke computational tools created by the group and used to analyze the NGS data are not published. Furthermore, they are not well-described in the main text, have no dedicated methods sections, and are not accessible via the github links placed in the paper which appear to be dead. Without explanation or code, it’s very difficult to assess the veracity of the claims made.

We apologize that the original Github token didn’t work; a new (working) token has now been supplied that will remain active until 31/12/25. Reviewers may access the code directly by cloning the GitHub repo via their command-line using the command “git clone https://<username>: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@github.com/Boyes-Lab/<REPO_name>.git/” where <username> is replaced with an active GitHub account name and <REPO_name> is replaced with one of the repositories, namely NGS-Analysis, Structural-Variants, and LAM-ESC-Recombination. According to Nature’s policy on publishing code, we have added detailed Read me files that better describe each code to the Github repository. We have furthermore included worked examples for each of the codes.

3) There are no mention of sequencing characteristics e.g. coverage for each dataset or what parameters or thresholds were used.

The Amplicon sequencing reads are given for each LAM-Recombination and LAM-ESC experiment in Supplementary Tables 2 and 3, respectively. The parameters for mapping the reads are given as part of the scripts. We hope that this is acceptable. The coverage for the new whole exome sequencing is 100x and is given within the methods (line 782-783). The parameters for mapping structural variants using these data are given within the script for

analysing breakpoints at structural variants (<https://github.com/Boyes-Lab/Structural-Variants>).

5) It would be helpful if the authors could comment on whether existing computational tools for inferring ecDNA identify circular DNA in hematological malignancies and the reasons for the rate of detection relative to other cancer types. Have others described the presence of excised signal circles, if not, why not? What results do existing tools produce for samples included in this study.

A recent publication identified ecDNAs in other haematological cancers using AmpliconArchitect (doi: 10.1016/j.neo.2024.101025). However, no ecDNAs whatsoever were found in 44 BCP-ALL patient samples, although they were present at low levels in a single BCP-ALL cell line. It is impossible to know why this rate of detection is so low in BCP-ALL patients compared to other cancers. However, this may explain why ESCs have not been identified by other groups in BCP-ALL as the absence of ecDNAs would preclude further analysis for presence or absence of RSSs.

We have now analysed available WGS data for 20 of the BCP-ALL samples used in this study (13 from patients who subsequently relapsed and 7 from those who remain in remission) using AmpliconArchitect (doi: 10.1038/s41467-018-08200-y) and WGS from EGA accession code: EGAS00001006863. No ecDNAs were identified even though the sequencing depth meets the requirement for AmpliconArchitect. We suggest that circular DNAs, and specifically ESCs, were not identified by AmpliconArchitect due to individual ESC being present in only ~1-2%, or less, of cells (response to point 4(e), above). Consequently, distinct ESC sequences would not meet the increased copy number criterion required by AmpliconArchitect.

The low level of multiple different ESCs in BCP-ALL differs markedly from ecDNAs where the same sequence is amplified, giving many copies of oncogenes on the ecDNAs per cell and conferring a growth advantage.

As originally discussed (lines 355-365, now 422-435), ESCs can trigger mutations as part of RAG/ESC complexes. Consequently, a collection of different ESCs can confer a growth advantage even when present at low numbers and notably, the mutations generated will accumulate through time.

We have now added a note to the text stating that AmpliconArchitect did not detect ecDNAs in WGS from 20 of the BCP-ALL patients analysed here (lines 286-287 and 766-771). It was not possible to examine the presence of ESCs via Circle-seq (doi: 10.1038/s41588-019-0547-z) using available WGS data since none of the samples had been treated to remove linear DNA prior to sequencing.

Referee #3:

In their manuscript entitled “Excised DNA Circles from V(D)J Recombination Promote Relapsed Leukaemia” Zeqian Gao and colleagues present evidence that excision circles (ESC) in lymphocytes can be maintained during cell division and that through their recombination potential may promote genomic instability in leukemias. I am neither a B-cell nor a leukemia expert, but this seems like a very provocative and fascinating hypothesis, which as far as I can tell has the potential to change our current understanding of how ESC contribute to leukemia development.

Here are some suggestions to that may help further strengthen the manuscripts:

Major comments:

1. ESC were studied using PCR and sequencing. In order to validate their extrachromosomal presence, it would be crucial to perform metaphase FISH.

We have performed metaphase FISH analyses as requested. As outlined to Reviewer 2 (response 3), ESCs are not found in cell lines. Moreover, since the samples used previously are only available as DNA or non-viable frozen cells, they are unsuitable for metaphase FISH. We therefore prepared metaphase chromosome spreads from recently isolated patient samples that were surplus to diagnostic requirements and that had been expanded briefly in culture. We verified that ESC levels are similar between these samples and those used previously via ddPCR and confirmed that the ESCs are present as circular DNA, using RecBCD (Extended Data Fig. 7a). We furthermore verified that our FISH probes are specific using a control human cell line with a normal karyotype (Supplementary Fig. 4). Chromosome spreads from patient samples were then hybridised with locus-specific probes against regions that are normally excised upon ESC formation (KV2-4 to KJ1 & KJ5 to KDE for primary and secondary *IGK* recombinations, respectively and LV4-3 to LJ1 for *IGL* ESCs) as well as control regions that lie adjacent to *IGK* and *IGL* and remain chromosomal (Fig. 5f). DAPI staining of metaphase spreads confirmed the presence of extrachromosomal DNAs (Fig. 5d). EcDNAs are not detectable in BCP-ALL patient samples (doi: 10.1016/j.neo.2024.101025 and discussed in response to point 8, below), implying that these extrachromosomal DNAs are ESCs. However, most ESCs are smaller than ecDNAs and unfortunately, the DAPI-stained “ESCs” are lost during the high temperature washes required to obtain specific DNA FISH signals. Therefore, to investigate if the extrachromosomal entities are indeed ESCs, we compared the number of DAPI-stained extrachromosomal DNAs in metaphase spreads with the ESC FISH signals in interphase cells.

ESC levels in interphase cells were determined from the ESC-specific FISH signals (red or far red, Fig. 5e,f,g and Extended Data Fig. 7b) that do not overlap with the control regions that remain chromosomal (green signal). In patient P, with ESC levels similar to those in patients who later relapse (albeit at the lower end of the range; compare Extended Data Fig. 7a with Extended Data Fig. 3b), up to seven ESCs (Fig. 5i; average 1.65) are observed in ~50% of the cells. By contrast, in patients M and N, with ESC levels similar to those who do not subsequently relapse (Extended Data Fig. 7a), only one (occasionally two) ESC(s) are observed in ~35% of cells (Fig. 5e-g and Extended Data Fig. 7b). These may be an underestimate if the signal from small ESCs is weak and/or ESCs are occluded in the interphase nuclei. Nonetheless, the numbers correspond to the DAPI-stained extrachromosomal DNAs in metaphase spreads, implying that these are indeed ESCs (Extended Data Fig. 7c). Moreover, the concentration of 3-7 ESCs in some cells is intriguing and is discussed further in terms of ESC replication in point 5 below and in lines 302-305 in the manuscript.

Although the ESC numbers per cell are lower than ecDNAs, ESCs and ecDNAs cause cell dysregulation in different ways. EcDNAs frequently encode oncogenes and here, higher ecDNA copies confer a greater growth advantage. By contrast, ESCs trigger DSBs when complexed with RAG proteins, leading to mutations (doi: 10.1016/j.molcel.2019.02.025). Crucially, once ESC-mediated damage has occurred, that damage will be present in all daughter cells even if those daughter cells do not inherit the ESC. Not only this, but (a) ESC-mediated damage can accumulate through time by ongoing action of the RAG/ESC complex and this additional damage also will be passed to daughter cells, regardless of whether the ESC is inherited (b) new ESCs can be generated in parallel in different cells, leading to damage in those cells and (c) different ESCs can trigger additional mutations in the same cell at different stages of cancer progression. Consequently, ESCs can confer a cumulative and lasting growth advantage at low numbers, consistent with the observed expansion of cells with recombination events corresponding to ESCs (Fig. 6d,e).

The new FISH data are now added as the Fig. 5d-i and described in the text (lines 280-305) whereas the different modes of action of ecDNAs and ESCs is discussed in lines 422-435.

2. ESC replication is not directly shown. The current evidence is not strong. In vitro growth of such cells with nucleotide analogues, eg. BRDU, and measurements of analogue inclusion in replicated ESC would be necessary.

We have performed BrdU incorporation experiments as requested. Recently isolated patient samples were expanded briefly (~24 hours) prior to culturing in the presence of BrdU for 28 and 48 hours. Primary BCP-ALL cells in culture have a heterogeneous doubling time of 26 to 240 hours (doi: 10.1634/stemcells.22-6-1111); therefore, cells that are in metaphase after 28 hours cannot have undergone more than one S-phase in the presence of BrdU. Metaphase spreads were prepared, hybridised with mouse anti-BrdU antibodies, followed by FITC-labelled antibodies and DAPI staining of the chromosomes. This confirmed that BrdU is incorporated into extrachromosomal DNAs. Since ecDNAs have not been found in BCP-ALL (point 8, below), these extrachromosomal DNAs are likely to be ESCs (new Extended Data Fig. 7d and described in lines 305-311).

Interphase FISH experiments also strongly support ESC replication. In the new Fig. 5i, we detected ESCs generated by recombination to KDE. Given that there are only two KDE RSSs in any diploid cell (one per allele), the maximum number of ESCs resulting from excision of the region between KJ5 and KDE is two. We observe at least seven in one cell and three in each of two others, strongly implying that the ESCs have replicated (discussed in lines 302-305).

3. PCRs shown in Fig 1 need negative controls. How specifically do the primers amplify the recombination sequence? This is not clear. Ideally, the authors would perform the PCR also in other cell types (not B-cells) to show the negative signal there.

We have now performed negative control PCRs for the recombination and SJ primers, using mouse NIH3T3 DNA as the template. As expected, the values are very small indeed. Since the plots in Fig. 1b show the SJ/recombination ratio, adding the negative control values to this plot would be uninformative. Instead, we have added the negative control data to the new Extended Data Fig. 1a. To further verify the validity of our PCR primers, their specificity was checked by BLAST against the relevant genome and no template controls were routinely included in all PCR reactions as negative controls. Furthermore, in response to Reviewer 2, point 14, we have performed additional negative control PCRs for our RT-qPCR and ddPCR experiments.

4. If ESC behave like ecDNA, they should randomly segregate to daughter cells during mitosis. That should lead to differences in ESC copy numbers between cells. This again needs to be tested using FISH.

The FISH experiments described in response to point 1 have been extremely illuminating regarding ESC distribution between cells. For patient P with ESC levels similar to those who relapse, multiple ESCs are present in only ~7% of cells; two ESCs are found in ~13% of cells; about 29% of cells have one ESC whereas ESCs appear to be absent from the remainder (Fig. 5e). This therefore means that ESC copy number is inherently heterogeneous and in this respect, is similar to the random distribution of ecDNAs previously reported, leading to intratumoral heterogeneity (doi: 10.1038/s41588-022-01177-x).

We appreciate that this has not examined segregation at telophase/cytokinesis. However, given that in most cells where an ESC is present, there is just one ESC, these experiments are unlikely to be informative: The daughter cell will either inherit the ESC or it will not. Not only this, but even in patients with “high copy” ESCs, only ~7% of cells have three or more ESCs. Given the minimal number of BCP-ALL cells in M phase in our cultures (~<0.1%), and even fewer cells at telophase/cytokinesis, detecting the distribution of ESCs at cell division would be extremely challenging – and would involve trying to find approximately 1 in ~15,000 cells. Instead, we believe that the heterogeneous distribution of ESCs has already been demonstrated by the interphase FISH experiments.

5. LAM-HTGTS should also be performed in cells that do not express RAG1/2 in order to get a sense for the background signal of recombination between the Ig locus and other chromosomes. Some of these recombination events may represent artifacts of amplification

with the degenerate primer. In order to exclude this one would need to include negative controls.

LAM-HTGTS was used in our previous study to map the breakpoints caused by cut-and-run (doi: 10.1016/j.molcel.2019.02.025). We refer to it here only for Fig. 6c where we reanalysed our published LAM-HTGTS data for breaks at relapse-associated genes. We outline the controls we used for LAM-HTGTS at the end of this section. It seems that instead of LAM-HTGTS, the Reviewer may be requesting controls for LAM-recombination and LAM-ESC that were used extensively in the studies presented here. We have now carried out both LAM-ESC and LAM-recombination experiments in HeLa cells that do not express RAG1/2. In these control LAM-ESC experiments, reads that correspond to SJs from deletional recombination events were not detected. Although reads corresponding to one SJ from an inversional recombination event was obtained, this would have been removed from our analyses, that focussed solely on deletional SJs. Notably, the LAM-ESC protocol has an integral blocking step (Extended Data Fig. 2b) which removes any genomic (non-recombined) sequences prior to amplification and sequencing of the LAM-ESC products. Consequently, the assay is robust. For LAM-recombination, a blocking step is not included but nonetheless, only six sequencing reads were obtained using HeLa DNA, which is negligible compared to 5,000-10,000 reads that are typically obtained with patient DNA. These data therefore imply that artefactual amplification and cross contamination between experiments, is minimal.

A key step in the LAM-recombination and LAM-ESC protocols is to align the sequenced products to bespoke databases that only includes *IGK* and *IGL* recombination or ESC junctions, respectively. Therefore, even if translocations to other chromosomes occur and are amplified via LAM-recombination or LAM-ESC, these sequences will be filtered out at the analysis stage. We therefore believe that these experiments are robustly controlled to remove any non-Ig loci sequences. We have added a statement to the legend of Extended Data Fig. 2 to confirm that these negative controls have been performed (lines 1300-1302).

Concerning our published LAM-HTGTS experiments that were reanalysed to determine cut-and-run breaks at relapse-associated genes (Fig. 6c): These experiments were carried out in RAG-expressing (REH) cells; however, our controls were to omit lentiviruses that mimic ESCs from transductions. This prevents cut-and-run-mediated breaks which rely on the RAG-ESC complex. A second, more stringent control examined breaks in the presence of a lentivirus with a single consensus RSS as this allowed us to examine off-target RAG cutting. As shown in the original Fig. 6c, we see a significant increase in breaks at the relapse-associated genes in the presence of the ESC/SJ (cut-and-run) compared to the RSS (off target RAG cutting).

6. The RNA sequencing analysis seems very preliminary and vague without any functional support of the link between the differentially expressed genes and the amount of ESC. I would suggest omitting this part in the manuscript, or to perform perturbation experiments to establish/test functional links.

We have now established better links between increased expression of genes encoding DNA repair and replication proteins with high ESC levels and hope this is acceptable. Specifically, we compared expression of the DNA replication and repair genes in patients with (a) high ESCs who later relapse, (b) low ESCs who later relapse and (c) low ESCs who do not later relapse. A significant increase in expression of *PCNA*, *POLE3*, *POLE4* and *RBX1* is observed in patients with high ESCs who later relapse but not in patients with low ESCs, regardless of whether the latter patients relapse. This suggests that increased expression of these genes is linked to higher ESC levels, rather than relapse.

By contrast, we find that *MLST8* and *RPA2* expression is higher in patients prone to relapse compared to those who do not, regardless of ESC levels, suggesting that increased expression of these genes may be linked to relapse. We have therefore removed these genes from the Figure and have removed discussion of MTORC1 from the manuscript and thank the Reviewer for asking us to do this important control.

The new data for *PCNA*, *POLE3*, *POLE4* and *RBX1* expression that better link expression of these genes with high ESC levels are now shown as the revised Fig. 4e and are described in the text (lines 233-242).

In response to Reviewer 2, we have also examined RNA-seq data for patients where ESC levels are known. RNA-seq is available (doi: 10.1038/s41375-022-01806-8) for four patients with high ESC levels and three with low ESC levels. A significant increase in expression of the genes encoding DNA repair and replication proteins is observed in the patients with high ESC levels (new Extended Data Fig. 4b). Further GSEA studies show a highly significant increase in these groups of genes in the patients with high ESC levels. These data further support the link between expression of DNA replication and repair genes with high ESC levels and have been added as the new Fig. 4f.

7. If ESC re-integration is causing disease progression/relapse, shouldn't these integrations also clonally expand? Could the authors use their HT-GTS to test this and verify the impact these integrations have on the expression of tumor suppressor genes? Without one example of such tumor suppressor disruption, the association remains purely speculative. At least one example of such recombinations/integrations should be investigated in depth with further validation and functional readouts to support this claim.

Reintegration and cut-and-run are distinct mechanisms by which the RAG/ESC complex triggers genome instability. In response to Reviewer 1, we have compared the frequency of reintegration and cut-and-run and find that cut-and-run occurs >60-fold more frequently than reintegration in BCP-ALL (new Extended Data Fig. 8a).

Consistent with the relatively low tumorigenic potential of reintegration, since it was described in 2007 (doi: 10.1371/journal.pbio.0050043; doi: 10.1084/jem.20070583), there has been only one documented event of where ESC reintegration is linked to clonal expansion of an ALL. Here, a hotspot for reintegration was observed in the *ZNFP36L2* gene, corresponding to clonal expansion of a T-cell ALL (doi: 10.1186/s12943-023-01794-y). To identify a reintegration event in our patients that leads to disruption of a tumour suppressor gene that is the main mechanism leading to transformation/clonal expansion, would be a major project in itself.

Instead, we focussed on the mutations triggered by cut-and-run. We previously published that cut-and-run-mediated breaks co-localise with the breaks mapped in BCP-ALL patients and also occur at frequently mutated genes (doi: 10.1016/j.molcel.2019.02.025). Here, we show further that cut-and-run breaks also occur at relapse-associated genes (Fig. 6c), providing an underpinning mechanism for clonal expansion between diagnosis and relapse. Crucially, because cut-and-run is stochastic, different genes can be mutated in different cancer sub-clones. Hence, it is unlikely that a single mutation will lead to clonal expansion.

Instead, to address this comment and to achieve a more direct link between ESC-mediated damage and ESC levels, we analysed WGS and WES from patients where we had previously analysed *RAG* expression and ESC levels. We divided the patient samples according to *RAG* expression and ESC levels as follows: High *RAG1* expression plus high ESCs; high *RAG1* expression plus low ESCs and low *RAG1* expression plus low ESC levels (Extended Data Fig. 8b,c). We then compared the structural variants and mutated genes in these patient groups. We find significantly more breaks at single cRSS (consistent with cut-and-run) in the patient cohort with both high *RAG1* expression and high ESCs compared to patients with low *RAG1* expression and low ESC levels. Importantly, we also find that SVs at genes that are frequently mutated in BCP-ALL or relapsed BCP-ALL, are markedly increased in the high *RAG1* expression/high ESC patient group compared to either of the other patient groups (Extended Data Fig. 8d, lower panel compared to top and middle panels). Moreover, there is a significant increase in SVs with a single cRSS at the breakpoint in the patients high *RAG1* expression and high ESCs as would be expected from cut-and-run mediated breaks (Extended Data Fig. 8d, lower panel). This is not observed in the high *RAG1* plus low ESC group, nor the low *RAG1* expression, low ESC group (Extended Data Fig. 8d upper and middle panels). These data therefore link the presence of high ESC levels to mutations at frequently mutated/relapse associated genes in BCP-ALL.

As requested by the Reviewer, we have also identified a specific SV, that shows the hallmarks of cut-and-run, that undergoes expansion between diagnosis and relapse. Although we were limited by the number of patients for whom both diagnosis and relapse samples are available, an SV was observed within the *SYK* (spleen tyrosine kinase) gene of patient R-8 in WES data

at relapse. *SYK* is a critical signalling molecule in the pre-B cell receptor pathways, amongst other functions. Deletions, insertions and alternative splicing of *SYK* were reported in pro-B ALL (doi: 10.1038/sj.onc.1204515) and other studies implied it is required for pre-B receptor-induced cell cycle arrest in pre-B ALL (<http://doi.org/10.1182/blood.V116.21.4199.4199>). We find a 67 bp insertion at a single cRSS. Using patient samples taken at diagnosis and relapse, the region spanning the insertion was PCR amplified. At diagnosis, the insertion is present in 37% of the total amplified products, rising to 48.6% by relapse; by contrast, the non-inserted product shows a corresponding decrease between diagnosis and relapse. These data are consistent with insertion into one *SYK* allele that occurs before diagnosis and expansion of the cells with the inserted allele by relapse. Due to the multiple mutations associated with relapse, this is unlikely to be the sole driver of clonal expansion but it is consistent with cut-and-run triggering mutations that contribute to cancer progression. We have added these data as the new Extended Data Fig. 8b-e and describe this in the text on lines 368-378.

8. Did the authors observe any other ecDNA in leukemias using their methods? Is there evidence of RAG1 recombination signals (RSS) at the junctions of these ecDNA? This may be outside the scope of the manuscript, but would further support the importance of RAG1 recombination in leukemia development.

We did not detect ecDNAs using our script to detect ESCs in WGS data but this script specifically identifies discordant reads that map to the immunoglobulin and T-cell receptor loci. Notably, we did detect numerous ESCs that map to the T-cell receptor loci (*TCRA* and *TCRG*; Supplementary Table 1) that likely result from known mis-targeting of the V(D)J recombinase to TCR loci in BCP-ALL (doi: 10.1038/sj.leu.2401277).

In response to Reviewer 2, we analysed available WGS from 20 of the patients used in this study using AmpliconArchitect (13 from patients who subsequently relapsed and seven from those who remain in remission). No ecDNAs were identified in any patients even though the sequencing depth meets the requirement for AmpliconArchitect. We suggest that circular DNAs, and specifically ESCs, were not identified by AmpliconArchitect in these samples since any given ESC is present in only ~1-2%, or less, of cells (Reviewer 2, point 4, response part e). Consequently, each specific ESC sequence would not meet the copy number criteria for AmpliconArchitect.

The absence of ecDNAs in BCP-ALL means that it is not possible to examine the junctions for RSSs.

Our data are consistent with a recent publication where ecDNAs were not detected in 44 BCP-ALL patient sequences via AmpliconArchitect, although ecDNAs were found at low levels in a single BCP-ALL cell line. By contrast, ecDNAs were identified in other haematological cancers (doi: 10.1016/j.neo.2024.101025).

We have now added a note to the text stating that AmpliconArchitect did not detect ecDNAs in WGS from 20 of the BCP-ALL patients analysed here (lines 285-287 and 768-771).

Minor comments:

1. The publication by Koche et al. Nature Genetics 2020 proposed a similar mechanism through which ecDNA could contribute to cancer progression/relapse. They showed that ecDNA can re-integrate into the genome of cancers and that this occurs at sites of tumor suppressor genes and near oncogenes and that this affects the expression of these genes. The authors seem to have missed citing/acknowledging this paper in their introduction/discussion.

This has now been cited (lines 320-321) and we thank the reviewer for asking us to do this.

2. The method the authors use to isolate the ESC is similar to Circle-seq. The authors should mention this and acknowledge the work done on this method in the past.

This has also now been cited and discussed (lines 764-767).

Response to Reviewers comments

We thank all three referees for their positive comments and for their previous helpful, constructive comments. No remaining issues were apparent.